# Bad Values but Good Behavior: Learning Highly Misspecified Bandits and MDPs

## Abstract

Parametric, feature-based reward models are employed by a variety of algorithms in decision-making settings such as bandits and Markov decision processes (MDPs). The typical assumption under which the algorithms are analysed is realizability, i.e., that the true values of actions are perfectly explained by some parametric model in the class. We are, however, interested in the situation where the true values are (significantly) misspecified with respect to the model class. For parameterized bandits, contextual bandits and MDPs, we identify structural conditions, depending on the problem instance and model class, under which basic algorithms such as $\varepsilon$-greedy, LinUCB and fitted Q-learning provably learn optimal policies under even highly misspecified models. This is in contrast to existing worst-case results for, say misspecified bandits, which show regret bounds that incur a linear scaling with time horizon, and shows that there can be a nontrivially large set of bandit instances that are robust to misspecification.

## 1 Introduction

Sequential optimization over a set of decisions, e.g., actions in a multi-armed bandit and policies in an MDP, is often carried out by assuming a parametric model for the payoff of a decision that is learnt over time. Well-known instantiations of this approach are algorithms for structured multi-armed bandits, e.g., linear bandits (Rusmevichientong & Tsitsiklis, 2010) and generalized linear bandits (Filippi et al., 2010), linear contextual bandits (Chu et al., 2011), and, more generally, value function approximation-based methods for Markov Decision Processes (Sutton & Barto, 2018).

Learning algorithms that make decisions based on an estimated reward model are known to enjoy strong performance (e.g., regret) guarantees when the true rewards encountered are *perfectly realizable* by the model, e.g., Abbasi-Yadkori et al. (2011); Filippi et al. (2010); Jin et al. (2020). However, it is often the case that a parametric class of models is, at best, only an approximation of reality, succinctly expressed by the aphorism 'All models are wrong, but some are useful' (Box, 1976). But even if the rewards of, say, all arms in a multi-armed bandit are estimated with a large error, one may still hope to discern the optimal arm if its (admittedly erroneous) estimate ends up dominating those of the other arms. This begs the natural question: While the task of reward estimation can be fraught with (arbitrarily large) error under misspecified models, when (if at all) can the task of optimal action learning remain immune to it?

We initiate a rigorous study of the extent to which sequential decision-making algorithms based on reward model estimation can be robust to misspecification in the model. In particular, we are interested in characterizing the interplay between i) the actual (ground truth) rewards in a decision-making problem and ii) the reward model class used by the algorithm, and how it governs whether the algorithm can still learn to play optimally if the true rewards are not realizable by the reward model. In this respect, our specific contributions are as follows:

1. For misspecified linear bandits, we identify a new family of instances (reward vectors) that are elements of a robust region. Reward vectors in this region are characterized by an invariance property of the greedy action that they induce after being projected, with respect to any weighted Euclidean norm, onto the linear feature subspace (space of all instances that are realizable by features). This region, depending upon the feature subspace, can be non-trivially large, and need not be confined to within a small deviation from the subspace.

2. We prove that for any instance (i.e., the vector of true mean arm rewards) in the robust observation region, both (i) the $\varepsilon$-greedy algorithm, with least-squares parameter estimation and an exploration rate of $1/\sqrt{t}$ in each round $t$, and (ii) the LinUCB (or OFUL) algorithm, achieve $O(\sqrt{T})$ cumulative regret in time $T$.

3. We extend our characterization of robust instances to linear contextual bandits, for which we provide a generalization of the robust observation region. We show that both the $\varepsilon$-greedy and LinUCB algorithms for linear contextual bandits get $O(\sqrt{T})$ regret whenever the true, misspecified, reward vector belongs to this robust observation region.

4. We provide a structural criterion for a finite-horizon Markov Decision Process (MDP), together with a Q-value approximation function class, for which the fitted Q-iteration algorithm provably learns a (near) optimal policy in spite of arbitrarily large approximation error (in the $\infty$-norm sense).

We stress that our results pertain to the original algorithms (i.e., not modified to be misspecification-aware in any manner). Our analytical approach shows that they can achieve nontrivial sublinear regret even under arbitrarily large misspecification error[1]. This is in contrast to, and incomparable with, existing results that argue that, in the 'worst case' across all reward vectors that are a constant distance away from the feature subspace, any algorithm must incur regret that scales linearly with the time horizon, e.g., (Lattimore et al., 2020, Thm. F.1).

Our results help lend credence, in a rigorous sense, to the observation that reinforcement learning algorithms presumably equipped with only approximate value function models are often able to learn (near-) optimal behavior in practice across challenging benchmarks (Mnih et al., 2013; Lillicrap et al., 2015). They also show the precise structure of bandit problems that makes robustness possible in the face of significant misspecification.

**Illustrative Examples**    This paper's key concepts and results can be understood using two stylized examples:

**Example 1: A misspecified 2-armed (non-contextual) linear bandit.** Consider a 2-armed (non-contextual) bandit in which the vector of mean rewards of the arms (the "instance") is $\boldsymbol{\mu} = \left[\mu_1, \mu_2\right]^\top = \left[20, 3\right]^\top$ in $\mathbb{R}^2$ (marked by $\times$ in Fig. 1(a)). Suppose one attempts to learn this bandit via a 1-dimensional linear reward model in which the arms' features are assumed to be $\varphi_1 = 3$ and $\varphi_2 = 1$. It follows that (i) any (2-armed) bandit instance in this linear model is of the form $\boldsymbol{\Phi}\theta$, where $\theta \in \mathbb{R}$ and $\boldsymbol{\Phi} = \left[\varphi_1, \varphi_2\right]^\top = \left[3, 1\right]^\top \in \mathbb{R}^{2 \times 1}$, and corresponds to an element in the range space of $\boldsymbol{\Phi}$, and (ii) the instance $\boldsymbol{\mu}$ is misspecified as it is off this subspace[2].

For ease of exposition, we also assume that there is no noise in the rewards observed by pulling arms. In this case, the ordinary least squares estimate of $\theta$, computed at time $t$ from $n_1$ observations of arm 1 and $n_2$ observations of arm 2, is $\widehat{\theta}_t = \frac{n_1\varphi_1\mu_1 + n_2\varphi_2\mu_2}{n_1\varphi_1^2 + n_2\varphi_2^2}$. A key observation is that $\widehat{\theta}_t$ can always be written as a convex combination of $\mu_1/\varphi_1 = 6.7$ and $\mu_2/\varphi_2 = 3$: $\widehat{\theta}_t = \frac{n_1\varphi_1^2}{n_1\varphi_1^2 + n_2\varphi_2^2}\left(\mu_1/\varphi_1\right) + \frac{n_2\varphi_2^2}{n_1\varphi_1^2 + n_2\varphi_2^2}\left(\mu_2/\varphi_2\right)$, for any sampling distribution of the arms. The corresponding parametric reward estimate $\boldsymbol{\Phi}\widehat{\theta}_t$, must thus lie in the set $\{[3\theta, \theta]^\top : \theta \in [3, 6.7]\}$, which appears as the hypotenuse of the right triangle with vertex $(20, 3)$ in Fig. 1(a).

Note that if a greedy rule is applied to play all subsequent actions ($A_{t+1} = \mathrm{argmax}_{i \in \{1,2\}} \varphi_i^\top \widehat{\theta}_t$), then the action will be 1 since the point $\boldsymbol{\Phi}\widehat{\theta}_t$ will always be 'below' the standard diagonal $\mu_1 = \mu_2$ (the black line in Fig. 1(a)). Since action 1 is optimal for the (true) rewards $\boldsymbol{\mu}$, the algorithm will never incur regret in the future.

The instance above has a misspecification error significantly smaller than the reward gap ($2.75 < 17$). One can also find instances at the other extreme, e.g., $\tilde{\boldsymbol{\mu}} = [20, 18]^\top$ (marked by $\circ$ in Fig. 1(a)) for which the misspecification error is much larger than the gap ($8.5 > 2$), that remain robust in the sense of regret. Such instances (all of them colored green) fall outside the scope of existing work on misspecified bandits (Zhang et al., 2023), and we address them in our work.

**Example 2 (an extreme case of misspecification): A misspecified nonlinear bandit.** A rather extreme form of robustness in the face of arbitrary misspecification is depicted in Fig. 1(b). Here, the model class is a, tube-like set of radius[3] $\varepsilon$ in $\mathbb{R}^2$, defined as $\{(x, y) \in \mathbb{R}^2 : |y - x| = \varepsilon\}$, of effective dimension 1. Here, any bandit instance in the green-shaded region is robust. For example, for the instance $\boldsymbol{\mu} = [-10, 10]^\top$, the model estimate for any sampling distribution is $\left[-10\alpha + 10(1 - \alpha) - (1 - \alpha)\varepsilon, -10\alpha + 10(1 - \alpha) + \alpha\varepsilon\right]^\top$, for any any sampling fraction $\alpha$ of arm 1. This is a convex combination of the *extreme* points $[-10, -10 + \varepsilon]^\top$ and $[10 - \varepsilon, 10]^\top$, and thus a greedy strategy would result in arm 2 being pulled. One can show from similar arguments as above that almost all instances $\boldsymbol{\mu} \in \mathbb{R}^2$ yield sublinear regret under, say, $\varepsilon$-greedy sampling.

---

[1]The term 'misspecification error' is to be understood as the distance of the arms' reward vector to the model reward subspace.

[2]The $l_\infty$ misspecification error (deviation from subspace) of $\boldsymbol{\mu}$, for this example, is 2.75.

[3]If a connected set is desired, then one can replace the tube with a very thin and long rectangle stretched along the diagonal, with similar conclusions.

For a non-linear *parameterized* model class, consider the following extension of the previous example. The class is defined as the set of all $(x,y) \in \mathbb{R}^2$, which satisfies the following piecewise constraints

$$(x(t),y(t)) = \begin{cases} x_0 + 400t\vec{u}, & \text{if } 0 \leqslant t \leqslant 1/4 \\ x_0 + 100\vec{u} - (2t-1)2\varepsilon\vec{v}, & \text{if } 1/4 \leqslant t \leqslant 1/2 \\ x_0 + 100\vec{u} - 2\varepsilon\vec{v} - 100(4t-2)\vec{u}, & \text{if } 1/2 \leqslant t \leqslant 3/4 \\ x_0 - 2\varepsilon\vec{v} + 2\varepsilon(4t-3)\vec{v}, & \text{if } 3/4 \leqslant t \leqslant 1. \end{cases}$$

where $\vec{u} = (1,1)$ and $\vec{v} = (-1,1)$, and $x_0 = [-50-\varepsilon, -50+\varepsilon]^\top$. This represents a long narrow rectangle oriented along the $y = x$, with length $100$ and width $2\varepsilon$. Figure 1(b), can be considered as a zoomed-in picture of this model class.

Consider, again the bandit instance $[\mu_1, \mu_2] = [-10, 10]$ as in the previous example. If arm-1 has been sampled $\alpha_1$ fraction of times and arm 2 has been sampled $\alpha_2$ fraction of times, it is clear the the estimate $\widehat{t} = \frac{\alpha_1(40+\varepsilon) + \alpha_2(60-\varepsilon)}{400}$, can be written as a convex combination of $\frac{40+\varepsilon}{400}$ and $\frac{60-\varepsilon}{400}$. The corresponding model estimate is $\begin{bmatrix} -10\alpha_1 + (10-2\varepsilon)\alpha_2 \\ (-10+2\varepsilon)\alpha_1 + 10\alpha_2 \end{bmatrix}$, which itself is a convex combination of the extreme points $\begin{bmatrix} -10 \\ -10+2\varepsilon \end{bmatrix}$ and $\begin{bmatrix} 10-2\varepsilon \\ 10 \end{bmatrix}$, corresponding to the estimates $\frac{40+\varepsilon}{400}$ and $\frac{60-\varepsilon}{400}$ respectively.

The remainder of the paper formalizes this observation for a variety of decision problems (bandits, both contextual and otherwise, and MDPs), and algorithms that incorporate some form of exploration ($\varepsilon$-greedy and optimism-based). It explicitly characterizes the set of all (true) reward instances for which no-regret learning is possible.

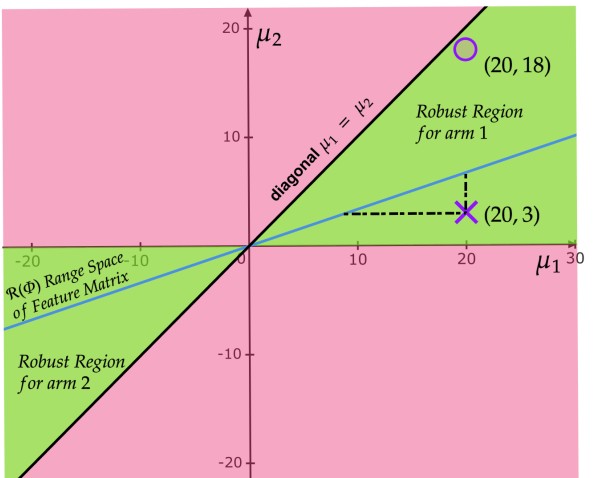

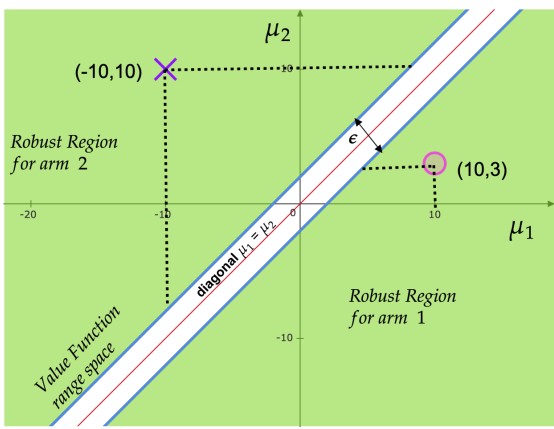

(a) Example 1: A 2-armed, noiseless bandit with 1-dimensional linear approximation. Each point in the plane represents the true rewards of both arms (the "instance"). The blue line is the set of instances expressed by the linear approximation. The green and red regions denote the robust regions and the non-robust regions for this linear function approximation. The misspecified instances $(20,3)$ and $(20,18)$ yield no regret under greedy arm selection based on an estimated linear model since any linear estimate of the rewards always has arm 1 dominating arm 2.

(b) Example 2: A function approximation class which is described as an $\varepsilon$-radius tube about the diagonal. We give a representative diagram for a $\mathbb{R}^2$ space corresponding to a bandit problem with two arms. We see that except for a measure zero set of bandit instances on the diagonal, which can be interpreted as both arms having the same rewards, all instances are robust.

Figure 1: Illustration of robust regions for two bandits with function approximation

## 2  Related Work

**Linear Bandits**   The text by Lattimore & Szepesvári (2020) provides a comprehensive text on the study of bandit algorithms. The classical works on stochastic linear bandits with finitely many arms have been studied by Auer (2002). Algorithms based on the principle of *optimism in the face of uncertainty* (Dani et al., 2008; Abbasi-Yadkori et al., 2011) and Thompson Sampling (Agrawal & Goyal, 2012; Abeille & Lazaric, 2017) are popular choices of algorithms which enjoy sub-linear regret even in the worst case of $\widetilde{O}(\sqrt{T})$. *Phased elimination* with optimal design based algorithms attain a regret of $\widetilde{O}(\sqrt{dT})$ as shown in Lattimore & Szepesvári (2020).

**Contextual Linear Bandits**    The bandit problem where the arm set changes at every time, described in the literature as the contextual bandit setup, has also been studied extensively. We mention the classic work of Auer (2002); Chu et al. (2011). The classic algorithms of SupLinRel (Auer, 2002) and SupLinUCB (Chu et al., 2011) are *elimination* based algorithms and enjoy a worst-case regret bound of $\widetilde{O}(\sqrt{dT})$. In practice, optimism-based algorithms like LinUCB (Abbasi-Yadkori et al., 2011) as well as Thompson Sampling (Agrawal & Goyal, 2013) enjoy sub-linear regret of $\widetilde{O}(\sqrt{T})$.

**Misspecified Bandits With Linear Regret**    Model misspecification in linear bandits was first studied in Ghosh et al. (2017). They pointed that in the presence of model error, the worst-case regret of LinUCB (Abbasi-Yadkori et al., 2011) is of linear order. They further showed that in a favorable case, when one can test the linearity of the reward function, the RLB algorithm (Ghosh et al., 2017) can switch between the linear bandit algorithm and finite-armed bandit algorithm to address the misspecification issue and achieve a $O(\min\{\sqrt{K},d\}\sqrt{T})$ regret. Under the definition of uniform $l_\infty$ model error $\varepsilon$, Lattimore et al. (2020) presented an algorithm based on the principle of *phased elimination* which achieves a worst case regret of $O(d\sqrt{T}+\varepsilon\sqrt{dT})$. They also showed that contextual linear bandits *LinUCB* achieve the same regret after modifying the confidence width using knowledge of the misspecification. In the same contextual bandit settings, Foster & Rakhlin (2020) showed a similar linear regret. They showed that under the assumption of an oracle regressor, the algorithm SquareCB (Foster & Rakhlin, 2020) suffers a regret of $O(\sqrt{dKT}+\varepsilon\sqrt{dT})$. Zanette et al. (2020b) has also shown a similar regret in the contextual setting. There has also been work based on altered definitions of misspecifications. Kumar Krishnamurthy et al. (2021) defines the misspecification as an expected square loss between the true reward class and the approximation function class and uses a model-based algorithm $\varepsilon$- FALCON, which again suffers linear regret. Foster et al. (2020) defines an empirical misspecification as observed by the data. However, these still suffer linear regret.

**Missecified Bandits With Sub-Linear Regrets**    Recently, there has been some work along a positive direction: to develop conditions and associated algorithms under which misspecified bandits can give sub-linear regret. This positive type of result is the main focus of this work. Recent works as that of Liu et al. (2023) analyzes the LinUCB algorithm when the sub-optimality gaps of the arms bound the misspecification. They show that when the misspecification is of low order, the algorithm enjoys sub-linear regret. Under a similar condition, Zhang et al. (2023) were able to extend the study to the contextual setting. They propose a phased arm elimination algorithm, which performs similarly to SupLinUCB (Chu et al., 2011) but requires knowledge of the sub-optimality gap.

**Markov Decision Processes**    Function approximation in Reinforcement Learning has had a rich history. As a comprehensive reference we direct the reader to the manuscript of Bertsekas & Tsitsiklis (1996). There has been recent interest in the finite-time analysis of function approximation in reinforcement learning. For example Bhandari et al. (2018) shows the convergence of policy evaluation with linear function approximators. Control problems, i.e., problems that require both policy evaluation and improvement, are notoriously hard to evaluate under function approximators, and well-known algorithms like $Q$-learning and SARSA are known to not converge with function approximations (Bertsekas & Tsitsiklis, 1996). Theoretically, there have been efforts to mitigate this problem, for example in Zou et al. (2019) analyses the sample complexity of SARSA with linear function approximators under a Lipschitz continuous policy improvement operator. On the other hand, there have been works on developing online algorithms founded on bandit literature that focus on the exploration-exploitation dynamics in an MDP. For example, Van Roy & Wen (2014) introduced RLSVI, a Thompson Sampling-based algorithm that has gone on to receive some attention in the recent past (Osband et al., 2016; Zanette et al., 2020a; Agrawal et al., 2021). However, these algorithms are based on a realizability and closedness assumption termed Linear MDPs. The framework of Linear MDPs has been popular in the recent literature on the online learning framework of Reinforcement Learning because of its amenability to theory. It has been shown that under the Linear MDP model algorithms enjoy sublinear regret, (Jin et al., 2020) and has been extended to general function classes under the realizability and closedness assumption of the Bellman Operator (Dann et al., 2022). However, without the assumption of realizability and closedness, the theory fails, in the sense that one is not able to show the algorithm learns an optimal policy. For example, Jin et al. (2020) shows that Least-Squares-Value-Iteration with UCB exploration bias suffers linear regret if the Linear MDP assumption is removed. Similarly Zanette et al. (2020b) shows a linear regret under a misspecification notion termed as Inherent Bellman Error. In this regard we believe our work is a first of its kind to show that standard algorithms, like fitted $Q$-learning under a behavioral policy can learn the optimal policy, even if the model is grossly misspecified.

# 3 Multi Armed Bandits

**Problem Statement**  We consider a $K$-armed bandit with mean rewards $\{\mu_i\}_{i=1}^K$. We assume each arm $i$ is associated with a known feature vector $\varphi_i \in \mathbb{R}^d$. We also have a given parametric class of functions serving to (approximately) model the mean reward of each arm; each function in this class is of the form $f_\theta : \mathbb{R}^d \to \mathbb{R}$ for a parameter $\theta \in \Theta \subset \mathbb{R}^d$, and applying it to arm $i$ yields the expressed reward $f_\theta(\varphi_i) \in \mathbb{R}$. Let us assume that the number of arms, $K$, is larger than the dimensionality of the parameter $\theta$, that is $K > d$ and, for ease of analysis, consider the set of features $\{\varphi_i\}_{i=1}^K$ to span $\mathbb{R}^d$. We denote the set of true mean rewards by the vector, $\boldsymbol{\mu} = \left[\mu_i\right]_{i \in [K]} \in \mathbb{R}^K$ and the feature matrix $\boldsymbol{\Phi} = \left[\cdots \varphi_i^\top \cdots\right]_{i \in [K]} \in \mathbb{R}^{K \times d}$. We use the standard matrix norm notation of $\|x\|_A^2$ to denote $x^\top A x$.

*Remark* 3.1 (**Linear Bandits**).  Our setting is rather general and covers a broad class of parametric bandits. For example, in linear bandits (Dani et al., 2008), it is assumed that the means are linear functions of the features, that is, there exists a $\theta^*$, such that $\boldsymbol{\mu} = \boldsymbol{\Phi}\theta^*$.

*Remark* 3.2 (**Misspecification**).  The novelty of our setting is that we allow the vector of the true rewards $\boldsymbol{\mu}$ to be arbitrary, without imposing a realizability condition like $\boldsymbol{\mu} \in \{f_\theta(\varphi_i) \forall i \in [K] : \theta \in \mathbb{R}^d\}$.

**Main Result**  We begin by defining *greedy regions* in $\mathbb{R}^K$ which characterize the reward vectors that share the same *unique* optimal arm.

**Definition 3.3** (*Greedy Region $\mathcal{G}_k$*).  Define by $\mathcal{G}_k$, for any $k \in [K]$, as the region in $\mathbb{R}^K$ for which the $k^{th}$ arm is the unique optimal arm, i.e., $\mathcal{G}_k \triangleq \left\{\boldsymbol{\mu} \in \mathbb{R}^K : \mu_k > \mu_i \forall i \neq k\right\}$.

Note that these $K$-dimensional greedy regions partition the entire $\mathbb{R}^k$ space into $K$ disjoint spaces. Any $K$-armed bandit with a unique optimal arm must belong to an unique greedy region $\mathcal{G}_i$, by definition.

For the purposes of clarity, let us fix our model class to be linear so that the least squares estimate has a closed-form solution. Denoting the sampling frequency of arm $i$ as $\lambda_i$, the least squares estimate $\widehat{\theta}$ can be written in closed form as $(\boldsymbol{\Phi}^\top \boldsymbol{\Lambda} \boldsymbol{\Phi})^{-1} \boldsymbol{\Phi}^\top \boldsymbol{\Lambda} \widehat{\boldsymbol{\mu}}$, where $\widehat{\boldsymbol{\mu}}$ is the vector of estimated sample means of the rewards, $\boldsymbol{\Lambda}$ is a diagonal matrix with $\lambda_i$s on its diagonal (Gopalan et al., 2016). We shall assume for the remainder of this work that $\boldsymbol{\Phi}^\top \boldsymbol{\Lambda} \boldsymbol{\Phi}$ is invertible and discuss in detail in Appendix E how to remove this assumption.

**Definition 3.4** (*Model Estimate under Sampling Distribution*).  For any bandit instance $\boldsymbol{\mu}$ in $\mathbb{R}^K$, we shall denote the model estimate of $\boldsymbol{\mu}$ under any sampling distribution, $\boldsymbol{\Lambda} = \mathrm{diag}(\{\lambda_i\}_{i=1}^K)$, where $\{\lambda_i\}_{i=1}^K \in \Delta_K$, the $K-1$ dimensional simplex, as $\mathbf{P}^{\boldsymbol{\Lambda}}(\boldsymbol{\mu}) \triangleq \mathrm{argmin}_\theta \left\|\boldsymbol{\Phi}\theta - \boldsymbol{\mu}\right\|_{\boldsymbol{\Lambda}^{1/2}}$.

The model estimate is an algorithmic dependent estimate. It is the canonical ordinary least squares estimate of $\boldsymbol{\mu}$, where we denote show the dependence on the sampling distribution $\{\lambda_i\}_{i=1}^K$.

Given a given bandit instance $\boldsymbol{\mu}$, assume, without loss of generality, that it belongs to the $k^{th}$ greedy region $\mathcal{G}_k$. If the projection, $\boldsymbol{\Phi}\mathbf{P}^{\boldsymbol{\Lambda}}(\boldsymbol{\mu})$ belongs to the same greedy region $\mathcal{G}_k$, then under the sampling distribution of $\boldsymbol{\Lambda}$, the function approximation of $\boldsymbol{\mu}$ would return the optimal arm under a greedy strategy. That is, $\mathrm{argmax}_{i \in [K]} \varphi_i^\top \mathbf{P}^{\boldsymbol{\Lambda}}(\boldsymbol{\mu})$ would result in the $k^{th}$ arm being pulled. With this motivation, we define *robust regions*.

**Definition 3.5** (*Robust Parameter Region*).  We define the $k^{th}$ *robust parameter region* $\boldsymbol{\Theta}_k$ as the the set of all $\theta$ such that the range space of $\boldsymbol{\Phi}$ restricted to $\boldsymbol{\Theta}_k$ lies in the $k^{th}$ greedy region $\mathcal{G}_k$. That is $\boldsymbol{\Theta}_k = \left\{\theta \in \mathbb{R}^d : \boldsymbol{\Phi}\theta \in \mathcal{G}_k\right\}$, for any arm $k$.

*Remark* 3.6.  Note that $\boldsymbol{\Phi}\boldsymbol{\Theta}_k$ is the set of all realizable bandit instances with optimal arm as $k$. Specifically, these are the instances which suffers no misspecification.

**Definition 3.7** (*Robust Observation Region*).  We define the $k^{th}$ *robust observation region* $\mathcal{R}_k$, as the set of all bandit instances with optimal arm $k$, such that the model estimate, computed under any sampling distribution, lies in the $k^{th}$ robust parameter region, $\boldsymbol{\Theta}_k$. That is,

$$\mathcal{R}_k = \left\{\boldsymbol{\mu} \in \mathcal{G}_k : \mathbf{P}^{\boldsymbol{\Lambda}}(\boldsymbol{\mu}) \in \boldsymbol{\Theta}_k \forall \boldsymbol{\Lambda} \in \mathcal{P}(\Delta_K)\right\},$$

where $\mathcal{P}(\Delta_K)$ is defined as

$$\mathcal{P}(\Delta_K) = \left\{\boldsymbol{\Lambda} = \mathrm{diag}(\{\lambda_i\}_{i=1}^K) : \{\lambda_i\}_{i=1}^K \in \Delta_k \wedge \boldsymbol{\Phi}^\top \boldsymbol{\Lambda} \boldsymbol{\Phi} \text{ is invertible}\right\},$$

where $\Delta_K$ is the $K-1$ dimensional simplex.

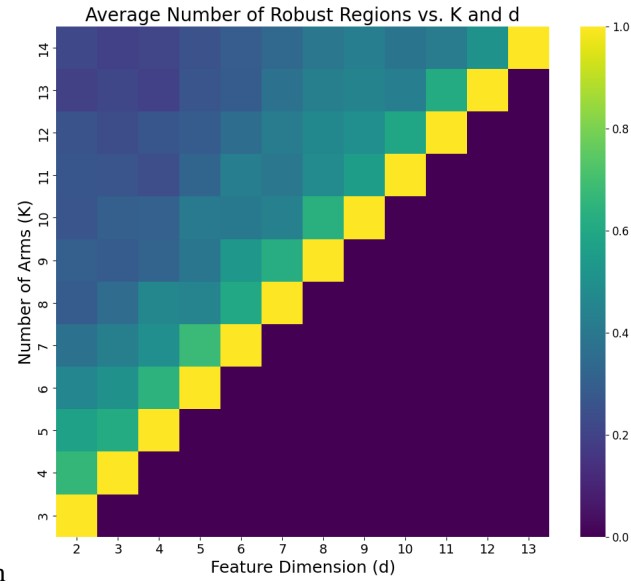

!h

**Figure 2:** For ambient dimension $K$, (the number of arms), we can expect a maximum of $K$ robust regions for any $K \times d$- dimensional feature matrix. For each $(K,d)$ pair, we sample 50 random feature matrices, and for each feature matrix, we comute the number of robust regions it can accommodate. We plot the number of robust regions as a fraction of the total number of arms $K$. We restrict our feature dimension $d < K$. For any $K$, the higher the feature dimension, the more robust regions it can express. However, interestingly, even low dimension feature matrices can express a non-trivial number of robust regions as can be seen on the left-upper corner of the figure.

*Remark* 3.8. In Appendix E we introduce the standard regularized least squares estimate, and remove the extra assumtion about the invertibility of $\mathbf{\Phi}^\top \mathbf{\Lambda} \mathbf{\Phi}$.

*Remark* 3.9. The robust region $\mathcal{R}_k$ is the set of all bandit instances with optimal arm $k$, whose projection on the feature space $\mathbf{\Phi}\theta$ always belongs to the greedy region $\mathcal{G}_k$. Note that, this observation is independent of any measure of misspecification, that is, the distance that the bandit instance is from the feature space.

*Remark* 3.10. We agree that our notion of $\mathcal{R}_K$ is rather a strong sufficient condition. In practice, we do not need to be bothered by all the sampling distributions in $\Delta_K$. The sampling distributions generated by an algorithm is sufficient to define robust regions for that particular algorithm.

*Thus, $\boldsymbol{\mu} \in \mathcal{R}_k$ is a sufficient condition that the greedy policy based on $\mathbf{\Phi}\mathbf{P}^{\mathbf{\Lambda}}(\boldsymbol{\mu})$ is the optimal under any algorithm-driven sampling strategy.*

*Remark* 3.11. Note that, from the definitions of robust regions, that these are functions on the feature matrix $\mathbf{\Phi}$. We omit this dependency in notation to avoid clutter, especially when the context is clear.

**Characterization of the Robust Region** In the class of linear models, $\mathcal{R}_k$ has a closed-form solution as shown in the following Theorem 3.12. This result allows us to compute robust observation regions given a feature matrix $\mathbf{\Phi}$ as illustrated In Appendix F.1.

**Theorem 3.12.** *$\boldsymbol{\mu}$ belongs to the* robust observation region $\mathcal{R}_k$ *if and only if every $d \times d$ full rank sub-matrix of $\mathbf{\Phi}$, denoted by $\Phi_d$, along with the corresponding $d$ rows of $\boldsymbol{\mu}$, denoted by $\boldsymbol{\mu}_d$, satisfies the condition that $\Phi_d^{-1}\boldsymbol{\mu}_d \in \mathbf{\Theta}_k$. In other words, $\boldsymbol{\mu} \in \mathcal{R}_k$ if and only if $\Phi_d^{-1}\boldsymbol{\mu}_d \in \mathbf{\Theta}_k$ for every $d \times d$ full rank sub-matrices of $\mathbf{\Phi}$ and the corresponding $d$ rows of $\boldsymbol{\mu}$.*

*Proof.* The proof uses a result of Forsgren (1996), presented in Lemma I.4, that for any any sampling distribution $\{\lambda_i\}_{i=1}^K \in \Delta_K$, the model estimate $\mathbf{P}^{\mathbf{\Lambda}}(\boldsymbol{\mu})$ lies in the convex hull of the *basic solutions* $\Phi_d^{-1}\boldsymbol{\mu}_d$. Thus for any $\boldsymbol{\mu} \in \mathcal{G}_k$

$$\boldsymbol{\mu} \in \mathcal{R}_k \iff \mathbf{P}^{\mathbf{\Lambda}}(\boldsymbol{\mu}) \in \mathbf{\Theta}_k \; \forall \; \mathbf{\Lambda} \in \mathcal{P}(\Delta_K) \iff \text{conv}\{\Phi_d^{-1}\boldsymbol{\mu}_d \; \forall \; \Phi_d \subset \mathbf{\Phi}\} \subset \mathbf{\Theta}_k \iff \Phi_d^{-1}\boldsymbol{\mu}_d \in \mathbf{\Theta}_k \; \forall \; \Phi_d \subset \mathbf{\Phi}.$$

The second condition follows from Lemma I.4. (We abuse notation to denote $d \times d$ full rank sub-matrices of $\mathbf{\Phi}$ by $\Phi_d \subset \mathbf{\Phi}$ and use conv to denote the convex hull.) The last assertion follows because $\mathbf{\Theta}_k$ is a convex set. $\square$

**Example** We return to our example 1 presented at the beginning (in Figure 1(a)) to highlight the definitions we have made so far. The greedy regions, $\mathcal{G}_1$ and $\mathcal{G}_2$, are the two half-spaces separated by the diagonal $\mu_1 = \mu_2$. From our choice of the feature matrix $\boldsymbol{\Phi}$ as $\begin{bmatrix} 3, 1 \end{bmatrix}^\top$, we note that for any parameter $\theta$ more than 0, the range space of $\boldsymbol{\Phi}$ belong to $\mathcal{G}_1$. Thus, $\boldsymbol{\Theta}_1$, the robust parameter region corresponding to arm 1, is the set of all positive scalars. Similarly, $\boldsymbol{\Theta}_2$, is the set of all negative scalars. $\mathcal{R}_1$, the robust observation region, corresponding to arm 1, is given by the set, $\{\boldsymbol{\mu} \in \mathbb{R}^2 : \mu_1 > \mu_2 > 0\}$. The robust observation region for arm 2, $\mathcal{R}_2$ is given by the set $\{\boldsymbol{\mu} \in \mathbb{R}^2 : \mu_1 < \mu_2 < 0\}$. This illustrates the existence of a large class of bandit problems, which are misspecified but robust for our model class.

A natural question that arises is: how frequently do the conditions of Theorem 3.12 hold in practice? While a precise characterization is challenging, we offer a few remarks that may provide some insight. First, it is straightforward to observe that for any feature matrix $\boldsymbol{\Phi}$, there always exists at least one robust region associated with it. This follows from the fact that the range space of $\boldsymbol{\Phi}$ must intersect with some greedy region $\mathcal{G}_i$ over a set of positive measure. Around any point in this intersection, one can construct a sufficiently small neighborhood that lies entirely within the greedy region and hence forming a robust region.

To further explore this question, we conduct an empirical study to estimate the average number of robust regions one can expect to encounter at a particular ambient dimension($K$) and feature dimensions ($d$). We plot the results in Figure 2. For a fixed ($K$,$d$) pair, we sample random feature matrices $\boldsymbol{\Phi}$, and compute the number of feasible robust regions, using Theorem 3.12. Particularly, we say that if there exists a $\boldsymbol{\mu}$ which satisfies the conditions of Theorem 3.12, then there exists a robust region. The maximum number of robust regions any arbitrary $\boldsymbol{\Phi}$ can have is $K$, which implies that all arms can be expressed correctly. We plot the average fraction of robust regions out of $K$ for various ($K$,$d$) pairs. The results indicate that even in settings with low feature dimensions and high ambient dimensions, a typical feature matrix $\boldsymbol{\Phi}$ can express a relatively high number of robust regions, while high dimensional features can almost express any arm correctly. This could suggest why even low dimensional feature spaces are able to give rise to reasonably good policies.

## 3.1 Instance Dependent Zero Regret Algorithms

We analyze two well-known algorithms $\varepsilon$-greedy (Sutton & Barto, 2018) and LinUCB (Abbasi-Yadkori et al., 2011) and show that if the bandit instance belongs to a robust region, then the algorithms enjoy zero regrets, even under misspecification. We assume the following conditions: that the noise of the observations is sub-Gaussian and that the instance $\boldsymbol{\mu}$ is an interior point of the robust region $\mathcal{R}_k$.

**Assumption 3.13** (*sub-Gaussian Noise*)**.** We shall assume that the $K$ armed bandit instance $\boldsymbol{\mu}$ is $1/2$ sub-Gaussian[4].

**Assumption 3.14** (*Interior Point*)**.** We shall assume that $\boldsymbol{\mu}$ is an interior point of $\mathcal{R}_k$, that is, there exists a $\delta > 0$, such that the $K$-dimensional open rectangle, with length $\delta$ and center $\boldsymbol{\mu}$, is entirely contained in $\mathcal{R}_k$. [5]

*Remark* 3.15. As can be observed from Figure 1(a), larger the separation between the arms ($\Delta_i$), larger the $\delta$ that can be chosen so that the open rectangle centered at $\boldsymbol{\mu}$,[6] is contained in $\mathcal{R}_k$. Thus, instances with larger sub-optimality gaps are more interior and hence more robust. This observation leads us to consider $\delta$ as a measure of robustness.

### 3.1.1 $\varepsilon$-Greedy Algorithm

We show that a misspecified bandit instance enjoys sublinear regret (*asymptotically*) under the $\varepsilon$-greedy algorithm, provided it belongs to the robust region as defined by the feature matrix. The proof is deferred to in Appendix A.1.

**Theorem 3.16.** *Given a feature matrix $\boldsymbol{\Phi}$ and a fixed bandit instance $\boldsymbol{\mu}$ satisfying Assumptions 3.13 and 3.14, the $\varepsilon$-greedy algorithm with $\varepsilon_t$ being varied as $1/\sqrt{t}$ over episodes, the cumulative regret, asymptotically enjoy sub-linear regret, that is $\lim\limits_{T \to \infty} \frac{Regret}{\sqrt{T}} \leqslant \Delta_{\max}$, where $\Delta_{\max}$ is the maximum sub-optimality gap, $\Delta_{\max} = \max_{i \in [K]} \mu_k - \mu_i$ and regret is the expected cumulative regret, $\sum_{t=1}^{T} \mathbb{E}[\mu_k - \mu_{A_t}]$.*

*Remark* 3.17. The proof uses the same technique as presented in Auer et al. (2002). The critical observation is that the least squares estimate $\widehat{\theta}_t$, in our notation $\mathbf{P}^{\boldsymbol{\Lambda}}(\widehat{\boldsymbol{\mu}}_t)$, is guaranteed to generate optimal play under the greedy strategy if the sample mean estimate $\widehat{\boldsymbol{\mu}}_t$ falls inside the robust region $\mathcal{R}_k$. The concentration of sub-Gaussian random variables ensures that given enough samples $\widehat{\boldsymbol{\mu}}_t$ will fall within a $\delta$-neighbourhood of the true rewards $\boldsymbol{\mu}$. The fact that $\boldsymbol{\mu}$ is an interior point of $\mathcal{R}_k$, ensures a that there exists $\delta$- neighbourhood about $\boldsymbol{\mu}$ that is contained in the robust region $\mathcal{R}_k$. Our experiments illustrated in Appendix F.1 corroborate this.

---

[4]The reason for choosing $1/2$ is purely for ease of calculation and can be replaced by any other constant.

[5]For the purpose of analysis we take the topology of $\mathbb{R}^K$, as open rectangles instead of open balls.

[6]We shall denote the open rectangle with length $\delta$, centered at $\boldsymbol{\mu}$ as $\mathbf{1}_\delta(\boldsymbol{\mu})$

### 3.1.2 LinUCB Algorithm

Our notion of robust instances captures the sublinear regret enjoyed by misspecified instances under the LinUCB algorithm.

The difficulty in the proof for LinUCB type algorithms is that while it is true that the sample estimates $\widehat{\mu}_t$ of robust instances $\mu$ fall within *robust observation regions* with high probability (because of the SaubGaussian nature), one cannot trivially conclude the same for the parameter estimates $\widehat{\theta}_t$. We rely on the following constant, as defined in Lemma 3.18, which can be thought of as the sub=optimality gap in the feature space for any robust instance $\mu$. The key observation is that, for any robust instance, the suboptimality gap in the feature space must also be (it could albeit be small but nevertheless) a finite positive constant.

**Lemma 3.18.** *If $\mu$ is an interior point of the robust observation region, that is, if $\mu \in \text{Int}(\mathcal{R}_{\text{OPT}(\mu)})$ then there exists a $\Delta_{\min} > 0$, such that for any sampling distribution $\{\lambda_{i(t)}\}_{i \in [K]} \in \mathcal{P}(\Delta_K)$ for time $t \geqslant 1$, we have $\varphi_{\text{OPT}(\mu)}^\top \mathbf{P}^{\mathbf{\Lambda}_t}(\mu) - \varphi_i^\top \mathbf{P}^{\mathbf{\Lambda}_t}(\mu) \geqslant \Delta_{\min}$ for any suboptimal arm $i$.*

*Remark* 3.19. Note that $\Delta_{\min}$ is a function of only the feature matrix $\mathbf{\Phi}$ and the bandit instance $\mu$ and is algorithm independent.

With this notion, we have the following regret bound, (proof in Appendix C, which follows a standard technique as in Abbasi-Yadkori et al. (2011)),

**Theorem 3.20.** *Given a feature matrix $\mathbf{\Phi}$ satisfying Assumptions C.2 and C.3, and any bandit parameter $\mu$ which is an interior point of the* robust observation region, $\mathcal{R}_{\text{OPT}(\mu)}$ *and satisfies Assumption C.1, the LinUCB algorithm achieves regret of the order $\tilde{O}(d\sqrt{t})$. That is, with $\beta_t(\delta)$ set as $2R^2 \log\left(\frac{(1+t/d)^{d/2}}{\delta}\right)$ for any $\delta > 0$, we have with probability at least $1 - \delta$*

$$\forall t > 1, \quad \sum_{s=1}^{t} \mu_{\text{OPT}(\mu)} - \mu_{A_s} \leqslant \frac{4\sqrt{t}R\Delta_{\max}}{\Delta_{\min}} \sqrt{\log\left(\frac{(1+t/d)^{d/2}}{\delta}\right)} \sqrt{\log(1+t/d)^d},$$

*where $\Delta_{\min}$ is as defined in Lemma 3.18 and $\Delta_{\max}$ is defined as the worst sub-optimal gap, that is, $\Delta_{\max} = \max_{i \in [K]} \mu_{\text{OPT}(\mu)} - \mu_i$.*

*Remark* 3.21. The theorem asserts that for any $\mathbf{\Phi}$ and a *fixed* bandit instance $\mu$ belonging to a robust region with respect to $\mathbf{\Phi}$, the LinUCB algorithm achieves a regret of $\tilde{O}(d\sqrt{T})$.

**Comparison to the works of Liu et al. (2023)**    The authors study the problem of misspecification under a robustness criterion which characterizes the misspecification to be dominated by the suboptimality gap. Under such a condition, they show that LinUCB enjoys $O(\sqrt{T})$ regret when the misspecification is of low order, specifically, it is of order $O\left(\frac{1}{d\sqrt{\log T}}\right)$. Note that this result is still non-trivial since the worst-case regret for LinUCB under uniform model error is $\rho T$ if $\rho$ is the misspecification error. However, we would like to address the following points while comparing our work : $(i)$ Our notion of robustness is significantly different from theirs as we are able to show examples which achieve sub-linear regret even if the misspecification error dominates the sub-optimality gap. $(ii)$ For our analysis of LinUCB we do not require the assumption of low misspecification error.

*Remark* 3.22. We would like to take this opportunity to paraphrase our contribution at this stage. While previous works have analyzed the same algorithms as presented here, it relied on the realizability of the instance by the features. We, show that this realizability assumption is not required to achieve the same results, for some special instances.

## 3.2 Non-Linear Function Approximation

In this subsection, we illustrate the robust conditions for parameterized function classes.

**Definition 3.23** (*Parameterized Function Class*). We consider a real-valued function class parameterized by $\theta \in \mathbb{R}^d$, that takes any arm $i \in [K]$ and returns a real value, $\{f_\theta : \{1,2,\cdots,K\} \to \mathbb{R} \,\forall \theta \in \mathbb{R}^d\}$ as the feature representation. We shall denote by $\boldsymbol{f}_\theta$ as the $K$ dimensional vector of $\{f_\theta(i)\}_{i \in [K]}$.

The model estimate calculated under a sampling distribution $\mathbf{\Lambda} = \text{diag}(\{\lambda_i\}_{i \in [K]})$, for the observation pairs $(\boldsymbol{f}_\theta, \mu)$ is defined as, $\mathbf{P}^{\mathbf{\Lambda}}(\mu) \triangleq \text{argmin}_\theta \|\boldsymbol{f}_\theta - \mu\|_{\mathbf{\Lambda}^{1/2}}$. We define the robust regions as

**Definition 3.24** (*Robust Parameter Region*). We define the $k^{th}$ *robust parameter region* $\mathbf{\Theta}_k$ as the set of all parameters $\theta \in \mathbb{R}^d$ such that $\boldsymbol{f}_\theta$ belongs to the $k^{th}$ greedy region $\mathcal{G}_k$. That is, $\mathbf{\Theta}_k = \{\theta \in \mathbb{R}^d : f_\theta(k) > f_\theta(i) \forall i \neq k\}$.

**Definition 3.25** (*Robust Observation Region*). We define the $k^{th}$ *robust observation region* $\mathcal{R}_k$ as the set of all bandit instances with optimal arm $k$, such the model estimate calculated under any sampling distribution, lies in the $k^{th}$ *robust*

*parameter region* $\Theta_k$. That is, $\mathcal{R}_k = \left\{ \boldsymbol{\mu} \in \mathcal{G}_k : \mathbf{P}^{\boldsymbol{\Lambda}}(\boldsymbol{\mu}) \in \Theta_k \, \forall \, \boldsymbol{\Lambda} \in \mathcal{P}(\Delta_K) \right\}$, where $\mathcal{P}(\Delta_K)$ is the set of all diagonal matrices whose elements belong in the $K-1$ dimensional simplex.

**$\varepsilon$-greedy algorithm**   We have the following sub-linear regret guarantee for the $\varepsilon$-greedy algorithm. The result follows from the analysis of the $\varepsilon$-greedy algorithm as given in Appendix A.1.

**Theorem 3.26.** *Given feature representation $\boldsymbol{f}_\theta$ and any bandit instance $\boldsymbol{\mu}$ satisfying Assumptions 3.13 and 3.14, the $\varepsilon$-greedy algorithm with $\varepsilon_t$ being varied as $1/\sqrt{t}$ over episodes, the cumulative regret, asymptotically enjoy sub-linear regret, that is $\lim_{T \to \infty} \frac{Regret}{\sqrt{T}} \leqslant \Delta_{\max}$, where $\Delta_{\max}$ is the maximum sub-optimality gap between arms, $\Delta_{\max} = \max_{i \in [K]} \mu_a - \mu_i$.*

### 3.3   Robust Features in Bandits: Extension of Example 2 (extreme case of misspecification)

In this subsection, we design a feature representation using a *highly nonlinear* function with a provably large robust region. The motivation for this feature representation arises from Figure 1(b), and we develop a higher-dimensional variant of the function. We observe that the greedy regions $\{\mathcal{G}_i\}_{i=1}^K$ partition $\mathbb{R}^K$ into $K$-disjoint partitions. Consider the subset $\mathcal{M}_i(\varepsilon) \subset \mathcal{G}_i$ defined as $\left\{ \boldsymbol{\mu} \in \mathbb{R}^K \text{ s.t } \mu_i > \mu_j + \varepsilon \, \forall \, j \neq i \right\}$, for a fixed $\varepsilon > 0$ and for any arm $i \in [K]$. We define the feature-representation $\mathcal{F}(\varepsilon)$ by the disjoint union of the manifolds $\mathcal{F}(\varepsilon) = \bigsqcup_{i=1}^K \partial \mathcal{M}_i(\varepsilon)$, where $\partial \mathcal{M}_i(\varepsilon)$ is the boundary of $\mathcal{M}_i(\varepsilon)$.

**Theorem 3.27.** *For the feature representation defined above as $\mathcal{F}(\varepsilon)$, the region $\bigsqcup_{i=1}^K \mathcal{M}_i(\varepsilon)$ is robust. Note that as $\varepsilon$ decreases, the robust region increases, and hence we can have an arbitrarily large class of robust bandit instances.*

*Proof.* Let us take an arbitrary $\boldsymbol{\mu} \in \bigsqcup_{i=1}^K \mathcal{M}_i(\varepsilon)$ and w.l.o.g. assume $\boldsymbol{\mu} \in \mathcal{M}_k(\varepsilon)$ for some $k \in [K]$. This is valid since the $\mathcal{M}_i(\varepsilon)$-s form a disjoint set. Note that for any sampling distribution, the projection of the chosen $\boldsymbol{\mu}$ on $\mathcal{F}(\varepsilon)$ belongs to $\partial \mathcal{M}_k(\varepsilon)$ since $\partial \mathcal{M}_k(\varepsilon)$ is the boundary of the set $\mathcal{M}_k(\varepsilon)$. Since $\partial \mathcal{M}_k(\varepsilon)$ is a subset of the greedy region $\mathcal{G}_k$, we see that the robust condition is satisfied. This completes the proof. $\qquad\square$

**Illustration in $3$-dimensions**   For $\mathbb{R}^3$ we illustrate the feature representation as follows. Robust regions are described as the disjoint union of $\mathcal{M}_1(\varepsilon), \mathcal{M}_2(\varepsilon)$ and $\mathcal{M}_3(\varepsilon)$, expressed as $\left\{ (x,y,z) \text{ s.t } x > y + \varepsilon, x > z + \varepsilon \right\} \sqcup \left\{ (x,y,z) \text{ s.t } y > z + \varepsilon, y > x + \varepsilon \right\} \sqcup \left\{ (x,y,z) \text{ s.t } z > x + \varepsilon, z > y + \varepsilon \right\}$ for a fixed epsilon. Thus, the function $\mathcal{F}(\varepsilon)$ is defined as the union of the boundaries $\left\{ (x,y,z) \text{ s.t } x = y + \varepsilon, x = z + \varepsilon \right\} \sqcup \left\{ (x,y,z) \text{ s.t } y = z + \varepsilon, y = x + \varepsilon \right\} \sqcup \left\{ (x,y,z) \text{ s.t } z = x + \varepsilon, z = y + \varepsilon \right\}$.

### 3.4   $\varepsilon$-Optimal Arms

In this subsection, we discuss the phenomenon of the robust region being large enough to ensure the greedy policy returns an $\varepsilon$-optimal arm. In the context of multi-armed bandits, an $\varepsilon$ optimal arm is an arm if $\varepsilon$ is larger than the minimum sub-optimal gap of arms $\Delta_{\min}$. We shall assume that the optimal arm is $k$ for the following discussion.

**Definition 3.28** ($\varepsilon$-*optimal set*). Let the arms whose rewards are at most $\varepsilon$ worse than the optimal arm be defined as $A_\varepsilon(\boldsymbol{\mu}) = \{ i \in [K] : \mu_i > \mu_k - \varepsilon \}$

Note that $A_\varepsilon(\boldsymbol{\mu})$ is a nonempty set as it contains $\mu_k$ by definition. The cardinality of $A_\varepsilon(\boldsymbol{\mu})$ can be more than one only if $\varepsilon$ is more than $\Delta_{\min}$, the minimum suboptimal gap of the bandit instance $\boldsymbol{\mu}$.

**Definition 3.29** ($\varepsilon$-*robust observation region*). We define the $k^{th}$ robust observation region as $\mathcal{R}_k^\varepsilon = \Big\{ \boldsymbol{\mu} \in \mathcal{G}_k : \boldsymbol{\Phi} \mathbf{P}^{\boldsymbol{\Lambda}}(\boldsymbol{\mu}) \in \bigsqcup_{i \in A_\varepsilon(\boldsymbol{\mu})} \mathcal{G}_i \, \forall \, \boldsymbol{\Lambda} \in \mathcal{P}(\Delta_K) \Big\}$, where $\mathcal{P}(\Delta_K)$ is the set of all diagonal matrices whose elements belong to the simplex $\Delta_K$.

The above definition implies that the greedy policy based on the estimate $\boldsymbol{\Phi} \mathbf{P}^{\boldsymbol{\Lambda}}(\boldsymbol{\mu})$ would return an $\varepsilon$-optimal arm for any sampling strategy $\boldsymbol{\Lambda}$. By definition, the robust region $\mathcal{R}_k$, is a subset of the $\varepsilon$-robust region $\mathcal{R}_k^\varepsilon$.

**Characterization of Robust Region**   As in Theorem 3.12, we can characterize the robust region in terms of the feature matrix. The proof follows along the same line as for Theorem 3.12.

**Theorem 3.30.** *$\boldsymbol{\mu}$ belongs to the $\varepsilon$-robust observation region $\mathcal{R}_k^\varepsilon$ if and only if every $d \times d$ full rank sub-matrix of $\boldsymbol{\Phi}$, denoted by $\boldsymbol{\Phi}_d$, along with the corresponding $d$ rows of $\boldsymbol{\mu}$, denoted by $\boldsymbol{\mu}_d$, satisfies the condition that $\boldsymbol{\Phi} \boldsymbol{\Phi}_d^{-1} \boldsymbol{\mu}_d \in \mathcal{G}_i$ for some specific $i$ in $A_\varepsilon(\boldsymbol{\mu})$. In other words, $\boldsymbol{\mu} \in \mathcal{R}_k^\varepsilon$ if and only if every $\boldsymbol{\Phi} \boldsymbol{\Phi}_d^{-1} \boldsymbol{\mu}_d \in \mathcal{G}_i$ for some specific $i \in A_\varepsilon(\boldsymbol{\mu})$*

*Remark* 3.31.  Since $\sqcup \mathcal{G}_i$ is not a convex region, we require every basic solution to lie in the same greedy region.

$\varepsilon$-**Optimal Arm Identification**    We demonstrate polynomial sample complexity for identifying an $\varepsilon$-optimal arm when it belongs to an $\varepsilon$-robust region.

**Theorem 3.32.** *Let an arbitrary bandit instance $\boldsymbol{\mu}$ belong in the $\varepsilon$-robust observation region and satisfy Assumptions 3.13 and 3.14 with width $\delta$. Then, for any $\alpha > 0$, a sampling strategy that samples each arm at least $T_i > \ln\left(\frac{K}{\alpha}\right)\frac{1}{2\delta^2}$ times would play an $\varepsilon$-optimal arm under the greedy policy with probability at least $1 - \alpha$.*

*Proof.* Note that because of Assumption 3.13, when each arm has been sampled for at least $\ln\left(\frac{K}{\alpha}\right)\frac{1}{2\delta^2}$, the sample estimate of $\widehat{\boldsymbol{\mu}}$ lies within a $\delta$ open-rectangle of $\boldsymbol{\mu}$ with probability at least $1 - \alpha$. From Assumption 3.14, this implies $\widehat{\boldsymbol{\mu}}$ lies is the $\varepsilon$-robust region and the result follows from the definition of the robust region. $\qquad\square$

# 4    Contextual Bandits

We consider the more general setting of a contextual bandit in which contexts are drawn from a finite set $\mathcal{X}$, and each context has finite arms in $\mathcal{A}$ giving rewards with means $\{\mu_{x,a}\}_{(x,a)\in\mathcal{X}\times\mathcal{A}}$. We assume that each context-arm pair $(x,a)$ is associated with a known feature $\varphi(x,a)$ in $\mathbb{R}^d$. We also have an available parametric class of functions that serves to (approximately) model the mean reward of each context-arm pair; each function in this class is of the form $f_\theta : \mathbb{R}^d \to \mathbb{R}$ for a parameter $\theta \in \Theta \subset \mathbb{R}^d$, and applying it to arm $a$ at context $x$ yields the expressed reward $f_\theta(\varphi(x,a)) \in \mathbb{R}$.

**Notation**    We denote the context space size and the action space size by X and A, respectively. We shall assume that the number of arms is larger than the dimension of the parameter, that is $A > d$ and, for ease of analysis, consider the set of features $\{\varphi(x,a)\}_{x\in\mathcal{X},a\in\mathcal{A}}$ to span $\mathbb{R}^d$. We shall denote the true reward mean as a vector in XA-dimension, $\boldsymbol{\mu} = \left[\mu_{x,a}\right]_{x\in\mathcal{X},a\in\mathcal{A}} \in \mathbb{R}^{\mathrm{XA}}$ and the feature matrix by $\boldsymbol{\Phi} = \left[\cdots\varphi(x,a)^\top\cdots\right]_{x\in\mathcal{X},a\in\mathcal{A}}$ an element in $\mathbb{R}^{\mathrm{XA}\times d}$. We shall denote by $\boldsymbol{\Phi_x}$, the context specific feature matrix, as the $A \times d$ sub-matrix of $\boldsymbol{\Phi}$ corresponding to the features $[\varphi(x,a)]_{a\in\mathcal{A}}$ for a fixed context $x$. Similarly we shall denote by $\boldsymbol{\mu_x}$, the context specific reward vector, as the A dimensional sub-vector of $\boldsymbol{\mu}$ corresponding to the rewards $[\mu_{x,a}]_{a\in\mathcal{A}}$ for a fixed context $x$. We shall use the notation of $\mathrm{OPT}(x)$ to denote the optimal arm at context $x$

*Remark* 4.1 (**Linear Contextual bandits**).    In linear contextual bandits (Chu et al., 2011), it is assumed that the mean rewards for a context-action pair $\mu_{x,a}$ is a linear function of the features $\varphi(x,a)$, that is there exists a $\theta^*$ such that $\boldsymbol{\mu} = \boldsymbol{\Phi}\theta^*$. Note that we make no such assumption.

**Main Result**    We define analogous concepts as those introduced in the bandits setting for the contextual setting. The critical observation is that we recover the multi-armed bandit setup for any fixed context.

**Definition 4.2** (*Greedy Region $\mathcal{G}$ for context $x$*).    Define by $\mathcal{G}_a^x$, for any context $x$ and arm $a$, as the region in $\mathbb{R}^{\mathrm{A}}$ for which the $a^{th}$ arm is the optimal arm at context $x$, that is $\mathcal{G}_a^x \triangleq \left\{\boldsymbol{\mu_x} \in \mathbb{R}^{\mathrm{A}} : \mu_{x,a} > \mu_{x,b} \, \forall b \neq a\right\}$.

We fix our model class to be linear, for which least square estimate can be written as $\widehat{\theta} = (\boldsymbol{\Phi}^\top \boldsymbol{\Lambda} \boldsymbol{\Phi})^{-1}\boldsymbol{\Phi}^\top \boldsymbol{\Lambda}\widehat{\boldsymbol{\mu}}$, where $\boldsymbol{\Lambda}$ is a diagonal matrix with sample frequencies of context-action pair $\{\lambda_{x,a}\}_{(x,a)\in\mathcal{X}\times\mathcal{A}}$ on the diagonal. Analogous to the bandit section, we define the *model estimate* as the least squares estimate calculated using observation pairs of $(\boldsymbol{\Phi}, \boldsymbol{\mu})$ under a sampling distribution of $\boldsymbol{\Lambda}$ as $\mathbf{P}^{\boldsymbol{\Lambda}}(\boldsymbol{\mu}) \triangleq \mathrm{argmin}_\theta \left\|\boldsymbol{\Phi}\theta - \boldsymbol{\mu}\right\|_{\boldsymbol{\Lambda}^{1/2}}$. We then define the *robust parameter region* and *robust observation region*.

**Definition 4.3** (*Robust Parameter Region for context* x).    We define the $a^{th}$ *robust parameter region* $\boldsymbol{\Theta}_a^x$ for a context $x$ as the set of all parameters $\theta$ such that the range space of the context-specific feature matrix, $\boldsymbol{\Phi_x}$, restricted to $\boldsymbol{\Theta}_a^x$ lies in the greedy region $\mathcal{G}_a^x$. That is, $\boldsymbol{\Theta}_a^x = \left\{\theta \in \mathbb{R}^d : \boldsymbol{\Phi_x}\theta \in \mathcal{G}_a^x\right\}$.

**Definition 4.4** (*Robust Observation Region for context* x).    We define the $a^{th}$ *robust observation region* $\mathcal{R}_a^x$ for context $x$ as the set of all contextual bandit instances $\boldsymbol{\mu}$ satisfying (i) the A armed bandit problem at context $x$ has arm $a$ as the optimal and (ii) the model estimate $\mathbf{P}^{\boldsymbol{\Lambda}}(\boldsymbol{\mu})$ calculated under any sampling distribution belongs in the $(x,a)^{th}$ robust parameter region $\boldsymbol{\Theta}_a^x$. That is,

$$\mathcal{R}_a^x = \left\{\boldsymbol{\mu} : \boldsymbol{\mu_x} \in \mathcal{G}_a^x, \mathbf{P}^{\boldsymbol{\Lambda}}(\boldsymbol{\mu}) \in \boldsymbol{\Theta}_a^x \, \forall \boldsymbol{\Lambda} \in \mathcal{P}(\Delta_{\mathrm{XA}})\right\},$$

where $\mathcal{P}(\Delta_{\mathrm{XA}})$ is defined as

$$\mathcal{P}(\Delta_{\mathrm{XA}}) = \left\{\boldsymbol{\Lambda} = \mathrm{diag}(\{\lambda_{x,a}\}_{(x,a)\in\mathcal{X}\times\mathcal{A}}) : \mathrm{diag}(\{\lambda_{x,a}\}_{(x,a)\in\mathcal{X}\times\mathcal{A}} \in \Delta_{\mathrm{XA}} \wedge \boldsymbol{\Phi}^\top \boldsymbol{\Lambda}\boldsymbol{\Phi} \text{ is invertible}\right\},$$

where $\Delta_{\mathrm{XA}}$ is the $(\mathrm{XA} - 1)$ dimensional simplex.[7]

---

[7]In Appendix E, we discuss how to carry over all previous discussion to the case where $\boldsymbol{\Phi}^\top \boldsymbol{\Lambda}\boldsymbol{\Phi}$ is not assumed invertible by using a regularizer, of the form $\boldsymbol{\Phi}^\top(\lambda I + \boldsymbol{\Lambda})\boldsymbol{\Phi}$.

In the following theorem, we show that $\boldsymbol{\mu}$ belonging in $\mathcal{R}^{\mathcal{X}}$ defined as $\bigcap_{x\in\mathcal{X}}\mathcal{R}^x_{\mathrm{OPT}(x)}$[8] is a sufficient condition that the greedy policy based on $\boldsymbol{\Phi P^{\Lambda}}(\boldsymbol{\mu})$ is the optimal strategy under any algorithm-driven sampling strategy for every context $x$.

**Theorem 4.5** (*Sufficient Condition*)**.** *If* $\boldsymbol{\mu}\in\mathcal{R}^{\mathcal{X}}$ *then* $\boldsymbol{\Phi_x P^{\Lambda}}(\boldsymbol{\mu})\in\mathcal{G}^x_{\mathrm{OPT}(x)}$ *for any context* $x$ *and under any sampling distribution. That is,* $\mathrm{argmax}_{a\in\mathcal{A}}\varphi(x,a)^{\top}\mathbf{P^{\Lambda}}(\boldsymbol{\mu})=\mathrm{OPT}(x)$ *for every context* $x$.

*Proof.* If $\boldsymbol{\mu}\in\mathcal{R}^{\mathcal{X}}$, then $\boldsymbol{\mu}\in\mathcal{R}^x_{\mathrm{OPT}(x)}$ for every context $x$. Thus, for every context $x$, we have $\boldsymbol{\mu_x}\in\mathcal{G}^x_{\mathrm{OPT}(x)}$ and for every sampling distribution, the model estimate $\mathbf{P^{\Lambda}}(\boldsymbol{\mu})\in\boldsymbol{\Theta}^x_{\mathrm{OPT}(x)}$. Thus, from the definition of $\boldsymbol{\Theta}^x_{\mathrm{OPT}(x)}$, we have that $\boldsymbol{\Phi_x P^{\Lambda}}(\boldsymbol{\mu})\in\mathcal{G}^x_{\mathrm{OPT}(x)}$. Therefore, the greedy algorithm is guaranteed to play optimally in every context. $\qquad\square$

**Characterization of Robust Region**   In the class of linear models $\mathcal{R}^{\mathcal{X}}$ can be described analytically in the following Theorem 4.6. We can use this result to compute the robust region for any given feature matrix $\boldsymbol{\Phi}$ as illustrated in Appendix F.2. The proof is presented in Appendix B.3.

**Theorem 4.6.** *Define* $\boldsymbol{\Theta}^{\mathcal{X}}\triangleq\bigcap_{x\in\mathcal{X}}\boldsymbol{\Theta}^x_{\mathrm{OPT}(x)}$. *A contextual bandit instance* $\boldsymbol{\mu}$ *belongs to the* robust observation region $\mathcal{R}^{\mathcal{X}}$ *if and only if for each* $d\times d$ *full rank sub-matrix of* $\boldsymbol{\Phi}$, *denoted by* $\Phi_d$, *along with the corresponding* $d$ *rows of* $\boldsymbol{\mu}$, *denoted as* $\boldsymbol{\mu}_d$, *satisfy* $\Phi_d^{-1}\boldsymbol{\mu}_d\in\boldsymbol{\Theta}^{\mathcal{X}}$. *That is* $\boldsymbol{\mu}\in\mathcal{R}^{\mathcal{X}}$ *if and only if* $\Phi_d^{-1}\boldsymbol{\mu}_d\in\boldsymbol{\Theta}^{\mathcal{X}}$ *for every* $d\times d$ *full rank sub-matrices of* $\boldsymbol{\Phi}$ *and the corresponding* $d$ *rows of* $\boldsymbol{\mu}$.

## 4.1  Instance Dependent No-Regret Algorithms

We analyze and prove that $\varepsilon$-greedy and LinUCB algorithms can achieve sub-linear regret on misspecified contextual bandits, provided they are in the robust observation regions. We assume the following settings: the noise is sub-Gaussian, each context has a positive probability of observation, and the instance $\boldsymbol{\mu}$ is an interior point of the robust region.

**Assumption 4.7** (*sub-Gaussian Noise*)**.** We shall assume that for any context $x$, the A armed bandit instance $\boldsymbol{\mu_x}$ is $1/2$ sub-Gaussian.

**Assumption 4.8** (*Context Distribution*)**.** Each context $x\in\mathcal{X}$ has positive probability $\mathbf{p}_x$ of observation.

**Assumption 4.9** (*Interior Point*)**.** We shall assume that $\boldsymbol{\mu}$ is an interior point of $\mathcal{R}^{\mathcal{X}}$, that is there exists a $\delta>0$, such the XA-dimensional open rectangle, with length $\delta$ and centre $\boldsymbol{\mu}$, is entirely contained in $\mathcal{R}^{\mathcal{X}}$.

### 4.1.1  $\varepsilon$-Greedy Algorithm

Analogous to the bandit setting, we can show that $\varepsilon$-greedy enjoys sub-linear regret for misspecified instances provided they are robust, as shown in the next theorem. We present the proof in Appendix B.1.

**Theorem 4.10.** *Given feature matrix* $\boldsymbol{\Phi}$ *and a* fixed *contextual bandit instance* $\boldsymbol{\mu}$ *satisfying Assumptions 4.7, 4.8 and 4.9, the* $\varepsilon$-greedy algorithm with $\varepsilon_t$ varied as $1/\sqrt{t}$, the cumulative regret, asymptotically enjoys sub-linear regret. That is, $\lim_{T\to\infty}\frac{Regret}{\sqrt{T}}\leqslant\Delta_{\max}$, where $\Delta_{\max}$ is the maximum sub-optimality gap between context-arm rewards, $\Delta_{\max}=\max_{x\in\mathcal{X}}\max_{a\in\mathcal{A}}\mu_{x,\mathrm{OPT}(x)}-\mu_{x,a}$.

*Remark* 4.11. The proof follows the same lines as in the Bandits section. Note that our regret guarantee in Theorem 4.10 does not depend on the misspecification error and depends only on the suboptimality gap. Our experiments illustrated in Appendix F.2 corroborate this.

### 4.1.2  LinUCB Algorithm

We analyze the LinUCB algorithm in the contextual setting and show that misspecified instances can enjoy sub-linear regret under the LinUCB strategy, provided they are robust. Much, like the Bandits scenario, the proof for the LinUCB algorithm relies on the observation, that the minimum sub-optimal gap in the value-function space has to be positive, for any robust instance.

**Lemma 4.12.** *If* $\boldsymbol{\mu}$ *is an interior point of the robust observation region, that is, if* $\boldsymbol{\mu}\in\mathrm{Int}(\mathcal{R}^{\mathcal{X}})$, *then there exists a* $\Delta_{\min}>0$, *such that under any sampling distribution* $\{\alpha(x,a,t)\}_{a\in\mathcal{A}}$ *at any time* $t\geqslant1$, *we have for any context* $x$,

$$\varphi_{x,\mathrm{OPT}(x)}^{\top}\mathbf{P^{\Lambda_t}}(\boldsymbol{\mu})-\varphi_{x,a}^{\top}\mathbf{P^{\Lambda_t}}(\boldsymbol{\mu})\geqslant\Delta_{\min}\,,$$

*for any sub-optimal arm* $a$ *at context* $x$.

---

[8]Here we use the notation $\mathrm{OPT}(x)$ to denote the optimal arm of context $x$

Note that $\Delta_{\min}$ is an instance dependent quantity, which depends on the $\boldsymbol{\mu}$ and the feature matrix $\boldsymbol{\Phi}$. With this we can arrive at our reegret bound as shown in the following theorem.

**Theorem 4.13.** *Under Assumptions D.1, D.2 and D.3, for any contextual bandit instance $\boldsymbol{\mu}$ lying in the robust region $\mathcal{R}^{\mathcal{X}}$ of a given feature matrix $\boldsymbol{\Phi}$, we have for any $t \geqslant 1$, and for any $\delta > 0$, the regret of LinUCB as,*

$$\sum_{s=1}^{t} \mu_{X_s, \mathrm{OPT}(X_s)} - \mu_{X_s, A_S} \leqslant \frac{4\sqrt{t}R\Delta_{\max}}{\Delta_{\min}} \sqrt{\log\Big(\frac{(1+t/d)^{d/2}}{\delta}\Big)} \sqrt{\log(1+t/d)^d},$$

*with probability at least $1-\delta$. Here $\Delta_{\max} = \max_{(x,a) \in \mathcal{X} \times \mathcal{A}} \mu_x \mathrm{OPT}(x) - \mu_{x,a}$.*

The proofs are present in Appendix D.

**Comparison with the results of Zhang et al. (2023)**   Zhang et al. (2023) study misspecified contextual bandits with the misspecification dominated by the minimum sub-optimality gap. We see this characterization as being more in line with the work of Liu et al. (2023). Under this robustness criterion, they develop a sophisticated algorithm (DS-OFUL) which they show to be regret-optimal. We believe their characterization of robustness is not directly comparable to our work. For example, our framework allows for the study of problems whose misspecification error can be (much) larger than the sub-optimality gap.

# 5   Markov Decision Processes (MDPs)

**Problem Statement**   We consider a finite horizon episodic MDP setting where at horizon/stage, $h \in [H]$, states are drawn from a finite set $\mathcal{S}$, and each state has finite actions in $\mathcal{A}$. The reward function at any stage $h \in [H]$ is a deterministic function of the state and action, that is, $R_h : \mathcal{S} \times \mathcal{A} \to \mathbb{R}$. The state transition kernel at any stage $h \in [H]$ is denoted by $\mathcal{P}_h(s'|s,a)$ and the actions are chosen according to a behavioral policy $\pi_b : \mathcal{S} \to \Delta_{\mathrm{A}}$. The optimal $Q_h^*$ value at any stage $h \in [H]$ is a function of the state-action pair, that is, $Q_h^* : \mathcal{S} \times \mathcal{A} \to \mathbb{R}$ and satisfies the optimal Bellman equation $Q_h^*(s,a) = T_h Q_{h+1}^*(s,a)$ where $T_h Q_{h+1}^*(s,a)$ is the Bellman Operator, $R_h(s,a) + \mathbb{E}\big[\max_{a' \in \mathcal{A}} Q_{h+1}^*(s',a')|s_h = s, a_h = a\big]$.

**Function Approximation**   In RL with function approximation, we employ a parametric function class $\mathcal{F}_h$ to approximate $Q_h^*$ values. Consider a function class $\big\{f_{\theta_h}\big\}_{h=1}^{H}$, where $\theta_h \in \mathbb{R}^{d_h}$ is a learnable parameter for $\{Q_h^*\}_{h=1}^{H}$. Each $f_{\theta_h}$ is a real-valued function that takes states and actions and returns an approximate value function $f_{\theta_h} : \mathcal{S} \times \mathcal{A} \to \mathbb{R}$. Consider the *Fitted-Q learning Algorithm* (Algorithm 1) as presented in Szepesvári (2022). At each stage $(s_h, a_h, r_h, s_{h+1})$ transitions are observed according to behavior policy $\pi_b$ to estimate $\theta_h$.

---

**Algorithm 1** Fitted-Q Learning

**Input**: Behavioral Policy $\pi_b$
**Output**: Updated parameters $\{\widehat{\theta}_h\}_{h=1}^{H}$ after $T$ rounds
  1:  Set $\{\mathcal{D}_h\}_{h=1}^{H} = \emptyset$.
  2:  **for** episode $t = 1$ to $T$ **do**
  3:      Set $\widehat{\theta}_{H+1} = 0$.
  4:      **for** Horizon $h = H$ to $1$ **do**
  5:          Fit $Q$-function with least squares regression

$$\widehat{\theta}_h = \operatorname*{argmin}_{\theta} \sum_{(s_h, a_h, r_h, s_{h+1}) \in \mathcal{D}_h} \Big(f_\theta(s_h, a_h) - r_h - \max_a f_{\widehat{\theta}_{h+1}}(s_{h+1}, a)\Big)^2$$

  6:      **end for**
  7:      Sample one episode $(s_1, a_1, r_1, \cdots, s_H, a_H, r_H)$ using $\pi_b$
  8:      Update the observation dataset $\mathcal{D}_h \leftarrow \mathcal{D}_h \cup \{(s_h, a_h, r_h, s_{h+1})\}$ for all $h \in [H]$.
  9:  **end for**

---

**Main Result**   We shall develop robust conditions and give PAC guarantees for the Fitted-$Q$ Learning algorithm without any *realizability* or *completeness* assumption. Particularly, we do not assume $Q_h^* \in \mathcal{F}_h$ for any $h$, nor do we assume that the Bellman Operator satisfies $T_h f_{h+1} \in \mathcal{F}_h$ for all $f_{h+1} \in \mathcal{F}_{h+1}$.

**Robust Condition** We note that any $Q_h^*(s,\cdot)$ is an element of $\mathbb{R}^A$ and hence belongs to the greedy region $\mathcal{G}_{\mathrm{OPT}(s)}^s$. The model estimate at stage h under the behavioral policy is defined as

$$\mathbf{P}^{\pi_b(h)}(\theta') = \arg\min_\theta \sum_{s,a,s'} \alpha_h^{\pi_b}(s,a,s')\Big(f_\theta(s,a) - r - \max_{a'} f_{\theta'}(s',a')\Big)^2, \text{ for any } \theta'.$$

Here $\alpha_h^{\pi_b}(s,a,s')$ is the true distribution of observing the pair $(s,a,s')$ under the behavioral policy $\pi_b$ at stage $h$ given by $\mathbb{P}^{\pi_b}\{s_h = s, a_h = a, s_{h+1} = s'\}$. The robust parameter region at stage h for a state $s$, is $\Theta_h^s = \{\theta : \boldsymbol{f}_\theta(s,\cdot) \in \mathcal{G}_{\mathrm{OPT}(s)}^s\}$. We denote the robust parameter region at stage h as $\Theta_h = \bigcap_{s \in \mathcal{S}} \Theta_h^s$. The *robust condition* can thus be described as

$$\mathbf{P}^{\pi_b(h)}(\theta_{h+1}) \in \Theta_h \ \forall \theta_{h+1} \in \Theta_{h+1}, \text{ for all } h \in [H]. \tag{1}$$

**PAC Guarantees** We assume that if a MDP $\mathcal{M}$, function class $\{f_{\theta_h}\}_{h=1}^H$ and a behavior policy $\pi_b$ satisfies the robust condition, then any behavior policy "close" to $\pi_b$ would also be robust.

**Assumption 5.1** (*Interior Point*). Given an MDP $\mathcal{M}$, a function class $\{f_{\theta_h}\}_{h=1}^H$, and a behavior policy $\pi_b$ that satisfies the robust condition, Equation 1, there exists a $\delta > 0$ such that any policy $\pi$ satisfying $|\alpha_h^\pi(s,a,s') - \alpha_h^{\pi_b}(s,a,s')| \leqslant \delta$ for all $(s,a,s')$ and at every stage $h \in [H]$, is robust, that is, satisfies $\mathbf{P}^{\pi(h)}(\theta_{h+1}) \in \Theta_h \ \forall \theta_{h+1} \in \Theta_{h+1}$ for all $h \in [H]$.

We present our PAC guarantee for the Fitted-$Q$ Learning algorithm for an MDP $\mathcal{M}$, function class $\{f_{\theta_h}\}_{h=1}^H$ and behavior policy $\pi_b$ for which it satisfies the robust condition.

**Theorem 5.2** (*Sample Complexity of Fitted Q-Learning*). *Let an MDP $\mathcal{M}$, function class $\{f_{\theta_h}\}_{h=1}^H$, and behavioral policy $\pi_b$ satisfy the robust condition 1 along with Assumption 5.1 with parameter $\delta$. Then for any $\varepsilon > 0$ and for $T \geqslant \ln\left(\frac{S^2 A H}{\varepsilon}\right)\frac{1}{2\delta^2}$ with probability more than $1 - \varepsilon$, the greedy policy, defined as $\pi_h^{greedy}(s) = \arg\max_a f_{\widehat{\theta}_T}(s,a)$ is the optimal policy* $\arg\max_{a \in \mathcal{A}} Q^*(s,a)$.

*Proof.* Let $n_h(s, a, s', t)$ denote the number of times the transition $(s_h = s, a_h = a, s_{h+1} = s')$ is observed till time $t$ under the behavioral policy $\pi_b$. Since every trajectory is sampled independently we have $\mathbb{E}\left[\frac{n_h(s,a,s',T)}{T}\right] = \mathbb{E}\left[\frac{\sum_{i=1}^T \mathbb{1}\{s_{hi}=s,a_{hi}=a,s_{h+1,i}=s'\}}{T}\right] = \alpha_h^{\pi_b}(s, a, s')$. Thus, from Hoeffding's Inequality, we get $\mathbb{P}\left\{\left|\frac{n_h(s,a,s',T)}{T} - \alpha_h^{\pi_b}(s,a,s')\right| > \delta\right\} \leqslant 2\exp(-2\delta^2 T)$. Taking a uniform bound over all $(s,a,s')$ observations and all $H$ stages, we find that the Assumption 5.1 is not satisfied with probability less than $S^2 A H \exp(-2\delta^2 T)$. $\square$

## 5.1 Example of a Misspecified but Robust (Deterministic) MDP

In this subsection, we illustrate robust conditions for a simple two-stage deterministic MDP and a theoretical feature representation under a uniform behavioral policy. For a stochastic MDP under an arbitrary behavioral policy, we present an analysis of robust conditions in Appendix H.

**MDP description** We present a simple two-stage MDP, $\mathcal{M}$, with three states, each having two actions as shown in Figure 3(b). The state transitions are deterministic based on the actions. At stage $h = 1$, the process starts at state $s_1$ and, based on the action, gets the associated reward and moves on to either state $s_2$ or state $s_3$. At each of the subsequent states, one chooses one of two available actions again, observes the reward, and the process ends. The rewards are such that $r_{11} > r_{12}$, $r_{21} > r_{22}$ and $r_{31} > r_{32}$. To ensure that employing a myopic greedy strategy fails, we require that the optimal policy at each state be $\pi^*(s_1) = a_2$, $\pi^*(s_2) = a_1$ and $\pi^*(s_3) = a_1$.

**Behavior Policy** We assume the behavior policy, $\pi_b$, to sample uniformly across all actions at any state $s$. This ensures that the approximate values are orthogonal projections to the function space.

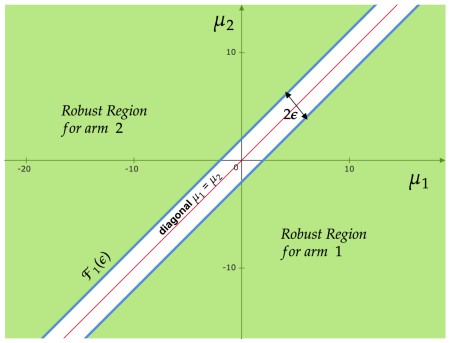

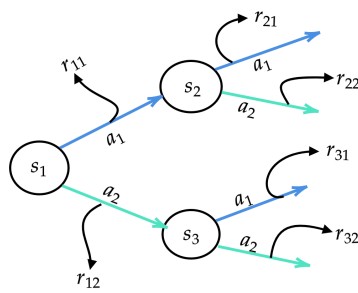

(a) A function approximation feature class which is described as an $\varepsilon$-radius tube about the diagonal. We give a representative diagram for a $\mathbb{R}^2$ space corresponding to a bandit problem with two arms. All instances are robust except for a measure zero set of bandit instances on the diagonal, which can be interpreted as both arms having the same rewards.

(b) A 2 stage deterministic MDP with three states and each state having two actions. The rewards are ordered as $r_{11} > r_{12}, r_{21} > r_{22}, r_{31} > r_{32}$. However, the rewards are designed such that the optimal action in state $s_1$ is $a_2$, because $r_{31}$ is significantly higher than $r_{21}$.

Figure 3: An example of a MDP and a function class we designed to approximate the $Q$ value. The optimal $Q^*$ values are misspecified in the function class, yet we can learn the optimal policy using this function class.

**Function Class**   We choose a function class that is a tube of radius $\varepsilon$ about the diagonal of the $\mathbb{R}^{\mathrm{SA}}$ space. At stage $h = 2$, the function approximation class is defined by

$$\mathcal{F}_2 = \left\{ [x, x, x, x]^\top + \varepsilon \frac{\boldsymbol{v}}{\|\boldsymbol{v}\|_2} \, \forall \boldsymbol{v} \in \left( [1,1,1,1]^\top \right)^\perp \forall x \in \mathbb{R} \right\}, \text{ for a fixed } \varepsilon > 0.$$

where $\left( [1,1,1,1]^\top \right)^\perp$ represents the orthogonal space to $[1,1,1,1]^\top$. At stage $h = 1$, it is defined as

$$\mathcal{F}_1 = \left\{ [x, x]^\top + \varepsilon \frac{\boldsymbol{v}}{\|\boldsymbol{v}\|_2} \, \forall \boldsymbol{v} \in \left( [1,1]^\top \right)^\perp \forall x \in \mathbb{R} \right\}, \text{ for a fixed } \varepsilon > 0.$$

The function class $\mathcal{F}_1$ is shown in Figure 3(a) [9]. The figure shows that almost all 2 armed bandit instances are robust.

**Condition for Robustness**   **Stage Two** At stage $h = 2$, there are four possible state-action pairs, $(s_2, a_1), (s_2, a_2), (s_3, a_1)$ and $(s_3, a_2)$. Thus at stage $h = 2$, the reward vector $\boldsymbol{r_2} = [r_{21}, r_{22}, r_{31}, r_{32}]^\top$ is an element of $\mathbb{R}^4$. Thus the value function $\boldsymbol{f_2}$ is such that

$$\boldsymbol{f_2} = \arg\min_{\boldsymbol{f} \in \mathcal{F}_2} \|\boldsymbol{f} - \boldsymbol{r_2}\|_2$$

The function approximated $Q$ values functions for the states can be read off as

$$\begin{bmatrix} f_2(s_2, a_1) \\ f_2(s_2, a_2) \end{bmatrix} = \begin{bmatrix} x_0 \\ x_0 \end{bmatrix} + \frac{\varepsilon}{\|\boldsymbol{r_2} - \boldsymbol{x_0}\|} \begin{bmatrix} r_{21} - x_0 \\ r_{22} - x_0 \end{bmatrix} \text{ and } \begin{bmatrix} f_2(s_3, a_1) \\ f_2(s_3, a_2) \end{bmatrix} = \begin{bmatrix} x_0 \\ x_0 \end{bmatrix} + \frac{\varepsilon}{\|\boldsymbol{r_2} - \boldsymbol{x_0}\|} \begin{bmatrix} r_{31} - x_0 \\ r_{32} - x_0 \end{bmatrix},$$

where $x_0 = (r_{21} + r_{22} + r_{31} + r_{32})/4$ and $\boldsymbol{x_0} = [x_0, x_0, x_0, x_0]^\top$. Note that one can have a potential huge misspecification error of $l_2$ norm approximately $\|\boldsymbol{r_2} - \boldsymbol{x_0}\|$. Observe that since $r_{21} > r_{22}$ and $r_{31} > r_{32}$ we have, $f_2(s_2, a_1) > f_2(s_2, a_2)$ and $f_2(s_3, a_1) > f_2(s_3, a_2)$, and thus $\arg\max_a f_2(s_2, a) = \pi^*(s_2)$ and $\arg\max_a f_2(s_3, a) = \pi^*(s_3)$.

**Stage One**   At stage $h = 1$ we use the function class $\mathcal{F}_1$. The value function $\boldsymbol{f_1}$ is such that

$$\boldsymbol{f_1} = \arg\min_{\boldsymbol{f} \in \mathcal{F}_1} \|\boldsymbol{f} - \boldsymbol{r_1}\|_2$$

where $\boldsymbol{r_1} = \begin{bmatrix} r_{11} + \max_a f_2(s_2, a) \\ r_{12} + \max_a f_2(s_3, a) \end{bmatrix} = \begin{bmatrix} r_{11} + x_0 + \frac{\varepsilon}{\|\boldsymbol{r_2} - \boldsymbol{x_0}\|}(r_{21} - x_0) \\ r_{12} + x_0 + \frac{\varepsilon}{\|\boldsymbol{r_2} - \boldsymbol{x_0}\|}(r_{31} - x_0). \end{bmatrix}$.

---

[9]We remove the dependency on $\varepsilon$ from the text to reduce burden of notation

Since $\mathcal{F}_1$ is such that almost all bandit instances are robust, to ensure $\mathrm{argmax}_a f_1(s_1, a) = \pi^*(s_1)$, we must have $\boldsymbol{r_1} \in \mathcal{G}_1$. This is true if $r_{11} < r_{12} + \frac{\varepsilon}{\|\boldsymbol{r_2} - \boldsymbol{x_0}\|}(r_{31} - r_{21})$. Thus, under the uniform behavioral policy, the given MDP $\mathcal{M}$, along with the function class $\mathcal{F}_1, \mathcal{F}_2$, is robust if the preceding condition is satisfied.

## 6 Discussion

In this work, we present a systematic study of model misspecified instances in the bandit, contextual bandit, and Markov-decision-process settings compliant with practical algorithms. Previous theoretical works have all indicated that these algorithms are insufficient to learn the optimal policy without the realizability assumption, despite their panacean use and often impressive performance. In this regard, we hope to have provided some insight into explaining why such learning systems perform as well as they do. However, we realize several limitations to our study, the primary of which is that we are uncertain how one can better utilize the concept of robust regions to construct better feature representations or design better algorithms. We can also not explain how these results might extend to settings with an exponential number of states and actions. But, we hope to have convinced the reader that there is a natural existence of benign instances in the model misspecified setting. We hope future research can be devoted to better explainability of function-approximated learning algorithms.

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

---

**Algorithm 2** $\varepsilon$-greedy algorithm

---

1: **for** t = 1 to T **do**
2:    With an estimate $\widehat{\theta}_t$, play arm $i$ such that

$$A_t = \underset{i \in [K]}{\operatorname{argmax}} \varphi_i^\top \widehat{\theta}_t \, \text{w.p.} \, 1 - \varepsilon_t$$

$$= play \ uniformly \ over \ K \ arms \, \text{w.p.} \, \varepsilon_t$$

3:    Observe the reward $Y_t$.
4:    Update the estimate as

$$\widehat{\theta}_{t+1} = \underset{\theta}{\operatorname{argmin}} \sum_{s=1}^{t} [\varphi_{A_s}^\top \theta - Y_s]^2$$

5: **end for**

---

Csaba Szepesvári. *Algorithms for reinforcement learning*. Springer Nature, 2022.

Benjamin Van Roy and Zheng Wen. Generalization and exploration via randomized value functions. *stat*, 1050:4, 2014.

Andrea Zanette, David Brandfonbrener, Emma Brunskill, Matteo Pirotta, and Alessandro Lazaric. Frequentist regret bounds for randomized least-squares value iteration. In Silvia Chiappa and Roberto Calandra (eds.), *Proceedings of the Twenty Third International Conference on Artificial Intelligence and Statistics*, volume 108 of *Proceedings of Machine Learning Research*, pp. 1954–1964. PMLR, 26–28 Aug 2020a. URL https://proceedings.mlr.press/v108/zanette20a.html.

Andrea Zanette, Alessandro Lazaric, Mykel Kochenderfer, and Emma Brunskill. Learning near optimal policies with low inherent bellman error. In *International Conference on Machine Learning*, pp. 10978–10989. PMLR, 2020b.

Weitong Zhang, Jiafan He, Zhiyuan Fan, and Quanquan Gu. On the interplay between misspecification and sub-optimality gap in linear contextual bandits. In Andreas Krause, Emma Brunskill, Kyunghyun Cho, Barbara Engelhardt, Sivan Sabato, and Jonathan Scarlett (eds.), *Proceedings of the 40th International Conference on Machine Learning*, volume 202 of *Proceedings of Machine Learning Research*, pp. 41111–41132. PMLR, 23–29 Jul 2023. URL https://proceedings.mlr.press/v202/zhang23n.html.

Shaofeng Zou, Tengyu Xu, and Yingbin Liang. Finite-sample analysis for sarsa with linear function approximation. *Advances in neural information processing systems*, 32, 2019.

# A  Description of Algorithms and Proofs of Theorems in Bandits

## A.1  $\varepsilon$-Greedy Algorithm

$\varepsilon$-greedy algorithm is a popular forced-exploration-based algorithm widely used in practice. For completeness, we describe it in Algorithm 2. We give a detailed description and proof of the result presented in Theorem 3.16.

**Theorem 3.16.** *Given a feature matrix* $\mathbf{\Phi}$ *and a fixed bandit instance* $\boldsymbol{\mu}$ *satisfying Assumptions 3.13 and 3.14, the $\varepsilon$-greedy algorithm with $\varepsilon_t$ being varied as $1/\sqrt{t}$ over episodes, the cumulative regret, asymptotically enjoy sub-linear regret, that is* $\lim_{T \to \infty} \frac{Regret}{\sqrt{T}} \leqslant \Delta_{\max}$, *where* $\Delta_{\max}$ *is the maximum sub-optimality gap,* $\Delta_{\max} = \max_{i \in [K]} \mu_k - \mu_i$ *and regret is the expected cumulative regret,* $\sum_{t=1}^{T} \mathbb{E}[\mu_k - \mu_{A_t}]$.

*Remark* A.1.  The proof uses the same technique as presented in Auer et al. (2002). The critical observation is that the least squares estimate $\widehat{\theta}_t$, in our notation $\mathbf{P}^{\boldsymbol{\Lambda}}(\widehat{\boldsymbol{\mu}_t})$, is guaranteed to generate optimal play under the greedy strategy if the sample mean estimate $\widehat{\boldsymbol{\mu}_t}$ falls inside the robust region $\mathcal{R}_k$. The concentration of sub-Gaussian random variables ensures that given enough samples $\widehat{\boldsymbol{\mu}_t}$ will fall within a $\delta$-neighbourhood of the true rewards $\boldsymbol{\mu}$. The fact that $\boldsymbol{\mu}$ is an interior point of $\mathcal{R}_k$, ensures a that there exists $\delta$- neighbourhood about $\boldsymbol{\mu}$ that is contained in the robust region $\mathcal{R}_k$.

*Proof.* Since $\boldsymbol{\mu}$ is an interior point of the robust region $\mathcal{R}_k$, by Assumption 3.14, there exists a $\delta$-open-rectangle centred at $\boldsymbol{\mu}$, denoted as $\mathbf{1}_\delta(\boldsymbol{\mu})$ which is contained in the robust observation region $\mathcal{R}_k$. We observe that the probability of choosing a sub-optimal arm $i$ at $t^{th}$ round is either due to a random play, which happens with probability $\frac{\varepsilon_t}{K}$ or during the greedy play when the reward estimate $\widehat{\boldsymbol{\mu}}_t$ does not belong to $\mathbf{1}_\delta(\boldsymbol{\mu})$, the $\delta$ open-rectangle of $\boldsymbol{\mu}$. That is,

$$\mathbb{P}[A_t = i] \leqslant \frac{\varepsilon_t}{K} + (1 - \frac{\varepsilon_t}{K})\mathbb{P}[\widehat{\boldsymbol{\mu}}_t \notin \mathbf{1}_\delta(\boldsymbol{\mu})].$$

Now $\mathbb{P}[\widehat{\boldsymbol{\mu}}_t \notin \mathbf{1}_\delta(\boldsymbol{\mu})]$, from the definition of the $\delta$-open-rectangle, can be upper bounded by taking a union bound over all arms to give

$$\mathbb{P}[\widehat{\boldsymbol{\mu}}_t \notin \mathbf{1}_\delta(\boldsymbol{\mu})] \leqslant \sum_{i=1}^{K}\mathbb{P}\left[|\widehat{\mu}_i^{n_{i,t}} - \mu_i| \geqslant \delta\right],$$

where $\widehat{\mu}_i^{n_{i,t}}$ is the sample estimate of arm $i$, having played $n_{i,t}$ times till time $t$. The remainder of the proof has the same flavor as in the work of Auer et al. (2002). We provide here for the sake of completeness. Note that each term $\mathbb{P}\left[|\widehat{\mu}_i^{n_{i,t}} - \mu_i| \geqslant \delta\right]$ can be bounded in the following manner,

$$\mathbb{P}\left\{|\widehat{\mu}_i^{n_{i,t}} - \mu_i| \geqslant \delta\right\} = \sum_{s=1}^{t}\mathbb{P}\left\{|\widehat{\mu}_i^s - \mu_i| \geqslant \delta, n_{i,t} = s\right\}$$

$$= \sum_{s=1}^{t}\mathbb{P}\left\{n_{i,t} = s \,|\, |\widehat{\mu}_i^s - \mu_i| \geqslant \delta\right\}\mathbb{P}\left\{|\widehat{\mu}_i^s - \mu_i| \geqslant \delta\right\}$$

$$\leqslant \sum_{s=1}^{t}\mathbb{P}\left\{n_{i,t} = s \,|\, |\widehat{\mu}_i^s - \mu_i| \geqslant \delta\right\}2\exp(-2s\delta^2),$$

where we use the sub-Gaussian concentration by Assumption 3.13 and Lemma I.5,

$$\leqslant \sum_{s=1}^{t_0}\mathbb{P}\left\{n_{i,t} = s \,|\, |\widehat{\mu}_i^s - \mu_i| \geqslant \delta\right\} + \sum_{s=t_0+1}^{t}2\exp(-2s\delta^2)$$

$$\leqslant \sum_{s=1}^{t_0}\mathbb{P}\left\{n_{i,t} = s \,|\, |\widehat{\mu}_i^s - \mu_i| \geqslant \delta\right\} + \frac{1}{\delta^2}\exp(-2t_0\delta^2),$$

where we use the identity $\sum_{t=x+1}^{\infty}\exp(-\kappa t) \leqslant \frac{1}{\kappa}\exp(-\kappa x)$. Let $n_{i,t}^R$ be the number of times arm $i$ has been played randomly till time $t$, then,

$$\sum_{s=1}^{t_0}\mathbb{P}\left\{n_{i,t} = s \,|\, |\widehat{\mu}_i^s - \mu_i| \geqslant \delta\right\}$$

$$\leqslant \sum_{s=1}^{t_0}\mathbb{P}\left\{n_{i,t}^R \leqslant s \,|\, |\widehat{\mu}_i^s - \mu_i| \geqslant \delta\right\}$$

$$\leqslant \sum_{s=1}^{t_0}\mathbb{P}\{n_{i,t}^R \leqslant s\}$$

$$\leqslant t_0\mathbb{P}\{n_{i,t}^R \leqslant t_0\}.$$

Now

$$\mathbb{E}\left[n_{i,t}^R\right] = \frac{1}{K}\sum_{s=1}^{t}\varepsilon_s$$

and variance of $n_{i,t}^R$ is

$$\mathbb{V}[n_{i,t}^R] = \sum_{s=1}^{t}\frac{\varepsilon_s}{K}\left(1 - \frac{\varepsilon_s}{K}\right) \leqslant \sum_{s=1}^{t}\frac{\varepsilon_s}{K}.$$

Thus, choosing $t_0 = \frac{1}{2K}\sum_{s=1}^{t}\varepsilon_s$, we have

$$\mathbb{P}\{n_{i,t}^R \leqslant t_0\} \leqslant \exp(-t_0/5),$$

from Bernstein's Inequality, Lemma I.6. Thus, putting it all together, we have,

$$\mathbb{P}[A_t = i] \leqslant \frac{\varepsilon_t}{K} + Kt_0\exp(-t_0/5) + \frac{K}{\delta^2}\exp(-2t_0\delta^2).$$

Thus, to complete the proof, we only need to find a lower bound on $t_0$. As per our definition,

$$t_0 = \frac{1}{2K}\sum_{s=1}^{t}\varepsilon_s.$$

Plugging the value of $\varepsilon_s = 1/\sqrt{s}$, we get

$$t_0 = \frac{1}{2K}\sum_{s=1}^{t}\frac{1}{\sqrt{s}}$$
$$\geqslant \frac{1}{K}(\sqrt{t+1}-1).$$

Thus for $t \geqslant (1+K)^2 - 1$,

$$\mathbb{P}[A_t = i] \leqslant \frac{1}{K\sqrt{t}} + (\sqrt{t+1}-1)\exp\left(-\frac{\sqrt{t+1}-1}{5K}\right) + \frac{K}{\delta^2}\exp\left(-\frac{2\delta^2(\sqrt{t+1}-1)}{K}\right).$$

Therefore, regret at any time $t \geqslant (1+K)^2 - 1$ is

$$\text{Regret}_t = \sum_{i=1}^{K}\Delta_i\mathbb{P}[A_t = i] \leqslant \frac{\Delta_{\max}}{\sqrt{t}} + o(1/t^\alpha), \text{ for any } \alpha > 1$$

where $\Delta_i$ is the sub-optimality gap $\mu_k - \mu_i$ and $\Delta_{\max} = \max_{i\in[K]}\Delta_i$. The asymptotic behavior of the cumulative regret is

$$\lim_{T\to\infty}\frac{\sum_{t=1}^{T}\text{Regret}_t}{\sqrt{T}} \leqslant \Delta_{\max}$$

. $\qquad\qquad\qquad\qquad\qquad\qquad\qquad\qquad\qquad\qquad\qquad\qquad\qquad\qquad\qquad\qquad\qquad\square$

## A.2   Interior Point

The reason that we demand explicitly for $\boldsymbol{\mu}$ to be an interior point of $\mathcal{R}_k$ is because $\mathcal{R}_k$ is not necessarily an open set, as shown in Proposition A.2.

**Proposition A.2.** *The* robust observation region $\mathcal{R}_k$ *is a $G_\delta$ set that is a countable intersection of open sets.*

*Proof.* Note that the for any arbitrary but fixed sampling distribution $\{\lambda_i\}_{i=1}^{K}$ belonging in the $K$ dimensional simplex $\Delta_K$, we have

$$\left\{\boldsymbol{\mu}\in\mathcal{G}_k : \mathbf{P}^{\boldsymbol{\Lambda}}(\boldsymbol{\mu})\in\boldsymbol{\Theta}_k\right\},$$

is an open set as the set $\boldsymbol{\Theta}_k$ is an open set, and the projection operator is continuous. Now, since any sampling distribution $\{\lambda_i\}_{i=1}^{K}$ are rationals, that is $\{\lambda_i\}_{i=1}^{K}\in\mathbb{Q}^K$, we have

$$\mathcal{R}_k \triangleq \bigcap_{\{\lambda_i\}_{i=1}^{K}\in\Delta_K}\left\{\mu\in\mathcal{G}_k : \mathbf{P}^{\boldsymbol{\Lambda}}(\boldsymbol{\mu})\in\boldsymbol{\Theta}_k\right\},$$

is a $G_\delta$ set. $\qquad\qquad\qquad\qquad\qquad\qquad\qquad\qquad\qquad\qquad\qquad\qquad\qquad\qquad\qquad\qquad\qquad\square$

---

**Algorithm 3** Generic $\varepsilon$-greedy algorithm
___
1: **for** t = 1 to T **do**
2:     Observe context $X_t$ at time $t$
3:     With an estimate $\widehat{\theta}_t$, play arm $A_t$ such that

$$A_t = \underset{a \in \mathcal{A}}{\operatorname{argmax}} \varphi(X_t, a)^\top \widehat{\theta}_t \text{ w.p. } 1 - \varepsilon_t$$
$$= \textit{play uniformly over } \mathcal{A} \textit{ arms w.p. } \varepsilon_t$$

4:     Observe the reward $Y_t$.
5:     Update the estimate as

$$\widehat{\theta}_{t+1} = \underset{\theta}{\operatorname{argmin}} \sum_{s=1}^{t} [\varphi(X_s, A_s)^\top \theta - Y_s]^2.$$

6: **end for**
___

# B  Description of Algorithms and Proofs of Theorems in Contextual Bandits

## B.1  $\varepsilon$-Greedy Algorithm

The $\varepsilon$-greedy algorithm in the contextual setup is described in Algorithm 3 for completeness. We now give a detailed description and proof of the result presented in Theorem 4.10.

**Theorem 4.10.** *Given feature matrix $\mathbf{\Phi}$ and a* fixed *contextual bandit instance $\boldsymbol{\mu}$ satisfying Assumptions 4.7, 4.8 and 4.9, the $\varepsilon$-greedy algorithm with $\varepsilon_t$ varied as $1/\sqrt{t}$, the cumulative regret, asymptotically enjoys sub-linear regret. That is, $\lim_{T \to \infty} \frac{Regret}{\sqrt{T}} \leqslant \Delta_{\max}$, where $\Delta_{\max}$ is the maximum sub-optimality gap between context-arm rewards, $\Delta_{\max} = \max_{x \in \mathcal{X}} \max_{a \in \mathcal{A}} \mu_{x, \mathrm{OPT}(x)} - \mu_{x, a}$.*

*Remark* B.1. The proof uses the same technique for the bandits section. The key observation is that the least squares estimate $\widehat{\theta}_t$, in our notation $\mathbf{P^\Lambda}(\widehat{\boldsymbol{\mu}_t})$, is guaranteed to generate optimal play under the greedy strategy if the sample mean estimate $\widehat{\boldsymbol{\mu}_t}$ falls inside the robust region $\mathcal{R}^{\mathcal{X}}$. The concentration of sub-Gaussian random variables ensures that given enough samples $\widehat{\boldsymbol{\mu}_t}$ will fall within a $\delta$-neighbourhood of the true rewards $\boldsymbol{\mu}$. The fact that $\boldsymbol{\mu}$ is an interior point of $\mathcal{R}^{\mathcal{X}}$, ensures a that there exists $\delta$- neighbourhood about $\boldsymbol{\mu}$ that is contained in the robust region $\mathcal{R}^{\mathcal{X}}$.

*Proof.* Since $\boldsymbol{\mu}$ is an interior point of the robust region $\mathcal{R}^{\mathcal{X}}$, by Assumption 4.9, there exists a $\delta$-open-rectangle centred at $\boldsymbol{\mu}$, defined as $\mathbf{1}_\delta(\boldsymbol{\mu})$ which is contained in the robust observation region $\mathcal{R}^{\mathcal{X}}$. The probability of choosing a sub-optimal arm $a$ in state $x$ at $t^{th}$ round is either due to a random play, which happens with probability $\frac{\varepsilon_t}{\mathrm{A}}$ or, during the greedy play, the reward estimate $\widehat{\boldsymbol{\mu}_t} \in \mathbb{R}^{\mathrm{XA}}$ does not belong to the $\delta$ open-rectangle of $\boldsymbol{\mu}$, $\mathbf{1}_\delta(\boldsymbol{\mu})$. That is,

$$\mathbb{P}[A_t = a \mid X_t = x] \leqslant \frac{\varepsilon_t}{\mathrm{A}} + (1 - \frac{\varepsilon_t}{\mathrm{A}})\mathbb{P}[\widehat{\boldsymbol{\mu}_t} \notin \mathbf{1}_\delta(\boldsymbol{\mu})].$$

Now $\mathbb{P}[\widehat{\boldsymbol{\mu}_t} \notin \mathbf{1}_\delta(\boldsymbol{\mu})]$, from the definition of the $\delta$-open-rectangle, can be upper bounded by taking a union bound over all arms and contexts to give

$$\mathbb{P}[\widehat{\boldsymbol{\mu}_t} \notin \mathbf{1}_\delta(\boldsymbol{\mu})] \leqslant \sum_{a \in \mathcal{A}} \sum_{x \in \mathcal{X}} \mathbb{P}\left[|\widehat{\mu}_{x,a}^{n_{x,a,t}} - \mu_{x,a}| \geqslant \delta\right],$$

where $\widehat{\mu}_{x,a}^{n_{x,a,t}}$ is the sample estimate of the state-action pair $(x,a)$, having been sampled $n_{x,a,t}$ times till time $t$. The remainder of the proof has the same flavor as in the work of Auer et al. (2002). We provide here for the sake of completeness. Note

that each term $\mathbb{P}\left[||\widehat{\mu}_{x,a}^{n_{x,a,t}} - \mu_{x,a}|| \geqslant \delta\right]$ can be bounded in the following manner,

$$\mathbb{P}\left\{|\widehat{\mu}_{x,a}^{n_{x,a,t}} - \mu_{x,a}| \geqslant \delta\right\} = \sum_{s=1}^{t} \mathbb{P}\left\{|\widehat{\mu}_{x,a}^{s} - \mu_{x,a}| \geqslant \delta, n_{x,a,t} = s\right\}$$

$$= \sum_{s=1}^{t} \mathbb{P}\left\{n_{x,a,t} = s \,|\, |\widehat{\mu}_{x,a}^{s} - \mu_{x,a}| \geqslant \delta\right\} \mathbb{P}\left\{|\widehat{\mu}_{x,a}^{s} - \mu_{x,a}| \geqslant \delta\right\}$$

$$\leqslant \sum_{s=1}^{t} \mathbb{P}\left\{n_{x,a,t} = s \,|\, |\widehat{\mu}_{x,a}^{s} - \mu_{x,a}| \geqslant \delta\right\} 2\exp(-2s\delta^2),$$

where we use the sub-Gaussian concentration by Assumption 4.7 and Lemma I.5,

$$\leqslant \sum_{s=1}^{t_0} \mathbb{P}\left\{n_{x,a,t} = s \,|\, |\widehat{\mu}_{x,a}^{s} - \mu_{x,a}| \geqslant \delta\right\} + \sum_{s=t_0+1}^{t} 2\exp(-2s\delta^2)$$

$$\leqslant \sum_{s=1}^{t_0} \mathbb{P}\left\{n_{x,a,t} = s \,|\, |\widehat{\mu}_{x,a}^{s} - \mu_{x,a}| \geqslant \delta\right\} + \frac{1}{\delta^2}\exp(-2t_0\delta^2),$$

where we use the identity $\sum_{t=x+1}^{\infty} \exp(-\kappa t) \leqslant \frac{1}{\kappa}\exp(-\kappa x)$. Let $n_{x,a,t}^{R}$ be the number of times arm $a$ has been played randomly at state $x$ till time $t$, then,

$$\sum_{s=1}^{t_0} \mathbb{P}\left\{n_{x,a,t} = s \,|\, |\widehat{\mu}_{x,a}^{s} - \mu_{x,a}| \geqslant \delta\right\}$$

$$\leqslant \sum_{s=1}^{t_0} \mathbb{P}\left\{n_{x,a,t}^{R} \leqslant s \,|\, |\widehat{\mu}_{i}^{s} - \mu_{i}| \geqslant \delta\right\}$$

$$\leqslant \sum_{s=1}^{t_0} \mathbb{P}\{n_{x,a,t}^{R} \leqslant s\}$$

$$\leqslant t_0 \mathbb{P}\{n_{x,a,t}^{R} \leqslant t_0\}.$$

Now, since states $x \in \mathcal{X}$ is chosen statistically independently with probability $\mathbf{p}_x$ over the state-space (Assumption 4.8) and during random play actions are chosen independently and uniformly over action space $\mathcal{A}$, we have,

$$\mathbb{E}\left[n_{x,a,t}^{R}\right] = \frac{\mathbf{p}_x}{\mathrm{A}}\sum_{s=1}^{t}\varepsilon_s$$

and variance of $n_{x,a,t}^{R}$ is

$$\mathbb{V}[n_{x,a,t}^{R}] = \sum_{s=1}^{t}\frac{\mathbf{p}_x\varepsilon_s}{\mathrm{A}}\left(1 - \frac{\mathbf{p}_x\varepsilon_s}{\mathrm{A}}\right) \leqslant \frac{\mathbf{p}_x}{\mathrm{A}}\sum_{s=1}^{t}\varepsilon_s.$$

Thus, choosing $t_0 = \frac{\mathbf{p}_x}{2\mathrm{A}}\sum_{s=1}^{t}\varepsilon_s$, we have

$$\mathbb{P}\{n_{x,a,t}^{R} \leqslant t_0\} \leqslant \exp(-t_0/5),$$

from Bernstein's Inequality, Lemma I.6. Thus, putting it all together, we have,

$$\mathbb{P}[A_t = a \,|\, X_t = x] \leqslant \frac{\varepsilon_t}{\mathrm{A}} + \mathrm{XA}t_0\exp(-t_0/5) + \frac{\mathrm{XA}}{\delta^2}\exp(-2t_0\delta^2).$$

Thus, to complete the proof, we only need to find a lower bound on $t_0$. As per our definition,

$$t_0 = \frac{\mathbf{p}_x}{2\mathrm{A}}\sum_{s=1}^{t}\varepsilon_s$$

Plugging the value of $\varepsilon_s = 1/\sqrt{s}$, we get

$$t_0 = \frac{\mathbf{p}_x}{2A} \sum_{s=1}^{t} \frac{1}{\sqrt{s}}$$

$$\geqslant \frac{\mathbf{p}_x}{A}(\sqrt{t+1} - 1).$$

Thus for $t \geqslant (1 + A/p_x)^2 - 1$ (Note that by Assumption 4.8, we have $p_x > 0$ for any context $x$),

$$\mathbb{P}[A_t = a \mid X_t = x] \leqslant \frac{1}{A\sqrt{t}} + \mathbf{p}_x X(\sqrt{t+1} - 1)\exp\left(-\frac{\mathbf{p}_x(\sqrt{t+1} - 1)}{5A}\right) + \frac{XA}{\delta^2}\exp\left(-\frac{2\delta^2\mathbf{p}_x(\sqrt{t+1} - 1)}{A}\right).$$

Thus, the expected regret at time $t \geqslant (1 + A/p_x)^2 - 1$ at state $x$ is

$$\mathrm{Regret}_{x,t} = \sum_{a \in \mathcal{A}} \Delta_{x,a} \mathbb{P}[A_t = a \mid X_t = x]$$

$$\leqslant \Delta_x \frac{1}{\sqrt{t}} + o(1/t^\alpha) \text{ for any } \alpha > 1,$$

where $\Delta_x = \max_{a \in \mathcal{A}} \Delta_{x,a}$ and $\Delta_{x,a} = \mu_{x,\mathrm{OPT}(x)} - \mu_{x,a}$. Thus, the expected regret at time $t \geqslant (1 + A/p_x)^2 - 1$ is

$$\mathrm{Regret}_t = \sum_{x \in \mathcal{X}} \mathrm{Regret}_{x,t} \mathbb{P}[X_t = x] \leqslant \sum_{x \in \mathcal{X}} \mathbf{p}_x\left(\Delta_x \frac{1}{\sqrt{t}} + o(1/t^\alpha)\right)$$

$$\leqslant \frac{\Delta_{\max}}{\sqrt{t}} + o(1/t^\alpha),$$

where $\Delta_{\max} = \max_{x \in \mathcal{X}} \Delta_x$. The asymptotic behavior of the cumulative regret is, therefore,

$$\lim_{T \to \infty} \frac{\sum_{t=1}^{T} \mathrm{Regret}_t}{\sqrt{T}} \leqslant \Delta_{\max}.$$

$\square$

## B.2 Corollary of Theorem 4.5

As a corollary of Theorem 4.5 we can show that the model estimate $\mathbf{P^\Lambda}(\boldsymbol{\mu})$ computed under any sampling distribution $\{\lambda_{x,a}\}_{(x,a) \in \mathcal{X} \times \mathcal{A}}$ must belong to the robust parameter region $\boldsymbol{\Theta}_{\mathrm{OPT}(x)}^x$ for every context $x$.

**Corollary B.2.** *For any contextual bandit instance $\boldsymbol{\mu}$, we have $\boldsymbol{\mu} \in \mathcal{R}^\mathcal{X}$ if and only if the model estimate $\mathbf{P^\Lambda}(\boldsymbol{\mu})$, computed under any sampling distribution $\{\lambda_{x,a}\}_{(x,a) \in \mathcal{X} \times \mathcal{A}} \in \Delta_{\mathrm{XA}}$, belongs to $\boldsymbol{\Theta}^\mathcal{X}$, where $\boldsymbol{\Theta}^\mathcal{X} \triangleq \bigcap_{x \in \mathcal{X}} \boldsymbol{\Theta}_{\mathrm{OPT}(x)}^x$.*

*Proof.* For any $\boldsymbol{\mu}$ we have $\boldsymbol{\mu}_x \in \mathcal{G}_{\mathrm{OPT}(x)}^x$ for every context $x$, by definition. Thus, from definition of $\mathcal{R}^\mathcal{X} = \bigcap_{x \in \mathcal{X}} \mathcal{R}_{\mathrm{OPT}(x)}^x$ we have,

$$\boldsymbol{\mu} \in \mathcal{R}^\mathcal{X} \iff \boldsymbol{\mu}_x \in \mathcal{R}_{\mathrm{OPT}(x)}^x \, \forall x \in \mathcal{X}.$$

$$\iff \mathbf{P^\Lambda}(\boldsymbol{\mu}) \in \boldsymbol{\Theta}_{\mathrm{OPT}(x)}^x \, \forall x \in \mathcal{X}.$$

for any sampling distribution $\{\lambda_{x,a}\}_{(x,a) \in \mathcal{X} \times \mathcal{A}} \in \Delta_{\mathcal{X}\mathcal{A}}$. This follows from definition of $\mathcal{R}_{\mathrm{OPT}(x)}^x$. But this is the definition of set intersection, that is,

$$\iff \mathbf{P^\Lambda}(\boldsymbol{\mu}) \in \bigcap_{x \in \mathcal{X}} \boldsymbol{\Theta}_{\mathrm{OPT}(x)}^x$$

for any sampling distribution $\{\lambda_{x,a}\}_{(x,a) \in \mathcal{X} \times \mathcal{A}} \in \Delta_{\mathcal{X}\mathcal{A}}$. $\square$

### B.3 Characterization of the Robust Observation Region in Contextual Settings

When the model class is chosen to be linear, the *robust observation region*, $\mathcal{R}^{\mathcal{X}}$, has an explicit analytic description which was presented without proof in Theorem 4.6. We provide the proof here.

**Theorem 4.6.** *Define* $\Theta^{\mathcal{X}} \triangleq \bigcap_{x \in \mathcal{X}} \Theta^x_{\mathrm{OPT}(x)}$. *A contextual bandit instance* $\boldsymbol{\mu}$ *belongs to the* robust observation region $\mathcal{R}^{\mathcal{X}}$ *if and only if for each* $d \times d$ *full rank sub-matrix of* $\boldsymbol{\Phi}$, *denoted by* $\Phi_d$, *along with the corresponding* $d$ *rows of* $\boldsymbol{\mu}$, *denoted as* $\boldsymbol{\mu}_d$, *satisfy* $\Phi_d^{-1} \boldsymbol{\mu}_d \in \Theta^{\mathcal{X}}$. *That is* $\boldsymbol{\mu} \in \mathcal{R}^{\mathcal{X}}$ *if and only if* $\Phi_d^{-1} \boldsymbol{\mu}_d \in \Theta^{\mathcal{X}}$ *for every* $d \times d$ *full rank sub-matrices of* $\boldsymbol{\Phi}$ *and the corresponding* $d$ *rows of* $\boldsymbol{\mu}$.

*Proof.* The proof uses a result of Forsgren (1996), presented in Lemma I.4, that for any any sampling distribution $\{\lambda_{x,a}\}_{(x,a) \in \mathcal{X} \times \mathcal{A}} \in \Delta_{\mathrm{XA}}$, the model estimate $\mathbf{P}^{\boldsymbol{\Lambda}}(\boldsymbol{\mu})$, with $\boldsymbol{\Lambda} = \mathrm{diag}(\{\lambda_{x,a}\}_{(x,a) \in \mathcal{X} \times \mathcal{A}})$, lies in the convex hull of the *basic solutions* $\Phi_d^{-1} \boldsymbol{\mu}_d$. Note that for any $\boldsymbol{\mu}$, we have $\boldsymbol{\mu}_x \in \mathcal{G}^x_{\mathrm{OPT}(x)}$ for any context $x$ by definition. Therefore, we have

$$\boldsymbol{\mu} \in \mathcal{R}^{\mathcal{X}} \iff \mathbf{P}^{\boldsymbol{\Lambda}}(\boldsymbol{\mu}) \in \Theta^{\mathcal{X}} \, \forall \boldsymbol{\Lambda} \in \mathcal{P}\Delta_{\mathrm{XA}}$$
$$\iff \mathrm{conv}\{\Phi_d^{-1} \boldsymbol{\mu}_d \, \forall \Phi_d \subset \boldsymbol{\Phi}\} \subset \Theta^{\mathcal{X}},$$
$$\iff \Phi_d^{-1} \boldsymbol{\mu}_d \in \Theta^{\mathcal{X}} \, \forall \Phi_d \subset \boldsymbol{\Phi}.$$

The first line above follows from Corollary B.2, and the second line follows from Lemma I.4. (We abuse notation to denote $d \times d$ full rank sub-matrices of $\boldsymbol{\Phi}$ by $\Phi_d \subset \boldsymbol{\Phi}$ and use $\mathrm{conv}$ to denote the convex hull.) The last assertion follows because $\Theta^{\mathcal{X}}$ is a convex set. $\qquad \square$

## C LinUCB

LinUCB remains a canonical regret optimal bandit algorithm. In this section we show that under standard assumptions, we retain the optimal regret of LinUCB even under misspecified settings, given that the bandit instance belongs to the robust region. We begin by first introducing the algorithm in Algorithm 4.

---

**Algorithm 4** OFUL Algorithm

---

1: *Forced Exploration Phase of $d$ linearly independent features*
2: Set $V = \mathbf{0}^{d \times d}$ and $S = \mathbf{0}^d$
3: **for** $i = 1$ to $2d$ **do**
4:     Play feature $\varphi_i$ and observe noisy reward $y_i$
5:     Compute $V = V + \varphi_i \varphi_i^\top$
6:     Compute $S = S + \varphi_i y_i$
7: **end for**
8: *Standard OFUL Phase*
9: Set $V_t = V$ and $S_t = S$
10: **for** t = 1 to T **do**
11:     Estimate $\widehat{\theta}_t = [V_t]^{-1} S_t$
12:     Play arm $A_t$, such that $\varphi_{A_t}, \tilde{\theta}_t = \mathrm{argmax}_{i \in [K], \theta \in \mathcal{R}_t} \varphi_i^\top \theta$, where $\mathcal{R}_t = \left\{ \theta : \left\| \theta - \widehat{\theta}_t \right\|_{V_t} \leqslant \sqrt{\beta_t(\delta)} \right\}$
13:     Observe the reward $y_t$.
14:     Update $V_{t+1} = V_t + \varphi_{A_t} \varphi_{A_t}^\top$
15:     Update $S_{t+1} = S_t + \varphi_{A_t} y_t$
16: **end for**

---

We added a forced exploration of $d$ linearly-independent features to ensure the invertibility of the design matrix. In Appendix E, we remove the forced exploration phase by adding a regularizer to the design matrix. The rest of the assumptions are standard in the analysis of LinUCB (Abbasi-Yadkori et al., 2011).

**Assumption C.1** (Conditionally sub-Gaussian Noise). At any time $t$, the observation $y_t$ corresponding to the arm played $A_t$, is given by

$$y_t = \mu_{A_t} + \eta_t,$$

where $\eta_t$ is conditionally $R$-sub Gaussian, that is,

$$\mathbf{E}[e^{\lambda\eta_t}\,|\,A_{1:t},\eta_{1:t-1}]\leqslant\exp\!\left(\frac{\lambda^2 R^2}{2}\right)\ \forall\lambda\in\mathbb{R}.$$

**Assumption C.2** (Bounded Features)**.** We assume that for any arm $i$ in the arm set $[K]$, the corresponding feature $\varphi_i$ is bounded in the $l^2$ norm by 1, that is,

$$\|\varphi_i\|_2\leqslant 1\ \forall i\in[K].$$

**Assumption C.3** ($d$ rank feature matrix)**.** We assume that the design matrix computed in the forced exploration phase, $V$, has minimum eigen-value $\lambda_{\min}(V)\geqslant 1$.

*Remark* C.4. Note that, since we have the forced exploration phase to be $2d$ times, we have, $\mathrm{trace}(V)\leqslant 2d\|\varphi\|_2^2\leqslant 2d$. We also have $\mathrm{trace}(V)\geqslant d\lambda_{\min}(V)$. This ensures that $\lambda_{\min}(V)\leqslant 2$, thus resolving any contradictions with respect to Assumption C.3.

**Theorem 3.20.** *Given a feature matrix $\boldsymbol{\Phi}$ satisfying Assumptions C.2 and C.3, and any bandit parameter $\boldsymbol{\mu}$ which is an interior point of the* robust observation region, $\mathcal{R}_{\mathrm{OPT}(\boldsymbol{\mu})}$ *and satisfies Assumption C.1, the LinUCB algorithm achieves regret of the order $\tilde{O}(d\sqrt{t})$. That is, with $\beta_t(\delta)$ set as $2R^2\log\!\left(\frac{(1+t/d)^{d/2}}{\delta}\right)$ for any $\delta>0$, we have with probability at least $1-\delta$*

$$\forall t>1,\quad \sum_{s=1}^{t}\mu_{\mathrm{OPT}(\boldsymbol{\mu})}-\mu_{A_s}\leqslant\frac{4\sqrt{t}R\Delta_{\max}}{\Delta_{\min}}\sqrt{\log\!\left(\frac{(1+t/d)^{d/2}}{\delta}\right)}\sqrt{\log(1+t/d)^d},$$

*where $\Delta_{\min}$ is as defined in Lemma 3.18 and $\Delta_{\max}$ is defined as the worst sub-optimal gap, that is, $\Delta_{\max}=\max_{i\in[K]}\mu_{\mathrm{OPT}(\boldsymbol{\mu})}-\mu_i$.*

We define the $t^{\mathrm{th}}$ model estimate, $\mathbf{P}^{\boldsymbol{\Lambda_t}}(\boldsymbol{\mu})$ as the least squares estimate calculated using $\boldsymbol{\mu}$ under the sampling distribution $\{\lambda_i^t\}_{i=1}^K$ of the $K$ arms at time $t$, that is,

$$\mathbf{P}^{\boldsymbol{\Lambda_t}}(\boldsymbol{\mu})=\Big(\sum_{i\in[K]}\lambda_i^t\varphi_i\varphi_i^\top\Big)^{-1}\sum_{i\in[K]}\lambda_i^t\varphi_i\mu_i.$$

The proof of the Theorem depends on the following lemmas.

**Lemma 3.18.** *If $\boldsymbol{\mu}$ is an interior point of the robust observation region, that is, if $\boldsymbol{\mu}\in\mathrm{Int}(\mathcal{R}_{\mathrm{OPT}(\boldsymbol{\mu})})$ then there exists a $\Delta_{\min}>0$, such that for any sampling distribution $\{\lambda_{i(t)}\}_{i\in[K]}\in\mathcal{P}(\Delta_K)$ for time $t\geqslant 1$, we have $\varphi_{\mathrm{OPT}(\boldsymbol{\mu})}^\top\mathbf{P}^{\boldsymbol{\Lambda_t}}(\boldsymbol{\mu})-\varphi_i^\top\mathbf{P}^{\boldsymbol{\Lambda_t}}(\boldsymbol{\mu})\geqslant\Delta_{\min}$ for any suboptimal arm $i$.*

**Lemma C.5** (Upper Bound of Sub-Optimal Plays)**.** *Given a feature matrix $\boldsymbol{\Phi}$ satisfying Assumptions C.2 and C.3, and any bandit parameter $\boldsymbol{\mu}$ which is an interior point of the* robust observation region, $\mathcal{R}_{\mathrm{OPT}(\boldsymbol{\mu})}$ *and satisfies Assumption C.1, the LinUCB algorithm, Algorithm 4, plays sub-optimally at most $\tilde{O}(d\sqrt{t})$. That is, with $\beta_t(\delta)$ set as $2R^2\log\!\left(\frac{(1+t/d)^{d/2}}{\delta}\right)$ for any $\delta>0$, we have with probability at least $1-\delta$*

$$\sum_{s=1}^{t}\mathbb{1}\{A_s\neq\mathrm{OPT}(\boldsymbol{\mu})\}\leqslant\frac{4\sqrt{t}R}{\Delta_{\min}}\sqrt{\log\!\left(\frac{(1+t/d)^{d/2}}{\delta}\right)}\sqrt{\log(1+t/d)^d},$$

*where $\Delta_{\min}$ is as defined in the previous Lemma 3.18.*

With the above two lemmas, the regret bound is straightforward. Formally,

*Proof of Theorem 3.20.* The cumulative regret, defined as,

$$\sum_{s=1}^{t}\mu_{\mathrm{OPT}(\boldsymbol{\mu})}-\mu_{A_s}$$

$$=\sum_{s=1}^{t}\Big(\mu_{\mathrm{OPT}(\boldsymbol{\mu})}-\mu_{A_s}\Big)\mathbb{1}\{A_s\neq\mathrm{OPT}(\boldsymbol{\mu})\}$$

$$\leqslant\Delta_{\max}\sum_{s=1}^{t}\mathbb{1}\{A_s\neq\mathrm{OPT}(\boldsymbol{\mu})\}.$$

The proof is completed using Lemma C.5. $\qquad\square$

*Proof of Lemma 3.18.* We observe that if, $\boldsymbol{\mu} \in \mathrm{Int}\big(\mathcal{R}_{\mathrm{OPT}(\boldsymbol{\mu})}\big)$ then $\mathbf{P}^{\boldsymbol{\Lambda}_t}(\boldsymbol{\mu})$ belongs to the robust parameter region $\boldsymbol{\Theta}_{\mathrm{OPT}(\boldsymbol{\mu})}$ for all $t \geqslant 1$.

Note that, we have from Theorem 3.12, that $\mathbf{P}^{\boldsymbol{\Lambda}_t}(\boldsymbol{\mu})$ belongs to the closed convex hull of all $\binom{K}{d}$ basic solutions. We denote the convex hull by $\mathcal{K}$, and note that $\mathcal{K}$ is contained in $\boldsymbol{\Theta}_{\mathrm{OPT}(\boldsymbol{\mu})}$. Thus for any time $t \geqslant 1$, and for any sub-optimal arm $i$, we have

$$\big(\varphi_{\mathrm{OPT}(\boldsymbol{\mu})} - \varphi_i\big)^\top \mathbf{P}^{\boldsymbol{\Lambda}_t}(\boldsymbol{\mu}) \geqslant \min_{\theta \in \mathcal{K}} \big(\varphi_{\mathrm{OPT}(\boldsymbol{\mu})} - \varphi_i\big)^\top \theta \geqslant \min_{\theta \in \boldsymbol{\Theta}_{\mathrm{OPT}(\boldsymbol{\mu})}} \big(\varphi_{\mathrm{OPT}(\boldsymbol{\mu})} - \varphi_i\big)^\top \theta = \Delta_i > 0.$$

by definition of $\boldsymbol{\Theta}_{\mathrm{OPT}(\boldsymbol{\mu})}$, since we have restricted to instances with unique optimal arms. We take a finite minimum of $\Delta_i$ over $K$ minimums to define $\Delta_{\min} = \min_{i \in K} \Delta_i$. $\qquad\square$

*Remark* C.6. $\Delta_{\min}$ represents the minimum sub-optimality gap of the bandit instance in the model space.

The crux of the proof of Lemma C.5 is similar to the one found in Abbasi-Yadkori et al. (2011).

*Proof of Lemma C.5.* Consider the following quantity,

$$x^\top \mathbf{P}^{\boldsymbol{\Lambda}_t}(\widehat{\boldsymbol{\mu}_t}) - x^\top \mathbf{P}^{\boldsymbol{\Lambda}_t}(\boldsymbol{\mu}),$$

for any vector $x \in \mathbb{R}^d$. Recall the definitions of $\mathbf{P}^{\boldsymbol{\Lambda}_t}(\widehat{\boldsymbol{\mu}_t})$ and $\mathbf{P}^{\boldsymbol{\Lambda}_t}(\boldsymbol{\mu})$, as

$$\mathbf{P}^{\boldsymbol{\Lambda}_t}(\widehat{\boldsymbol{\mu}_t}) = \Big(\sum_{s=1}^t \varphi_{A_s} \varphi_{A_s}^\top\Big)^{-1} \sum_{s=1}^t \varphi_{A_s}(\mu_{A_s} + \eta_s) \triangleq \widehat{\theta}_t$$

$$\mathbf{P}^{\boldsymbol{\Lambda}_t}(\boldsymbol{\mu}) = \Big(\sum_{s=1}^t \varphi_{A_s} \varphi_{A_s}^\top\Big)^{-1} \sum_{s=1}^t \varphi_{A_s} \mu_{A_s},$$

where $\varphi_{A_s}$ is the feature of the arm played at time $s$ and $\mu_{A_s} + \eta_s$ is the observation at time $s$ having played arm $A_s$. This gives

$$x^\top \mathbf{P}^{\boldsymbol{\Lambda}_t}(\widehat{\boldsymbol{\mu}_t}) - x^\top \mathbf{P}^{\boldsymbol{\Lambda}_t}(\boldsymbol{\mu})$$

$$= x^\top \Big(\sum_{s=1}^t \varphi_{A_s} \varphi_{A_s}^\top\Big)^{-1} \sum_{s=1}^t \varphi_{A_s} \eta_s$$

$$\leqslant \|x^\top V_t^{-1}\|_{V_t} \Big\|\sum_{s=1}^t \varphi_{A_s} \eta_s\Big\|_{V_t^{-1}}$$

$$= \|x\|_{V_t^{-1}} \Big\|\sum_{s=1}^t \varphi_{A_s} \eta_s\Big\|_{V_t^{-1}}.$$

We can now use Lemma I.1 to write (after noting that $V = I$, the Identity Matrix in $d$ dimension),

$$x^\top \mathbf{P}^{\boldsymbol{\Lambda}_t}(\widehat{\boldsymbol{\mu}_t}) - x^\top \mathbf{P}^{\boldsymbol{\Lambda}_t}(\boldsymbol{\mu}) \leqslant \|x\|_{V_t^{-1}} R\sqrt{2\log\Big(\frac{\det V_t^{1/2}}{\delta}\Big)}.$$

Thus setting $x = V_t\Big(\mathbf{P}^{\boldsymbol{\Lambda}_t}(\widehat{\boldsymbol{\mu}_t}) - \mathbf{P}^{\boldsymbol{\Lambda}_t}(\boldsymbol{\mu})\Big)$, we have

$$\Big\|\mathbf{P}^{\boldsymbol{\Lambda}_t}(\widehat{\boldsymbol{\mu}_t}) - \mathbf{P}^{\boldsymbol{\Lambda}_t}(\boldsymbol{\mu})\Big\|_{V_t} \leqslant R\sqrt{2\log\Big(\frac{\det V_t^{1/2}}{\delta}\Big)}, \tag{2}$$

with probability at least $1 - \delta$ for all $t \geqslant 1$.

What this means is that under any sampling distribution, the Projection under that sampling distribution ( Definition 3.4) lies within the high confidence ellipsoid centered around the estimated least squares solution.

Now let us consider the regret in the function space at any time $t$ as defined by the sampling distribution $\boldsymbol{\Lambda}_t$, given by

$$\varphi_{\mathrm{OPT}(\boldsymbol{\mu})}^\top \mathbf{P}^{\boldsymbol{\Lambda}_t}(\boldsymbol{\mu}) - \varphi_{A_t}^\top \mathbf{P}^{\boldsymbol{\Lambda}_t}(\boldsymbol{\mu}).$$

The algorithm, at any time chooses $(\varphi_{A_t}, \tilde{\theta}_t)$ as the optimistic estimate, where $\tilde{\theta}_t$ lies in the high confidence ellipsoid defined by Equation 2. To be consistent in our notation, we shall call the optimistic estimate $\tilde{\theta}_t$ as $\mathbf{P}^{\mathbf{\Lambda_t}}(\tilde{\boldsymbol{\mu}}_t)$. This gives,

$$
\begin{aligned}
&\varphi_{\mathrm{OPT}(\boldsymbol{\mu})}^{\top}\mathbf{P}^{\mathbf{\Lambda_t}}(\boldsymbol{\mu}) - \varphi_{A_t}^{\top}\mathbf{P}^{\mathbf{\Lambda_t}}(\boldsymbol{\mu}) \\
&\leqslant \varphi_{A_t}^{\top}\mathbf{P}^{\mathbf{\Lambda_t}}(\tilde{\boldsymbol{\mu}}_t) - \varphi_{A_t}^{\top}\mathbf{P}^{\mathbf{\Lambda_t}}(\boldsymbol{\mu}) \\
&= \varphi_{A_t}^{\top}\mathbf{P}^{\mathbf{\Lambda_t}}(\tilde{\boldsymbol{\mu}}_t) - \varphi_{A_t}^{\top}\mathbf{P}^{\mathbf{\Lambda_t}}(\widehat{\boldsymbol{\mu}}_t) + \varphi_{A_t}^{\top}\mathbf{P}^{\mathbf{\Lambda_t}}(\widehat{\boldsymbol{\mu}}_t) - \varphi_{A_t}^{\top}\mathbf{P}^{\mathbf{\Lambda_t}}(\boldsymbol{\mu}) \\
&= \varphi_{A_t}^{\top}\left(\mathbf{P}^{\mathbf{\Lambda_t}}(\tilde{\boldsymbol{\mu}}_t) - \mathbf{P}^{\mathbf{\Lambda_t}}(\widehat{\boldsymbol{\mu}}_t)\right) + \varphi_{A_t}^{\top}\left(\mathbf{P}^{\mathbf{\Lambda_t}}(\widehat{\boldsymbol{\mu}}_t) - \mathbf{P}^{\mathbf{\Lambda_t}}(\boldsymbol{\mu})\right) \\
&\leqslant \|\varphi_{A_t}\|_{V_t^{-1}}\left\|\mathbf{P}^{\mathbf{\Lambda_t}}(\tilde{\boldsymbol{\mu}}_t) - \mathbf{P}^{\mathbf{\Lambda_t}}(\widehat{\boldsymbol{\mu}}_t)\right\|_{V_t} + \|\varphi_{A_t}\|_{V_t^{-1}}\left\|\mathbf{P}^{\mathbf{\Lambda_t}}(\widehat{\boldsymbol{\mu}}_t) - \mathbf{P}^{\mathbf{\Lambda_t}}(\boldsymbol{\mu})\right\|_{V_t} \\
&\leqslant 2\|\varphi_{A_t}\|_{V_t^{-1}}\sqrt{\beta_t(\delta)},
\end{aligned}
$$

where $\beta_t(\delta) = 2R^2\log\left(\frac{\det V_t^{1/2}}{\delta}\right)$, with probability at least $1-\delta$. Thus the cumulative regret in the function space is

$$
\begin{aligned}
&\sum_{s=1}^{t}\varphi_{\mathrm{OPT}(\boldsymbol{\mu})}^{\top}\mathbf{P}^{\mathbf{\Lambda_s}}(\boldsymbol{\mu}) - \varphi_{A_s}^{\top}\mathbf{P}^{\mathbf{\Lambda_s}}(\boldsymbol{\mu}) \\
&\leqslant \sum_{s=1}^{t}2\|\varphi_{A_s}\|_{V_s^{-1}}\sqrt{\beta_s(\delta)} \\
&= \sum_{s=1}^{t}2\|\varphi_{A_s}\|_{V_s^{-1}}R\sqrt{2\log\left(\frac{\det V_s^{1/2}}{\delta}\right)} \\
&\leqslant 2\sqrt{2}R\sum_{s=1}^{t}\|\varphi_{A_s}\|_{V_s^{-1}}\sqrt{\log\left(\frac{(1+s/d)^{d/2}}{\delta}\right)},
\end{aligned}
$$

where in the last inequality we use Lemma I.2 and our Assumption C.2 on bounded features. Thus continuing, we have,

$$
\begin{aligned}
&2\sqrt{2}R\sum_{s=1}^{t}\|\varphi_{A_s}\|_{V_s^{-1}}\sqrt{\log\left(\frac{(1+t/d)^{d/2}}{\delta}\right)} \\
&\leqslant 2\sqrt{2}R\sqrt{\log\left(\frac{(1+t/d)^{d/2}}{\delta}\right)}\sum_{s=1}^{t}\|\varphi_{A_s}\|_{V_s^{-1}} \\
&\leqslant 2\sqrt{2}R\sqrt{\log\left(\frac{(1+t/d)^{d/2}}{\delta}\right)}\sqrt{t}\sqrt{\sum_{s=1}^{t}\|\varphi_{A_s}\|_{V_s^{-1}}^2}.
\end{aligned}
$$

From our assumption that $\|\varphi_{A_s}\|_2 \leqslant 1$ and from the forced exploration start we have $\lambda_{\min}(V) = 1$, we use Lemma I.3 to get,

$$
\begin{aligned}
&2\sqrt{2}R\sqrt{\log\left(\frac{(1+t/d)^{d/2}}{\delta}\right)}\sqrt{t}\sqrt{\sum_{s=1}^{t}\|\varphi_{A_s}\|_{V_s^{-1}}^2} \\
&4\sqrt{t}R\sqrt{\log\left(\frac{(1+t/d)^{d/2}}{\delta}\right)}\sqrt{\log\det V_t} \\
&\leqslant 4\sqrt{t}R\sqrt{\log\left(\frac{(1+t/d)^{d/2}}{\delta}\right)}\sqrt{\log(1+t/d)^d} \\
&= \tilde{O}(d\sqrt{t}).
\end{aligned}
$$

Thus what we have shown is that the regret in the function space is of order $\tilde{O}(d\sqrt{t})$. Note that from our definition of the minimum regret in the function space, $\Delta_{\min}$ given by Lemma 3.18, we have

$$\sum_{s=1}^{t}\varphi_{\mathrm{OPT}(\boldsymbol{\mu})}^{\top}\mathbf{P}^{\boldsymbol{\Lambda}_s}(\boldsymbol{\mu})-\varphi_{A_s}^{\top}\mathbf{P}^{\boldsymbol{\Lambda}_s}(\boldsymbol{\mu})$$

$$\geqslant\sum_{s=1}^{t}\mathbb{1}\{A_s\neq\mathrm{OPT}(\boldsymbol{\mu})\}\Delta_{\min}.$$

Together, this gives the result. $\qquad\qquad\square$

## D LinUCB in Contextual Bandits

We show that in the contextual setting, LinUCB achieves sub-linear regret for any contextual bandit instance lying in the robust observation region.

---

**Algorithm 5** OFUL Algorithm

---

1: *Forced Exploration Phase of $d$ linearly independent features*
2: Set $V=\mathbf{0}^{d\times d}$ and $S=\mathbf{0}^d$
3: **for** $i=1$ to $2d$ **do**
4:     Observe context $X_i$
5:     Play feature $\varphi_{X_i,A_i}$ and observe noisy reward $Y_i$
6:     Compute $V=V+\varphi_{X_i,A_i}\varphi_{X_i,A_i}^{\top}$
7:     Compute $S=S+\varphi_{X_i,A_i}Y_i$
8: **end for**
9: *Standard OFUL Phase*
10: Set $V_t=V$ and $S_t=S$
11: **for** t = 1 to T **do**
12:     Estimate $\widehat{\theta}_t=\left[V_t\right]^{-1}S_t$
13:     Observe context $X_t$
14:     Play arm $A_t$, such that $\varphi_{A_t},\tilde{\theta}_t=\mathrm{argmax}_{a\in\mathcal{A},\theta\in\mathcal{C}_t}\varphi_{X_t,A}^{\top}\theta$, where $\mathcal{C}_t=\left\{\theta:\left\|\theta-\widehat{\theta}_t\right\|_{V_t}\leqslant\sqrt{\beta_t(\delta)}\right\}$
15:     Observe the reward $Y_t$.
16:     Update $V_{t+1}=V_t+\varphi_{X_t,A_t}\varphi_{X_t,A_t}^{\top}$
17:     Update $S_{t+1}=S_t+\varphi_{A_t}Y_t$
18: **end for**

---

We added a forced exploration phase of $d$ linearly independent features to ensure the invertibility of the design matrix. In Appendix E we remove the forced exploration phase by adding a regularizer. The rest of the assumptions are standard in the analysis of LinUCB (Abbasi-Yadkori et al., 2011).

**Assumption D.1** (Conditionally sub-Gaussian Noise). At any time $t$, the observation $Y_t$ corresponding to the arm played $A_t$ at context $X_t$, is given by

$$Y_t=\mu_{X_t,A_t}+\eta_t,$$

where $\eta_t$ is conditionally $R$-sub Gaussian, that is,

$$\mathbf{E}[e^{\lambda\eta_t}\,|\,A_{1:t},\eta_{1:t-1}]\leqslant\exp\left(\frac{\lambda^2R^2}{2}\right)\ \forall\lambda\in\mathbb{R}.$$

**Assumption D.2** (Bounded Features). We assume that the features $\varphi_{x,a}$ is bounded in the $l^2$ norm by 1, that is,

$$\|\varphi_{x,a}\|_2\leqslant1\ \forall x\in\mathcal{X},a\in\mathcal{A}.$$

**Assumption D.3** ($d$ rank feature matrix). We assume that the design matrix computed in the forced exploration phase, $V$ has minimum eigen value $\lambda_{\min}(V)\geqslant1$.

**Theorem 4.13.** *Under Assumptions D.1, D.2 and D.3, for any contextual bandit instance $\boldsymbol{\mu}$ lying in the robust region $\mathcal{R}^{\mathcal{X}}$ of a given feature matrix $\boldsymbol{\Phi}$, we have for any $t \geqslant 1$, and for any $\delta > 0$, the regret of LinUCB as,*

$$\sum_{s=1}^{t} \mu_{X_s,\mathrm{OPT}(X_s)} - \mu_{X_s,A_S} \leqslant \frac{4\sqrt{t}R\Delta_{\max}}{\Delta_{\min}}\sqrt{\log\left(\frac{(1+t/d)^{d/2}}{\delta}\right)}\sqrt{\log(1+t/d)^d},$$

*with probability at least $1-\delta$. Here $\Delta_{\max} = \max_{(x,a) \in \mathcal{X} \times \mathcal{A}} \mu_{x\mathrm{OPT}(x)} - \mu_{x,a}$.*

We define the $\mathrm{t}^{th}$ model estimate $\mathbf{P}^{\boldsymbol{\Lambda}_t}(\boldsymbol{\mu})$ as the least squares estimate calculated using $\boldsymbol{\mu}$ under a sampling distribution $\{\lambda_{x,a}^t\}_{(x,a) \in \mathcal{X} \times \mathcal{A}}$ at time $t$.

The proof of the Theorem depends on the following Lemmas

**Lemma 4.12.** *If $\boldsymbol{\mu}$ is an interior point of the robust observation region, that is, if $\boldsymbol{\mu} \in \mathrm{Int}(\mathcal{R}^{\mathcal{X}})$, then there exists a $\Delta_{\min} > 0$, such that under any sampling distribution $\{\alpha(x,a,t)\}_{a \in \mathcal{A}}$ at any time $t \geqslant 1$, we have for any context $x$,*

$$\varphi_{x,\mathrm{OPT}(x)}^{\top}\mathbf{P}^{\boldsymbol{\Lambda}_t}(\boldsymbol{\mu}) - \varphi_{x,a}^{\top}\mathbf{P}^{\boldsymbol{\Lambda}_t}(\boldsymbol{\mu}) \geqslant \Delta_{\min},$$

*for any sub-optimal arm $a$ at context $x$.*

The proof follows the same line as in the proof of Lemma 3.18.

**Lemma D.4** (High Confidence Ellipsoids). *Under Assumptions D.1, D.2 and D.3, we have for any that for $t \geqslant 1$ and for any $\delta > 0$, we have,*

$$\left\|\mathbf{P}^{\boldsymbol{\Lambda}_t}(\widehat{\boldsymbol{\mu}_t}) - \mathbf{P}^{\boldsymbol{\Lambda}_t}(\boldsymbol{\mu})\right\|_{V_t} \leqslant R\sqrt{2\log\left(\frac{(1+t/d)^{d/2}}{\delta}\right)},$$

*with probability at least $1-\delta$.*

**Lemma D.5** (Cumulative Regret in the Model Space). *Under Assumptions D.1, D.2 and D.3, we have for any that for $t \geqslant 1$ and for any $\delta > 0$, we have,*

$$\sum_{s=1}^{t}\varphi_{X_s,\mathrm{OPT}(X_s)}^{\top}\mathbf{P}^{\boldsymbol{\Lambda}_s}(\boldsymbol{\mu}) - \varphi_{X_s,A_s}^{\top}\mathbf{P}^{\boldsymbol{\Lambda}_s}(\boldsymbol{\mu}) \leqslant 4\sqrt{t}R\sqrt{\log\left(\frac{(1+t/d)^{d/2}}{\delta}\right)}\sqrt{\log(1+t/d)^d},$$

*with probability at least $1-\delta$*

**Lemma D.6** (Upper Bound on Sup-Optimal Plays). *Under Assumptions D.1, D.2 and D.3, for any contextual bandit instance $\boldsymbol{\mu}$ lying in the robust region $\mathcal{R}^{\mathcal{X}}$, we have for any $t \geqslant 1$, and for any $\delta > 0$,*

$$\sum_{s=1}^{t}\mathbb{1}\{A_S \neq \mathrm{OPT}(X_s)\} \leqslant \frac{4\sqrt{t}R}{\Delta_{\min}}\sqrt{\log\left(\frac{(1+t/d)^{d/2}}{\delta}\right)}\sqrt{\log(1+t/d)^d},$$

*with probability at least $1-\delta$.*

The proofs for the above three lemmas follow the same lines as in the Proof for Lemma C.5 (Here, we subdivide the proof section into three different lemmas just to make the proofs go more straightforward). This gives us the regret for LinUCB in contextual bandits,

*Proof of Theorem 4.13.*

$$\sum_{s=1}^{T}\mu_{X_s,\mathrm{OPT}(X_s)} - \mu_{X_s,A_S}$$

$$=\sum_{s=1}^{T}\mathbb{1}\{A_S \neq \mathrm{OPT}(X_s)\}\Delta_{X_s,A_s}$$

$$\leqslant\sum_{s=1}^{T}\mathbb{1}\{A_S \neq \mathrm{OPT}(X_s)\}\Delta_{\max}$$

$$\leqslant\frac{4\sqrt{t}R\Delta_{\max}}{\Delta_{\min}}\sqrt{\log\left(\frac{(1+t/d)^{d/2}}{\delta}\right)}\sqrt{\log(1+t/d)^d}.$$

$\square$

# E    Weighted Ridge Regression

In our definition of the *robust observation region*, we have implicitly assumed the invertibility of $\mathbf{\Phi}^\top\mathbf{\Lambda}\mathbf{\Phi}$. For the purposes of mathematical rigor, a consistent definition of the robust observation region would be (see Lemma I.4),

**Definition E.1** (**Robust Observation Region**)**.** For a given feature matrix $\mathbf{\Phi}$, we define the $k^{th}$ *robust observation region* $\mathcal{R}_k$, as the set of all $K$ armed bandit instances $\boldsymbol{\mu}$ with optimal arm $k$, such that under any sampling distribution $\{\lambda_i\}_{i=1}^K \in \Delta_K$, the corresponding model estimate, $\mathbf{P}^{\mathbf{\Lambda}}(\boldsymbol{\mu})$, lies in the $k^{th}$ robust parameter region, $\mathbf{\Theta}_k$.

$$\mathcal{R}_k = \Big\{ \boldsymbol{\mu}\in\mathcal{G}_k : \mathbf{P}^{\mathbf{\Lambda}}(\boldsymbol{\mu})\in\mathbf{\Theta}_k \,\forall\,\mathbf{\Lambda}\in\mathcal{P}(\Delta_K) \Big\} \text{ for any arm } k,$$

where

$$\mathcal{P}(\Delta_K) = \Big\{ \mathbf{\Lambda} = \mathrm{diag}(\{\lambda_i\}_{i=1}^K) : \{\lambda_i\}_{i=1}^K \in \Delta_K \,\wedge\, \mathbf{\Phi}^\top\mathbf{\Lambda}\mathbf{\Phi} \text{ is invertible} \Big\}.$$

This necessitates a forced exploration phase in our algorithms. In practice, however, one uses a regularizer $\lambda>0$ to bypass the issue of invertibility. We discuss how our theory of robust regions extends naturally to the case when one uses a weighted regularized least squares estimate rather than the ordinary least squares estimate. Consider the weighted regularized least squares estimate,

$$\mathbf{P}_\lambda^{\mathbf{\Lambda}}(\boldsymbol{\mu}) \triangleq \Big( \mathbf{\Phi}^\top\mathbf{\Lambda}\mathbf{\Phi}+\lambda I \Big)^{-1} \mathbf{\Phi}^\top\mathbf{\Lambda}\boldsymbol{\mu},$$

for some $\lambda>0$. Define the corresponding *regularized robust observation region* as

$$\mathcal{R}_k^\lambda = \Big\{ \boldsymbol{\mu}\in\mathcal{G}_k : \mathbf{P}_\lambda^{\mathbf{\Lambda}}(\boldsymbol{\mu})\in\mathbf{\Theta}_k \,\forall\,\mathbf{\Lambda}\in\mathcal{P}(\Delta_K) \Big\},$$

where

$$\mathcal{P}(\Delta_K) = \Big\{ \mathbf{\Lambda} = \mathrm{diag}(\{\lambda_i\}_{i=1}^K) : \{\lambda_i\}_{i=1}^K \in \Delta_K \Big\}.$$

**Characterizing Robust Region**    We can again give an explicit description of the set $\mathcal{R}_k^\lambda$. Given a feature matrix $\mathbf{\Phi}$, a sampling distribution $\mathbf{\Lambda}\in\mathcal{P}(\Delta_K)$ and a $K$ dimensional element $\boldsymbol{\mu}$, define the following augmented elements

$$\mathbf{\Phi}^* = \begin{bmatrix} \mathbf{\Phi} \\ \sqrt{\lambda}I_{d\times d} \end{bmatrix}, \quad \mathbf{\Lambda}^* = \begin{bmatrix} \mathbf{\Lambda} & 0 \\ 0 & I_{d\times d} \end{bmatrix} \text{ and } \boldsymbol{\mu}^* = \begin{bmatrix} \boldsymbol{\mu} \\ 0 \end{bmatrix},$$

where $\mathbf{\Phi}^*\in\mathbb{R}^{K+d\times d}$, $\mathbf{\Lambda}^*\in\mathbb{R}^{K+d\times K+d}$ and $\boldsymbol{\mu}^*\in\mathbb{R}^{K+d}$. Define the set $\mathcal{J}(\mathbf{\Phi}^*)$ as the set of row indices associated with non-singular $d\times d$ sub-matrices of $\mathbf{\Phi}^*$, and let $\mathcal{I}$ be the set of row indices corresponding to $\{K+1,\cdots,K+d\}$.

**Theorem E.2.** *For any reward vector $\boldsymbol{\mu}$ with optimal arm $k$,*

$$\boldsymbol{\mu}\in\mathcal{R}_k^\lambda \iff \Phi_J^{*^{-1}}\boldsymbol{\mu}_J^*\in\Theta_k$$

*for all $J\in\mathcal{J}(\mathbf{\Phi}^*)\backslash\mathcal{I}$.*

*Proof.* For a sampling distribution $\mathbf{\Lambda}$ we have

$$\mathbf{P}_\lambda^{\mathbf{\Lambda}}(\boldsymbol{\mu}) = \Big( \mathbf{\Phi}^{*^\top}\mathbf{\Lambda}^*\mathbf{\Phi}^* \Big)^{-1} \mathbf{\Phi}^{*^\top}\mathbf{\Lambda}^*\boldsymbol{\mu}^*$$

Thus, from the result in Lemma I.4, we have that regularized model estimate $\mathbf{P}_\lambda^{\mathbf{\Lambda}}(\boldsymbol{\mu})$ to lie within the convex hull of at most $\binom{K+d}{d}$ basic solutions $\Phi_d^{*^{-1}}\boldsymbol{\mu}_d^*$ for any $\mathbf{\Lambda}\in\mathcal{P}(\Delta_K)$. That is,

$$\mathbf{P}_\lambda^{\mathbf{\Lambda}}(\boldsymbol{\mu}) = \sum_{J\in\mathcal{J}(\mathbf{\Phi}^*)} \Big( \frac{\det\mathbf{\Lambda}_J^*\det\Phi_J^{*^2}}{\sum_{K\in\mathcal{J}(\mathbf{\Phi}^*)}\det\mathbf{\Lambda}_K^*\det\Phi_K^{*^2}} \Big) \Phi_J^{*^{-1}}\boldsymbol{\mu}_J^*, \,\forall\mathbf{\Lambda}\in\mathbf{\Lambda}$$

Corresponding to the set of indices of $\mathcal{I}$, we have $\Phi_{\mathcal{I}}^{*^{-1}} \boldsymbol{\mu}_{\mathcal{I}}^* = 0$. Thus decomposing the set of row indices $\mathcal{J}(\boldsymbol{\Phi}^*)$ into $\mathcal{J}(\boldsymbol{\Phi}^*)\backslash\mathcal{I}$ and $\mathcal{I}$, we have, from Lemma I.4,

$$
\begin{aligned}
\mathbf{P}_\lambda^{\boldsymbol{\Lambda}}(\boldsymbol{\mu}) &= \sum_{J \in \mathcal{J}(\boldsymbol{\Phi}^*)} \Big( \frac{\det\Lambda_J^* \det\Phi_J^{*^2}}{\sum_{K \in \mathcal{J}(\boldsymbol{\Phi}^*)} \det\Lambda_K^* \det\Phi_K^{*^2}} \Big) \Phi_J^{*^{-1}} \boldsymbol{\mu}_J^* \\
&= \sum_{J \in \mathcal{J}(\boldsymbol{\Phi}^*)\backslash\mathcal{I}} \Big( \frac{\det\Lambda_J^* \det\Phi_J^{*^2}}{\sum_{K \in \mathcal{J}(\boldsymbol{\Phi}^*)\backslash\mathcal{I}} \det\Lambda_K^* \det\Phi_K^{*^2} + \lambda^d} \Big) \Phi_J^{*^{-1}} \boldsymbol{\mu}_J^* \\
&= \sum_{J \in \mathcal{J}(\boldsymbol{\Phi}^*)\backslash\mathcal{I}} \Big( \frac{\det\Lambda_J^* \det\Phi_J^{*^2}}{\sum_{K \in \mathcal{J}(\boldsymbol{\Phi}^*)\backslash\mathcal{I}} \det\Lambda_K^* \det\Phi_K^{*^2}} \Big) c^\lambda \Phi_J^{*^{-1}} \boldsymbol{\mu}_J^*,
\end{aligned}
$$

where $c^\lambda = \frac{\sum_{K \in \mathcal{J}(\boldsymbol{\Phi}^*)\backslash\mathcal{I}} \det\Lambda_K^* \det\Phi_K^{*^2}}{\sum_{K \in \mathcal{J}(\boldsymbol{\Phi}^*)\backslash\mathcal{I}} \det\Lambda_K^* \det\Phi_K^{*^2} + \lambda^d}$ is necessarily positive for any $\boldsymbol{\Lambda} \in \mathcal{P}(\Delta_{\mathrm{K}})$. Thus we have $\boldsymbol{\mu} \in \mathcal{R}_k^\lambda$, if and only if, $c^\lambda \Phi_J^{*^{-1}} \boldsymbol{\mu}_J^* \in \boldsymbol{\Theta}_k$ for all row indices $J$ in $\mathcal{J}(\boldsymbol{\Phi}^*)\backslash\mathcal{I}$. But since $\boldsymbol{\Theta}_k$ is a convex cone, this implies the result holds if $\Phi_J^{*^{-1}} \boldsymbol{\mu}_J^* \in \boldsymbol{\Theta}_k$.

$\square$

## E.1 LinUCB

We now present an analysis of LinUCB (Abbasi-Yadkori et al., 2011) without a forced exploration phase and under the standard assumptions. That is without the Assumption C.3. This is the result that one would find, say in, Abbasi-Yadkori et al. (2011). But we present here without the realizability assumption.

**Assumption E.3** (Bounded Features). We assume that for any arm $i$ in the arm set $[K]$, the corresponding feature $\varphi_i$ is bounded in the $l^2$ norm by $L$, that is,

$$\|\varphi_i\|_2 \leqslant L \ \forall i \in [K].$$

**Assumption E.4** (Regularization). We assume that the regularizer $\lambda$, so chosen satisfies

$$\lambda \geqslant \max\{1, L^2\},$$

where $L$ is as defined in Assumption E.3

**Theorem E.5.** *Given a feature matrix $\boldsymbol{\Phi}$ satisfying Assumptions E.3, regularization parameter $\lambda$ satisfying Assumption E.4, and any bandit parameter $\boldsymbol{\mu}$ which is an interior point of the regularized robust observation region, $\mathcal{R}_{\mathrm{OPT}(\boldsymbol{\mu})}^\lambda$ and satisfies Assumption C.1, the LinUCB algorithm, (Abbasi-Yadkori et al., 2011), achieves regret of at most $\tilde{O}(d\sqrt{t})$.*

*That is, with $\beta_t(\delta)$ set as $2R^2 \log\Big( \frac{(\lambda + tL^2/d)^{d/2}}{\delta} \Big)$ for any $\delta > 0$, we have with probability at least $1 - \delta$, for $t \geqslant 1$,*

$$\sum_{s=1}^t \mu_{\mathrm{OPT}(\boldsymbol{\mu})} - \mu_{A_s} \leqslant \frac{4\sqrt{t}R\Delta_{\max}}{\Delta_{\min}} \sqrt{\log\Big( \frac{(1 + tL^2/\lambda d)^{d/2}}{\delta} \Big)} \sqrt{\log(1 + tL^2/\lambda d)^d},$$

*where $\Delta_{\min}$ is as defined above and $\Delta_{\max}$ is defined as the worst sub-optimal gap, that is, $\Delta_{\max} = \max_{i \in [K]} \mu_{\mathrm{OPT}(\boldsymbol{\mu})} - \mu_i$.*

We provide a sketch of the proof: As in Lemma, 3.18, we have the following result for any bandit instance $\boldsymbol{\mu}$ belonging to the interior of the regularized robust observation region $\mathcal{R}_{\mathrm{OPT}(\boldsymbol{\mu})}^\lambda$.

**Lemma E.6** (Lower Bound of per instant Regret in the Function Space). *If $\boldsymbol{\mu}$ is an interior point of the regularized robust observation region, that is, if $\boldsymbol{\mu} \in \mathrm{Int}\big(\mathcal{R}_{\mathrm{OPT}(\boldsymbol{\mu})}^\lambda\big)$, then there exists a $\Delta_{\min} > 0$, such that for any sampling distribution $\{\lambda_{i(t)}\}_{i \in [K]}$ at any time $t \geqslant 1$, we have $\varphi_{\mathrm{OPT}(\boldsymbol{\mu})}^\top \mathbf{P}_\lambda^{\boldsymbol{\Lambda_t}}(\boldsymbol{\mu}) - \varphi_i^\top \mathbf{P}_\lambda^{\boldsymbol{\Lambda_t}}(\boldsymbol{\mu}) \geqslant \Delta_{\min}$ for any sub-optimal arm $i$.*

We, also have the following result, as in Lemma C.5,

**Lemma E.7** (Upper Bound of Sub-Optimal Plays). *Given a feature matrix $\mathbf{\Phi}$ satisfying Assumptions E.3, regularization parameter $\lambda$ satisfying Assumption E.4, and any bandit parameter $\boldsymbol{\mu}$ which is an interior point of the regularized robust observation region, $\mathcal{R}_{\text{OPT}(\boldsymbol{\mu})}^{\lambda}$ and satisfies Assumption C.1, the LinUCB algorithm, (Abbasi-Yadkori et al., 2011), plays sub-optimally at most $\tilde{O}(d\sqrt{t})$.*

*That is, with $\beta_t(\delta)$ set as $2R^2\log\left(\frac{(1+tL^2/\lambda d)^{d/2}}{\delta}\right)$ for any $\delta > 0$, we have with probability at least $1-\delta$, for $t \geqslant 1$*

$$\sum_{s=1}^{t}\mathbb{1}\{A_s \neq \text{OPT}(\boldsymbol{\mu})\} \leqslant \frac{4\sqrt{t}R}{\Delta_{\min}}\sqrt{\log\left(\frac{(1+tL^2/\lambda d)^{d/2}}{\delta}\right)}\sqrt{\log(1+tL^2/\lambda d)^d},$$

*where $\Delta_{\min}$ is as defined in the previously.*

*Proof.* Note that, following the arguments as in Lemma C.5, we have, from Technical Lemma I.1,

$$\left\|\mathbf{P}_{\lambda}^{\boldsymbol{\Lambda}_t}(\widehat{\boldsymbol{\mu}}_t) - \mathbf{P}_{\lambda}^{\boldsymbol{\Lambda}_t}(\boldsymbol{\mu})\right\|_{V_t} \leqslant R\sqrt{2\log\left(\frac{\det V_t^{1/2}\det\lambda I^{-1/2}}{\delta}\right)}, \tag{3}$$

with probability at least $1-\delta$ for all $t \geqslant 1$. Therefore, the per-instant, value function regret, can be bounded by at most

$$\varphi_{\text{OPT}(\boldsymbol{\mu})}^{\top}\mathbf{P}_{\lambda}^{\boldsymbol{\Lambda}_t}(\boldsymbol{\mu}) - \varphi_{A_t}^{\top}\mathbf{P}_{\lambda}^{\boldsymbol{\Lambda}_t}(\boldsymbol{\mu}) \leqslant 2\|\varphi_{A_t}\|_{V_t^{-1}}\sqrt{\beta_t(\delta)},$$

where $\beta_t(\delta) = 2R^2\log\left(\frac{\det V_t^{1/2}\det\lambda I^{-1/2}}{\delta}\right)$, with probability at least $1-\delta$ for all $t \geqslant 1$. Using Technical Lemma I.2, we have

$$2R^2\log\left(\frac{\det V_t^{1/2}\det\lambda I^{-1/2}}{\delta}\right) \leqslant 2R^2\log\left(\frac{(1+tL^2/\lambda d)^{d/2}}{\delta}\right).$$

Therefore, the cumulative regret in the function space is (Following from the steps in Lemma C.5)

$$\sum_{s=1}^{t}\varphi_{\text{OPT}(\boldsymbol{\mu})}^{\top}\mathbf{P}^{\boldsymbol{\Lambda}_s}(\boldsymbol{\mu}) - \varphi_{A_s}^{\top}\mathbf{P}^{\boldsymbol{\Lambda}_s}(\boldsymbol{\mu})$$

$$\leqslant 2\sqrt{2}R\sum_{s=1}^{t}\|\varphi_{A_s}\|_{V_s^{-1}}\sqrt{\log\left(\frac{(1+sL^2/\lambda d)^{d/2}}{\delta}\right)}$$

$$\leqslant 2\sqrt{2}R\sqrt{\log\left(\frac{(1+tL^2/\lambda d)^{d/2}}{\delta}\right)}\sqrt{t}\sqrt{\sum_{s=1}^{t}\|\varphi_{A_s}\|_{V_s^{-1}}^2}$$

$$\leqslant 4\sqrt{t}R\sqrt{\log\left(\frac{(1+tL^2/\lambda d)^{d/2}}{\delta}\right)}\sqrt{\log\frac{\det V_t}{\det\lambda I}}$$

$$\leqslant 4\sqrt{t}R\sqrt{\log\left(\frac{(1+tL^2/\lambda d)^{d/2}}{\delta}\right)}\sqrt{\log(1+tL^2/\lambda d)^d},$$

where for the last two lines we have used technical Lemma I.2 and Lemma I.3. The proof finishes as in Lemma C.5. □

The proof of Theorem E.5 is completed analogously.

# F   Experiments

In this section, we run some simple experiments to corroborate our findings.

## F.1   Bandits

In this section, we shall use Theorem 3.12 to explicitly calculate the *robust observation region* for a simple feature matrix. We shall then sample bandit instances from the calculated *robust observation region* and run $\varepsilon$-greedy algorithm on these bandit instances.

**Calculation of Robust Regions**    We choose an arbitrary feature matrix, $\Phi$ as $\begin{bmatrix} 2 & 3 \\ 4 & 5 \\ 2 & 1 \end{bmatrix}$.

We shall start by computing the *robust parameter regions*. Recall from the definition of the robust parameter region $\Theta_i$ as the domain of the feature matrix $\Phi$ such that the range belongs to the $i^{th}$ greedy region $\mathcal{G}_i$. Thus for any arm in $[3]$, we have $\Theta_i = \left\{ \theta \in \mathbb{R}^2 : \Phi\theta \in \mathcal{G}_i \right\}$. Thus, for each $i \in [3]$, we must solve for 2 linear inequalities in 2 unknowns. Namely for $\Theta_1$ we have the following set of inequalities

$$\varphi_1^\top \theta - \varphi_2^\top \theta > 0$$
$$\varphi_1^\top \theta - \varphi_3^\top \theta > 0,$$

which upon solving, we get the following condition $\theta_1 < -\theta_2 \wedge \theta_2 > 0$ for any $\theta = \begin{bmatrix} \theta_1 \\ \theta_2 \end{bmatrix}$ to belong to $\Theta_1$. Similarly, for the regions $\Theta_2$ and $\Theta_3$ we have the following set of equations,

| $\varphi_2^\top \theta - \varphi_1^\top \theta > 0$ |
| --- |
| $\varphi_2^\top \theta - \varphi_3^\top \theta > 0$ |

| $\varphi_3^\top \theta - \varphi_2^\top \theta > 0$ |
| --- |
| $\varphi_3^\top \theta - \varphi_1^\top \theta > 0$ |

Upon solving, we arrive at the following descriptions of the *robust parameter regions*

$$\Theta_1 = \{\theta \in \mathbb{R}^2 : \theta_1 < -\theta_2 \wedge \theta_2 > 0\}$$
$$\Theta_2 = \{\theta \in \mathbb{R}^2 : (\theta_1 > 0 \wedge \theta_2 > -\theta_1/2) \vee (\theta_1 < 0 \wedge \theta_2 > -\theta_1)\}$$
$$\Theta_3 = \{\theta \in \mathbb{R}^2 : \theta_2 < 0 \wedge \theta_2 < -\theta_1/2\}.$$

In Figure 4, as a matter of interest, we plot *robust parameter regions* in $\mathbb{R}^2$. We note that the *robust parameter regions* partitions the parameter space $\mathbb{R}^2$; this is unsurprising since the greedy regions $\mathcal{G}_i$ partition the $\mathbb{R}^K$ space.

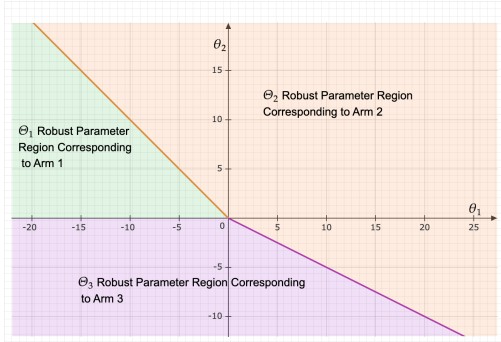

Figure 4: The parameter space $\mathbb{R}^2$ is partitioned into disjoint sets of the *robust parameter regions* corresponding to the different arms for feature matrix $\Phi = \begin{bmatrix} 2 & 3 \\ 4 & 5 \\ 2 & 1 \end{bmatrix}$.

With the exact descriptions of the sets $\Theta_i$, we can calculate the *robust observation regions*. Using Theorem 3.12), this turns out to be the set of all $\mu$ such that $\Phi_2^{-1}\mu_2$ belongs to $\Theta_i$ for every $2 \times 2$ full rank sub-matrix of $\Phi$. Thus for the robust region, $\mathcal{R}_1$, we have the following equations,

$$\begin{bmatrix} 2 & 3 \\ 4 & 5 \end{bmatrix}^{-1} \begin{bmatrix} \mu_1 \\ \mu_2 \end{bmatrix} \in \Theta_1, \quad \begin{bmatrix} 2 & 3 \\ 2 & 1 \end{bmatrix}^{-1} \begin{bmatrix} \mu_1 \\ \mu_3 \end{bmatrix} \in \Theta_1, \quad \begin{bmatrix} 4 & 5 \\ 2 & 1 \end{bmatrix}^{-1} \begin{bmatrix} \mu_2 \\ \mu_3 \end{bmatrix} \in \Theta_1.$$

Thus solving for $\mu_1, \mu_2$ and $\mu_3$ gives the description for $\mathcal{R}_1$ as

$$\mathcal{R}_1 = \left\{ \mu \in \mathbb{R}^3 : (\mu_1 > \mu_2/2) \wedge (\mu_1 > \mu_2) \wedge (\mu_2 > 2\mu_3) \wedge (\mu_2 < -\mu_3) \wedge (\mu_1 > \mu_3) \wedge (\mu_1 < -\mu_3) \right\}.$$

Similarly from the descriptions of $\Theta_2$ and $\Theta_3$, we get the following descriptions for $\mathcal{R}_2$ and $\mathcal{R}_3$ respectively,

$$\mathcal{R}_2 = \Big\{ \boldsymbol{\mu} \in \mathbb{R}^3 : (\mu_1 < \mu_2 < 3\mu_1) \wedge (-\mu_1 < \mu_3 < 3\mu_1) \wedge \big( (\mu_3 < \mu_2 < 5\mu_3) \vee (\mu_2 > 5\mu_3 \wedge \mu_2 > -\mu_3) \big) \Big\}$$

$$\mathcal{R}_3 = \Big\{ \boldsymbol{\mu} \in \mathbb{R}^3 : (\mu_1 < \mu_2/2) \wedge (\mu_1 < \mu_2/3) \wedge (\mu_3 > 3\mu_1) \wedge (\mu_3 > \mu_1) \wedge (\mu_3 > \mu_2) \wedge (\mu_3 > \mu_2/2) \Big\}.$$

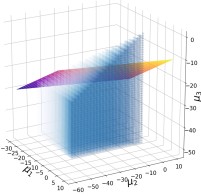
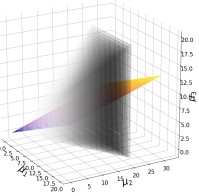

(a) The robust region for arm one $\mathcal{R}_1$, shown in the blue shade, is a subset of $\mathbb{R}^3$. We depict the range space of the feature matrix as the plane.

(b) The robust region for arm two $\mathcal{R}_2$ shown in the gray shade, is a subset of $\mathbb{R}^3$. We depict the range space of the feature matrix as the plane.

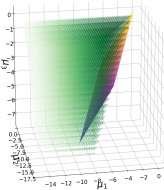

(c) The robust region for arm three $\mathcal{R}_3$, shown in the green shade, is a subset of $\mathbb{R}^3$. The range space of the feature matrix is depicted as the plane.

Figure 5: Visualization of the *robust observation regions* for a three-armed bandit problem, calculated for the feature matrix $\boldsymbol{\Phi} = \begin{bmatrix} 2 & 3 \\ 4 & 5 \\ 2 & 1 \end{bmatrix}$, along with the range space of the feature matrix $\boldsymbol{\Phi}\theta$. Note that these are 3-dimensional plots with the robust regions $\mathcal{R}_i$ shown in shaded regions of blue, gray, and green colors. These regions are subsets of $\mathbb{R}^3$ whereas the range space of the feature matrix, shown in a "plasma" color, spans $\mathbb{R}^2$.

In Figure 5, we try to visualize these regions in $\mathbb{R}^3$. In particular Figures 5(a), 5(b) and 5(c) represent the regions $\mathcal{R}_1, \mathcal{R}_2$ and $\mathcal{R}_3$ respectively. Note that these images represent regions in $\mathbb{R}^3$. In particular, we have used a shading effect to highlight the three-dimensional nature of the regions. The range space of the feature matrix, $\boldsymbol{\Phi}\theta$, has also been highlighted as a two-dimensional plane, passing through the robust regions in Figures 5(a) and 5(b), while bordering the region $\mathcal{R}_3$ in Figure 5(c). Note that we have plotted these images by restricting $\boldsymbol{\mu}$ to lie within a bounded region, which might give the appearance of being bound. As could be deduced from the set theoretic descriptions of the *robust observation regions* these are convex cones and are unbounded sets.

**Experiments with $\varepsilon$-greedy Algorithm.** We sample 10 instances from the robust region $\mathcal{R}_2$. We observe from Figure 6 the mean and dispersion of the cumulative regret generated by $\varepsilon$-greedy algorithm with $\varepsilon_t = 1/\sqrt{t}$ on these sampled instances. We also note the misspecification error (denoted by $\rho$), the maximum sub-optimality gap (denoted by $\Delta_{\max}$), and the minimum sub-optimality gap (denoted by $\Delta_{\min}$) for each of these instances. To demonstrate our results with high probability, we form confidence intervals of the cumulative regret with three standard deviations. We observe that the cumulative regret grows at a sub-linear rate with high probability.

**Observations** We note that instances with higher $\Delta_{\max}$ values tend to have higher regret than the instances with lower $\Delta_{\max}$ values. In this regard, we note that the misspecification error $\rho$ does not influence the regret as much as the $\Delta_{\max}$, corroborating our theory. For example, note that the regret curve corresponding to misspecification error $\rho = 9.02$ is lower than

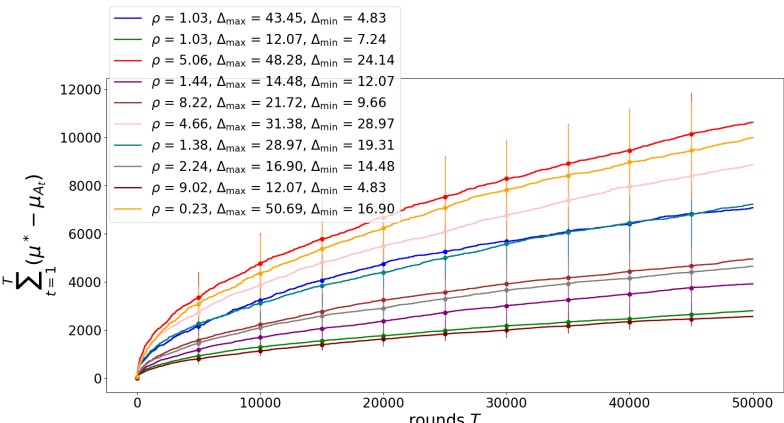

Figure 6: The growth of the cumulative regret for 10 misspecified bandit instances sampled from the robust region of $\mathcal{R}_2$ under the $\varepsilon$-greedy algorithm with $\varepsilon_t = 1/\sqrt{t}$. The plot represents the average of 10 trials. The $Y$-axis denotes the cumulative regret $\sum_{t=1}^{T} \mu^* - \mu_{A_t}$. The $X$-axis denotes the rounds $T$. We observe the sub-linear growth trend of the cumulative regret. For each instance the values of the $l_\infty$ misspecification error ($\rho$), the maximum sub-optimality gap ($\Delta_{\max}$) and the minimum sub-optimality gap ($\Delta_{\min}$) are also noted. We observe that instances with higher $\Delta_{\max}$ suffer more regret at any time than instances with lower $\Delta_{\max}$ as expected from our theorem.

the curve corresponding to $\rho = 0.23$; we can explain this using the fact that the $\Delta_{\max}$ of the former curve is 12.07 whereas, for the latter curve, it is 50.69. One can observe that misspecification error does not play a significant role in the near regret curves for the instances whose $\Delta_{\max}$ are the same marked as 12.07; however, one has misspecification error $\rho$ as 1.03 whereas the other has 9.02. We also note the presence of one sampled instance whose $\Delta_{\min}$ is less than the misspecification error $\rho$ (The example in focus has $\Delta_{\min} = 4.83$ while misspecification error $\rho$ is 9.02); this is in sharp contrast to the type of robust instances considered in the works of Liu et al. (2023).

### F.2 Contextual Bandits

In this section, we shall use Theorem 4.6 to explicitly calculate the *robust observation region* for a simple feature matrix. We shall then sample contextual bandit instances from the calculated *robust observation region* and run $\varepsilon$-greedy algorithm on these contextual bandit instances.

**Calculation of Robust Regions**   Let $\Phi_{x_1} = \begin{bmatrix} 2 & 3 \\ 4 & 5 \\ 2 & 1 \end{bmatrix}$ and $\Phi_{x_2} = \begin{bmatrix} 2 & 3 \\ 4 & 5 \\ 6 & 7 \end{bmatrix}$.

The above feature matrices represent a 2 context bandit, each with a 3 arms setting. Note that the instance is in $\mathbb{R}^6$, and thus the *robust observation regions* would be of dimension at most 6. From the definition of the *robust parameter region* for context $x_1$ we have,

$$\Theta_1^{x_1} = \{\theta \in \mathbb{R}^2 : \theta_1 < -\theta_2 \wedge \theta_2 > 0\}$$
$$\Theta_2^{x_1} = \{\theta \in \mathbb{R}^2 : (\theta_1 > 0 \wedge \theta_2 > -\theta_1/2) \vee (\theta_1 < 0 \wedge \theta_2 > -\theta_1)\}$$
$$\Theta_3^{x_1} = \{\theta \in \mathbb{R}^2 : \theta_2 < 0 \wedge \theta_2 < -\theta_1/2\}.$$

For context $x_2$, we can repeat the same procedure to find the *robust parameter space for context $x_2$* as

$$\Theta_1^{x_2} = \{\theta \in \mathbb{R}^2 : \theta_1 < -\theta_2\}$$
$$\Theta_3^{x_2} = \{\theta \in \mathbb{R}^2 : \theta_1 > -\theta_2\}.$$

Note that for this feature matrix choice, we have found that the robust parameter region for arm two at context $x_2$, $\Theta_2^{x_2}$ is $\emptyset$. We use Theorem 4.6 to evaluate the *robust observation regions*. Observe that a bandit instance $\boldsymbol{\mu} \in \mathcal{R}_i^{x_1} \cap \mathcal{R}_j^{x_2}$ if and only if the basic solutions belong to $\Theta_i^{x_1} \cap \Theta_j^{x_2}$. The non empty intersections of $\Theta_i^{x_1}$ and $\Theta_j^{x_2}$ are as follows

$$\Theta_1^{x_1} \cap \Theta_1^{x_2} = \Theta_1^{x_1}$$
$$\Theta_3^{x_1} \cap \Theta_1^{x_2} = \{\theta \in \mathbb{R}^2 : \theta_1 < -\theta_2 \wedge \theta_2 < 0\}$$
$$\Theta_2^{x_1} \cap \Theta_3^{x_2} = \Theta_2^{x_1}$$
$$\Theta_3^{x_1} \cap \Theta_3^{x_2} = \{\theta \in \mathbb{R}^2 : \theta_2 < 0, -\theta_1 < \theta_2 < -\theta_1/2\}.$$

Thus $\mathcal{R}_3^{x_1} \cap \mathcal{R}_1^{x_2}$ is the set of all $\boldsymbol{\mu} \in \mathbb{R}^6$, such that the basic solutions are in $\Theta_3^{x_1} \cap \Theta_1^{x_2}$. Solving, gives us the description of the set $\mathcal{R}_1^{x_1} \cap \mathcal{R}_1^{x_2}$ as the set of all $\boldsymbol{\mu} \in \mathbb{R}^6$ such that the following conditions are satisfied

$$\mathcal{R}_3^{x_1} \cap \mathcal{R}_1^{x_2} = \Big\{ \boldsymbol{\mu} \in \mathbb{R}^6 : \mu_1 > \mu_2, \mu_1 < \mu_2/2, \mu_1 < \mu_3, \mu_1 < -\mu_3,$$
$$\mu_1 > \mu_5, \mu_1 < \mu_5/2, \mu_1 > \mu_6, \mu_1 < \mu_6/3,$$
$$\mu_2 < 2\mu_3, \mu_2 < -\mu_3, \mu_4 < \mu_2/2, \mu_2 < \mu_4,$$
$$\mu_2 > \mu_6, \mu_2 < 2/3\mu_6, \mu_4 < \mu_3, \mu_3 < -\mu_4,$$
$$\mu_5 < 2\mu_3, \mu_3 < -\mu_5, \mu_6 < 3\mu_3, \mu_3 < -\mu_6,$$
$$\mu_4 > \mu_5, \mu_4 < \mu_5/2, \mu_4 > \mu_6, \mu_4 < \mu_6/3,$$
$$\mu_5 > \mu_6, \mu_5 < 2/3\mu_6 \Big\}.$$

**Experiments with $\varepsilon$-greedy Algorithm.** We observe from Figure 7 the mean and dispersion of the cumulative regret on 10 contextual bandit instances, sampled from the robust region $\mathcal{R}_3^{x_1} \cap \mathcal{R}_1^{x_2}$, for the $\varepsilon$-greedy algorithm with $\varepsilon_t = 1/\sqrt{t}$ for different context distributions. We also note the misspecification error (denoted by $\rho$), the maximum sub-optimality gap (denoted by $\Delta_{\max}$), and the minimum sub-optimality gap (denoted by $\Delta_{\min}$) for each such sampled instance. To demonstrate our results with high probability, we form confidence intervals of the cumulative regret with three standard deviations. We can observe that the growth of the cumulative regret is sub-linear with high probability.

**Observations** We note that instances with higher $\Delta_{\max}$ values tend to have higher regret than the instances with lower $\Delta_{\max}$ values. In this regard, we note that the misspecification error $\rho$ does not influence the regret as much as the $\Delta_{\max}$, corroborating our theory. For example, note that in Figure 7(c), the regret curve corresponding to misspecification error $\rho = 2.11$ is lower than the curve corresponding to $\rho = 1.58$. We can explain this observation by noting that the $\Delta_{\max}$ of the former curve is 10.53, whereas, for the latter curve, it is 25.26. One can observe the fact that misspecification error does not play a significant role by noting in the different regret curves for the instances whose misspecification error $\rho$ are the same marked as 2.11 but has different $\Delta_{\max}$ as 21.05, 16.84 and 10.53. We also note the presence of one sampled instance whose $\Delta_{\min}$ is less than the misspecification error $\rho$ (The example in focus has $\Delta_{\min} = 2.11$ while misspecification error $\rho$ is 2.63). This example contrasts sharply with the type of robust instances considered in the works of Zhang et al. (2023).

# G Robust Features for Contextual Bandits

The robust features construction for Bandits (Section 3.3) can be extended to Contextual Bandits as well. Note that any two-context, two-armed contextual bandit instance lies in $\mathbb{R}^4$ space and thus can be represented as an element $(w, x, y, z)$ in $\mathbb{R}^4$. Without loss of generality let any contextual bandit instance be represented by $[\mu_{s_1, a_1}, \mu_{s_1, a_2}, \mu_{s_2, a)}, \mu_{s_2, a_2}]^\top$, that is the coordinates of $\mathbb{R}^4$ represent the reward of first context first action, first context second action etc. Consider the greedy regions which partition $\mathbb{R}^4$, as

$$\mathcal{G}_1^1 = \big\{(w, x, y, z) \text{ s.t } w > x, y > z\big\}$$
$$\mathcal{G}_2^1 = \big\{(w, x, y, z) \text{ s.t } w > x, z > y\big\}$$
$$\mathcal{G}_1^2 = \big\{(w, x, y, z) \text{ s.t } x > w, y > z\big\}$$
$$\mathcal{G}_2^2 = \big\{(w, x, y, z) \text{ s.t } x > w, z > y\big\}.$$

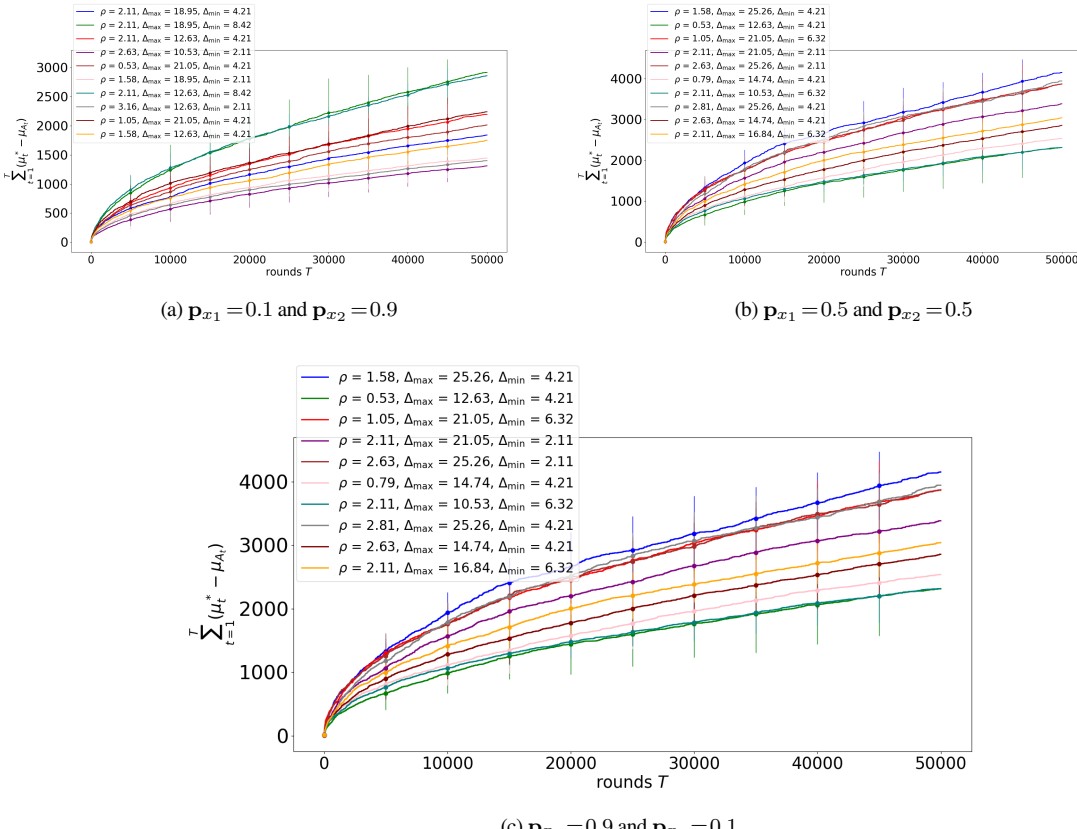

(a) $\mathbf{p}_{x_1}=0.1$ and $\mathbf{p}_{x_2}=0.9$

(b) $\mathbf{p}_{x_1}=0.5$ and $\mathbf{p}_{x_2}=0.5$

(c) $\mathbf{p}_{x_1}=0.9$ and $\mathbf{p}_{x_2}=0.1$

Figure 7: The growth of the cumulative regret for 10 misspecified bandit instances sampled from the robust region of $\mathcal{R}_3^{x_1}\cap\mathcal{R}_1^{x_2}$ under the $\varepsilon$-greedy algorithm with $\varepsilon_t=1/\sqrt{t}$ for different context distributions $\mathbf{p}_{x_1}$ and $\mathbf{p}_{x_2}$. The plots represent the average of 10 trials. The $Y$-axis denotes the cumulative regret $\sum_{t=1}^{T}\mu_t^*-\mu_{A_t}$. The $X$-axis denotes the rounds $T$. We observe the sub-linear growth trend of the cumulative regret. For each instance the values of the $l_\infty$ misspecification error ($\rho$), the maximum sub-optimality gap ($\Delta_{\max}$) and the minimum sub-optimality gap ($\Delta_{\min}$) are also noted. we can observe that instances with higher $\Delta_{\max}$ suffer more regret at any time than instances with lower $\Delta_{\max}$ as expected from our theorem.

Let

$$\mathcal{M}_1^1(\varepsilon)=\big\{(w,x,y,z) \text{ s.t } w>x+\varepsilon, y>z+\varepsilon\big\} \tag{*}$$
$$\mathcal{M}_2^1(\varepsilon)=\big\{(w,x,y,z) \text{ s.t } w>x+\varepsilon, z>y+\varepsilon\big\}$$
$$\mathcal{M}_1^2(\varepsilon)=\big\{(w,x,y,z) \text{ s.t } x>w+\varepsilon, y>z+\varepsilon\big\}$$
$$\mathcal{M}_2^2(\varepsilon)=\big\{(w,x,y,z) \text{ s.t } x>w+\varepsilon, z>y+\varepsilon\big\}$$

for a fixed $\varepsilon$ be the the the disjoint 4 dimensional manifolds with each $\mathcal{M}_i^j(\varepsilon)\subset\mathcal{G}_i^j$. Note that any contextual bandit $\boldsymbol{\mu}\in\mathbb{R}^4$ must lie in one of the greedy regions $\mathcal{G}_i^j$. We define the feature representation by the disjoint union of the manifolds

$$\mathcal{F}(\varepsilon)=\bigsqcup_{i,j}\partial\mathcal{M}_i^j(\varepsilon)=\{w=x+\varepsilon,y=z+\varepsilon\}\sqcup\{w=x+\varepsilon,z=y+\varepsilon\}\sqcup\{x=w+\varepsilon,y=z+\varepsilon\}\cup\{x=w+\varepsilon,z=y+\varepsilon\}.$$

We have the following theorem, whose proof follows along the lines of Theorem 3.27.

**Theorem G.1.** *Any contextual bandit instance lying in the region $\sqcup_{i,j}\mathcal{M}_i^j(\varepsilon)$ is robust for the feature representation defined above by the function $\mathcal{F}(\varepsilon)$.*

## H  Example of a Misspecified but Robust (Stochastic) MDP

This section analyzes a stochastic MDP under an arbitrary behavioral policy. For the function class, we shall use the one introduced in the previous Appendix G.

**MDP Description**  Consider the two-stage MDP, $\mathcal{M}$ as in Figure 3(b) with three states. At stage $h=1$, we are at state $s_1$. We can take two actions depending on a behavioral policy, and move to either state $s_2$ or state $s_3$ depending on the transition probability, $\mathbb{P}\{s_j \mid s_1, a_i\}$. We observe an associated reward of $r_{11}$ or $r_{12}$ respectively. At stage $h=2$, one can again choose action $a_1$ or action $a_2$ and get an associated reward of $r_{21}$ or $r_{22}$ or $r_{31}$ or $r_{32}$ respectively. The rewards are such that $r_{11} > r_{12}$, $r_{21} > r_{22}$ and $r_{31} > r_{32}$. To ensure that employing a myopic greedy strategy fails, we require that the optimal policy at each state be $\pi^*(s_1) = a_2$, $\pi^*(s_2) = a_1$ and $\pi^*(s_3) = a_1$.

**Behavioral Policy**  We have an arbitrary behavioral policy $\pi_b$ and denote the probability of a transition, $\mathbb{P}^{\pi_b}(s_h = a, a_h = a, s_{h+1} = s')$, by $\alpha(s, a, s')$. Thus, at stage $h = 1$, we have the following transition probabilities $\alpha(s_1, a_1, s_2)$, $\alpha(s_1, a_1, s_3)$, $\alpha(s_1, a_2, s_2)$ and $\alpha(s_1, a_2, s_3)$. For stage $h = 2$, since $s'$ is a terminal state, we drop the dependency of $s'$ from the notation and have $\alpha(s_2, a_1)$, $\alpha(s_2, a_2)$, $\alpha(s_3, a_1)$ and $\alpha(s_3, a_2)$.

**Function Class**  We choose the function class $\mathcal{F}_2$ and $\mathcal{F}_1$ for stages two and one as the one described in Appendix $G$. Specifically, $\mathcal{F}_2$ is the function class introduced for the two-context-two-armed contextual bandit, described in Appendix $G$. $\mathcal{F}_1$ is the two-dimensional variant of the function class as shown in Figure 3(a).

**Condition for Robustness  Stage Two**  At stage $h = 2$, let the function approximate value be $\boldsymbol{f_2}$ with elements $[\mu_{21}, \mu_{22}, \mu_{31}, \mu_{32}]^\top$. The reward instance $\boldsymbol{r_2}$ is the vector of elements $[r_{21}, r_{22}, r_{31}, r_{32}]^\top$. Thus, based on the frequency of observations, we solve for $\boldsymbol{f_2}$ as the following least squares problem.

$$\boldsymbol{f_2} = \underset{\boldsymbol{f} \in \mathcal{F}_2}{\arg\min} (\boldsymbol{f} - \boldsymbol{r_2})^\top \boldsymbol{\Lambda} (\boldsymbol{f} - \boldsymbol{r_2})$$

where $\boldsymbol{\Lambda} = \begin{pmatrix} \alpha(s_2, a_1) & 0 & 0 & 0 \\ 0 & \alpha(s_2, a_2) & 0 & 0 \\ 0 & 0, & \alpha(s_3, a_1) & 0 \\ 0 & 0 & 0 & \alpha(s_3, a_2) \end{pmatrix}$.

Note that since $r_{21} > r_{22}$ and $r_{31} > r_{32}$, we have, $\boldsymbol{r_2} \in \mathcal{M}_1^1$, where $\mathcal{M}_1^1$ is the robust region as per our notation in Section G, equation *. Thus, we have

$$\mu_{21} = \frac{\alpha(s_2, a_1) r_{21} + \alpha(s_2, a_2) r_{22} + \alpha(s_2, a_2)\varepsilon}{\alpha(s_2, a_1) + \alpha(s_2, a_2)}$$

$$\mu_{22} = \frac{\alpha(s_2, a_1) r_{21} + \alpha(s_2, a_2) r_{22} - \alpha(s_2, a_1)\varepsilon}{\alpha(s_2, a_1) + \alpha(s_2, a_2)}$$

$$\mu_{31} = \frac{\alpha(s_3, a_1) r_{31} + \alpha(s_3, a_2) r_{32} + \alpha(s_3, a_2)\varepsilon}{\alpha(s_3, a_1) + \alpha(s_3, a_2)}$$

$$\mu_{32} = \frac{\alpha(s_3, a_1) r_{31} + \alpha(s_3, a_2) r_{32} - \alpha(s_3, a_1)\varepsilon}{\alpha_2(s_3, a_1) + \alpha(s_3, a_2)}$$

It can be observed that $\mu_{21} > \mu_{22}$ and $\mu_{31} > \mu_{32}$. That is $\max_a f_2(s_2, a) = \mu_{21}$ and $\max_a f_2(s_3, a) = \mu_{31}$. Thus $\arg\max_a f_2(s_2, a) = \pi^*(s_2)$ and $\arg\max_a f_2(s_3, a) = \pi^*(s_3)$.

**Stage One**   Let $f_1 \in \mathcal{F}_1$ be the value function approximation at stage 1 with elements $[\mu_{11}, \mu_{12}]^\top$. Let the reward vector be $r_1$ be $[r_{11}, r_{12}]^\top$. Thus we have the following regression problem.

$$
\begin{aligned}
f_1 = \underset{f \in \mathcal{F}_1}{\operatorname{argmin}} \,& \alpha(s_1,a_1,s_2)\big[f(s_1,a_1) - r_{11} - \max_a f_2(s_2,a)\big]^2 + \alpha(s_1,a_1,s_3)\big[f(s_1,a_1) - r_{11} - \max_a f_2(s_3,a)\big]^2 \\
& + \alpha(s_1,a_2,s_2)\big[f(s_1,a_2) - r_{12} - \max_a f_2(s_2,a)\big]^2 + \alpha(s_1,a_2,s_3)\big[f(s_1,a_2) - r_{12} - \max_a f_2(s_3,a)\big]^2 \\
= \underset{(\mu_{11},\mu_{12})\in\mathcal{F}_1}{\operatorname{argmin}} \,& \big(\alpha(s_1,a_1,s_2) + \alpha(s_1,a_1,s_3)\big)\Big(\mu_{11} - r_{11} - \frac{\alpha(s_1,a_1,s_2)\mu_{21} + \alpha(s_1,a_1,s_3)\mu_{31}}{\alpha(s_1,a_1,s_2) + \alpha(s_1,a_1,s_3)}\Big)^2 + \\
& \big(\alpha(s_1,a_2,s_2) + \alpha(s_1,a_2,s_3)\big)\Big(\mu_{12} - r_{12} - \frac{\alpha(s_1,a_2,s_2)\mu_{21} + \alpha(s_1,a_2,s_3)\mu_{31}}{\alpha(s_1,a_2,s_2) + \alpha(s_1,a_2,s_3)}\Big)^2
\end{aligned}
$$

Thus, for the optimal policy at state $s_1$ to be action $a_2$, we need the following condition to hold,

$$
r_{11} + \frac{\alpha(s_1,a_1,s_2)\mu_{21} + \alpha(s_1,a_1,s_3)\mu_{31}}{\alpha(s_1,a_1,s_2) + \alpha(s_1,a_1,s_3)} < r_{12} + \frac{\alpha(s_1,a_2,s_2)\mu_{21} + \alpha(s_1,a_2,s_3)\mu_{31}}{\alpha(s_1,a_2,s_2) + \alpha(s_1,a_2,s_3)} \,.
$$

Thus, any MDP $\mathcal{M}$ with behavioral policy $\pi_b$ that satisfies the above condition is robust.

# I   Technical Lemmas

**Lemma I.1** (Self Normalized Bound for Vector Valued Martingales Abbasi-Yadkori et al. (2011)). *Let $\{\mathcal{F}_t\}_{t=0}^\infty$ be a filtration. Let $\{\eta_t\}_{t=1}^\infty$ be a real valued stochastic process such that $\eta_t$ is $\mathcal{F}_t$- measurable and $\eta_t$ is conditionally $R$-sub-Gaussian for some $R > 0$, i.e.*

$$
\forall \lambda \in \mathbb{R} \quad \mathbf{E}[e^{\lambda \eta_t} \mid \mathcal{F}_{t-1}] \leqslant \exp\Big(\frac{\lambda^2 R^2}{2}\Big).
$$

*Let $\{\varphi_t\}_{t=1}^\infty$ be a $\mathbb{R}^d$-valued stochastic process such that $\varphi_t$ is $\mathcal{F}_{t-1}$-measurable. Assume $V$ is a $d \times d$ positive definite matrix and for any $t \geqslant 0$ define*

$$
V_t = V + \sum_{s=1}^t \varphi_s \varphi_s^\top \quad S_t = \sum_{s=1}^t \eta_s \varphi_s.
$$

*Then for any $\delta > 0$, with probability at least $1 - \delta$, for all $t \geqslant 0$,*

$$
\|S_t\|_{V_t^{-1}}^2 \leqslant 2R^2 \log\Big(\frac{\det V_t^{1/2} \det V^{-1/2}}{\delta}\Big).
$$

**Lemma I.2** (Determinant Trace Inequality Abbasi-Yadkori et al. (2011)). *Suppose $\{\varphi_s\}_{s=1}^t \subset \mathbb{R}^d$ be such that $\|\varphi_s\|_2 \leqslant L \,\forall s \in [t]$. Let $V_t = \lambda I + \sum_{s=1}^t \varphi_s \varphi_s^\top$ for some $\lambda > 0$. Then*

$$
\det V_t \leqslant (\lambda + tL^2/d)^d
$$

**Lemma I.3** (Abbasi-Yadkori et al. (2011)). *Let $\{\varphi_t\}_{t=1}^\infty$ be a sequence in $\mathbb{R}^d$, $V$ a $d \times d$ positive definite matrix and $V_t = V + \sum_{s=1}^t \varphi_s \varphi_s^\top$. Then*

$$
\log\Big(\frac{\det V_t}{\det V}\Big) \leqslant \sum_{s=1}^t \|\varphi_s\|_{V_s^{-1}}^2.
$$

*Moreover, if $\|\varphi_s\|_2 \leqslant L$ for all $s$ and if $\lambda_{\min}(V) \geqslant \max\{1, L^2\}$, then*

$$
\sum_{s=1}^t \|\varphi_s\|_{V_s^{-1}}^2 \leqslant 2\log\Big(\frac{\det V_t}{\det V}\Big).
$$

**Lemma I.4** (Forsgren (1996))**.** *Let $\Phi$ be a $K \times d$ full column matrix, let $\boldsymbol{\mu}$ be a $K$ dimensional matrix. Let $\boldsymbol{\Lambda}$ be the set of all positive semi-definite diagonal matrices such that $\Phi^\top \Lambda \Phi$ is invertible for any $\Lambda \in \boldsymbol{\Lambda}$, that is*

$$\boldsymbol{\Lambda} = \{\Lambda \in \mathbb{R}^{K \times K} : \Lambda \text{ is diagonal and positive semi-definite} \wedge \Phi^\top \Lambda \Phi \text{ is invertible}\}.$$

*Then the solution to the weighted least-squares problem lies in the convex hull of the basic solutions, that is,*

$$\left(\Phi^\top \Lambda \Phi\right)^{-1} \Phi^\top \Lambda \mu = \sum_{J \in \mathcal{J}(\Phi)} \left(\frac{\det \Lambda_J \det \Phi_J^2}{\sum_{K \in \mathcal{J}(\Phi)} \det \Lambda_K \det \Phi_K^2}\right) \Phi_J^{-1} \boldsymbol{\mu}_J,$$

*where $\mathcal{J}(\Phi)$ is the is the set of column indices associated with non-singular $d \times d$ sub-matrices of $\Phi$.*

**Lemma I.5** (sub-Gaussian Concentration)**.** *Assume $\{X_i - \mu\}_{i=1}^n$ are $n$ independent $\sigma$-sub-Gaussian random variables, then*

$$\mathbb{P}[|\widehat{\mu}_n - \mu| \geqslant \varepsilon] \leqslant 2\exp(-\frac{n\varepsilon^2}{2\sigma^2}),$$

*where $\widehat{\mu}_n = \frac{\sum_{i=1}^N X_i}{n}$.*

**Lemma I.6** (Bernstein inequality)**.** *Let $\{T_i\}_{i=1}^n$ be random variables in $[0,1]$, such that*

$$\sum_{i=1}^n \mathbb{V}[T_i \,|\, T_{i-1}, T_{i-2}, \cdots, 1] = \sigma^2,$$

*then*

$$\mathbb{P}\left[\sum_{i=1}^n T_i \geqslant \mathbb{E}[\sum_{i=1}^n T_i] + \varepsilon\right] \leqslant \exp\left\{-\frac{\varepsilon^2/2}{\sigma^2 + \varepsilon/2}\right\}.$$

