# OpenReview forum: "Bad Values but Good Behavior: Learning Highly Misspecified Bandits and MDPs"
_TMLR — Rejected by TMLR_

### Review · Reviewer_nKr1 · 2025-02-27

**Summary Of Contributions:**

This paper introduces the concept of a robust region, where basic algorithms such as $\epsilon$-greedy, and OFUL-based methods can achieve optimal results despite model misspecification.
In other words, it extends the theoretical guarantees for these well-known algorithms, showing that they remain effective as long as the specified model falls within this robust region of true model.

**Audience:**

Yes

**Broader Impact Concerns:**

None.

**Claims And Evidence:**

Yes

**Requested Changes:**

### General
- The authors use both $K$ and $\text{K}$. (not only $K$, e.g., $d$ in Algorithm 5) I think it is better to unify.

### Section 1
- Example 1: it may be better to use $\approx$ instead of $=$.
---
### Section 2
- In (contextual) Linear bandits: use \citep (with parenthesis) for Dani et al., ... and Agarwal and Goyal..
- In Misspecified Bandits: in the third line, **T**he work of Liu -> **t**he work of Liu.
---
### Section 3
- It would be better to declare that this paper consider the bandit problems with the unique optimal arm in Definition 3.3.
- It would be better to define the matrix induced vector norm in Definition 3.4.
- It may be better to define robust parameter region with feature matrix $\mathbf{\Phi}$, and say we will drop the notation if it is obvious.
- Regret must be defined before the usage, since one generally considers pseudo-regret in multi-armed bandits (regret w.r.t. the best arm) while one may consider contextual regret in contextual bandits (regret w.r.t. the best arm at every round for given context).
- Proof of Theorem 3.19: I think the authors missed $\mu$, i.e., w.l.o.g. assume $\mu\in \mathcal{M}_k(\epsilon)$?
- In the second line after Section 3.4., an $\epsilon$-optimal arm is (missing sentences?) if $\epsilon$ is larger than...
---
### Section 4
- I think the definition of $\text{OPT}(x)$ should be in the main text.
---
### Section 5
- In Function Approximation: $\mathcal{F}$ -> $\mathcal{F}_h$?
- In Algorithm 1, $\mathcal{D}_h$ is not initialized. $\mathcal{D}_H$ should be explicitly defined like $\hat{\theta}$.
- In Robust condition: a weird space exists in front of "Here".
- In Assumption 5.1., it would be better to use (1) or Equation 1.
---
### Proof for $\epsilon$-greedy
Although the proofs in Sections A.1 and B.1 are largely correct, I think one part should be modified.
In the last part, the author uses the lower bound of $t_0$ to derive the upper bound of the form $xe^{-x}$.
When $t_0 \geq t_0' \geq 1$, we obviously have $t_0 e^{-t_0} \leq t_0' e^{-t_0'}$.
However, this may not hold if $t_0' \leq 1$. Therefore, it would be better to use the upper bound on $t_0$ with an order of $\sqrt{t}$, i.e.,
$$ t_0 e^{-t_0} \leq \sqrt{t}e^{-(\sqrt{t+1}-1)}. $$

The use of lower bound may be acceptable in A.1., but in B.1., the lower bound includes the term $p_x$, the probability of context, which can be sufficiently small in extreme case.  In such cases, directly using the lower bound could be problematic.

---
### Proof of Lemma C.6.
- In the first line, saying "if and only if" seems incorrect. The definition of $\mathcal{R}_{\text{OPT}}$ implies $P^{\Lambda_t}$ satisfying for all $t\geq 1$, but not vice versa. (unless $(\Lambda_t) _{t=1}^\infty$ can cover the whole power set).
- The definition of $\Delta_{\text{min}}=\min_{i\in [K]}\Delta_{i}=0$?
---
### Proof of Lemma C.7
- It may be helpful to add one more step in the derivation of the upper bound, between the second equation and third equality:
$$ x^\top V_t^{-1} \sum \varphi_{A_s} \eta_s \leq \Vert x^\top V_{t}^{-1} \Vert_{V_t} \cdot \Vert  \sum \varphi_{A_s} \eta_s \Vert_{V_t^{-1}} = \Vert x \Vert_{V_t^{-1}}\cdots $$
- In the next inequality, $|x||$ -> $\Vert x \Vert$
- The first equations of p.24, the second term of 5th inequality should be the term with $\hat{\mu}$ and $\mu$.
---
### Remark C.9 and Theorem E.5
I am not sure it is appropriate to say "we leave it as an exercise for the reader" in the paper.

---
### Assumption E.3
$l^2$ norm by $L$ instead of $1$?

---

**Strengths And Weaknesses:**

### Strength

- The concept of a robust region is intuitive (but may be obvious).
- Good illustrations with figures to clarify the concept.

### Weakness

As mentioned in Conclusion, it is unclear how to utilize the concept of robust regions.

Since the explicit formulation of robust regions will depend on the algorithms of interest, one naive idea to utilize this concept would be evaluating robustness of algorithm based on area of robust region.
Some algorithms might have smaller robust regions because certain parameters may not always yield the same optimal results due to their inherent randomness (or some properties).

---

> ### Author Response · Authors · 2025-03-04
>
> We express our thanks to the reviewer for going over the paper in such a thoroughness and pointing out corrections that needed to be done.
>
> **General** We agree that the notation of $K$ as the number of arms is consistent with the notation of $\mathbb{R}^K$ and as such we shall have it amended in the revised version.
>
> **Section 1** Will the reviewer kindly point out the exact location of the symbol which seems to be incorrect.
>
> **Section 2** We shall have corrected the usage of citep vs citet, so as to ensure that the citations are as per standards.
>
> **Section 3** Thanks for pointing out the suggestions. These shall have been adopted in the revised version of the paper.
>
> **Section 4** We agree. We shall have included the mentioned definition in the main text and not as a footnote.
>
> **Section 5** Thanks for the pointers. They shall be revised and updated in the revised version
>
> **Proof for $\epsilon$-greedy** Thanks for pointing out the lack of rigor in the proof. We had glanced over this point since, we were analyzing for the asymptotic regime ($T \to \infty$). But, to be rigorous, the inequality holds true for $t \geq (1 + K)^2 - 1$ (in the bandit case) and for $t \geq (1 + \mathrm{A}/p_x)^2 -1$ for the contextual bandit case. In the later case, we assume that each $p_x$, the probability of context is positive (Assumption 4.8). By this statement we mean that the context probability is a positive constant, and thus after sufficient time, the inequality holds true. We shall add this condition in the revised version of the paper.
>
> **Proof of Lemma C.6** Thanks for noticing. It would not be an "if and only if" condition. We shall change this in the revised version.
> The definition of $\Delta_i$ in the context of Lemma C.6, reflects the suboptimal gap in the value function space, that is $\phi_*^\top\theta - \phi_i^\top\theta$, where $\theta$ belongs to $\Theta_\ast$, the robust parameter region.
>
> Note that $\phi_\ast^\top \theta$ and $\phi_i^\top \theta$ belongs to the greedy region $\mathcal{G}^{\ast}$ for all $\theta \in \Theta_\ast$. This is because of the definition of the robust parameter region. In other words, the model estimate of the rewards are themselves falling in the correct greedy regions for robust instances. Now, since the greedy regions are themeselves such that there can be only one unique optimal arm, then the suboptimality gap in the value function space must also be strictly positive, for otherwise, we violating our definition of greedy regions. Stated simply, features which give rise to identical value function representation for all arms, cannot be hoped to give correct predictions based on approximate value functions.
>
> **Lemma C.7** We shall Incorporate the suggestions and typographical errors in the revised version.
>
> **Remark C.9** We shall remove the remark
>
> **Theorem E.5** We shall include the proof in the revised version.

---

### Review · Reviewer_SefY · 2025-03-17

**Summary Of Contributions:**

In this paper, the authors identify robust regions for misspecified bandit instances such that standard bandit algorithms can still work. The main goal of the authors is to show that while the ground truth rewards can be very much different from the estimated rewards using the hypothesis class, the regret can be o(1) as long as the ground truth rewards are in certain regions. The authors analyze several bandit problems/instances (MAB, linear bandits, contextual bandits, MDP) to characterize the robust regions and derive guarantees.

**Audience:**

Yes

**Claims And Evidence:**

Yes

**Requested Changes:**

Please see the weaknesses outlined above.

**Strengths And Weaknesses:**

The main strength of the paper is to provide a characterization on robust observation regions - this region can be non-trivially large and need not be within a small epsilon from the subspace. In other words, Theorem 3.7 and Theorem 4.6 are the most interesting and novel contributions of this work. However, apart from these, the novelty is a bit limited.

However there are several weaknesses of this paper. I list them below one by one:

1. Let me start with what seems like a technical error. In Appendix C, Assumptions C.2 and Assumptions C.3 seems to be contradictory to each other. Here is a simple proof - the trace of the matrix V after d iterations is at most d (from Assumption C.2) - trace is a linear operation and trace of each rank-1 matrix is simply the square of the L2 norm. Trace is also the sum of eigenvalues all of which are positive - sum of d positive number is less than d - therefore the smallest must be less than or equal to 1. The only possibility when there is not a contradiction is when the V matrix is identity.

2. Theorem 3.7 (and therefore Theorem 4.6) seems very strong to me. We need to understand how often it holds and how strong these assumptions are. Can the authors show that for random parameters generated from N(0,1) iid, the conditions are satisfied whp. I think that would go a long way to show that the assumptions are not extremely strong in high dimensions

3. Theorem 4.10 seems to be in the limit whereas Theorem 4.12 seems to be for finite time - I am very skeptical of the guarantees holding for finite time since the margins/boundaries are unknown. Is there something that I am missing? Is theorem 4.12 also in the limit of T going to infinity but phrased incorrectly?

4. In definition 3.6, the region is defined based on all probability vectors in the simplex. However, this definition is unclear to me - is it not possible that a probability vector places all its mass on a single arm - therefore for all the other arms, there is no estimate of the associated mean. In that case, how does the definition hold? Again, I must be missing something

5. Example 2 in Page 2 is unclear. WIll be good to elaborate. Why should the estimate be a convex combination if the model class is non-linear?  To my understanding the convex combination holds only for OLS estimates under the linear model - what am I missing? What is the estimate in this example?

6. (Minor) Contribution 1 needs to be rewritten - What does feature subspace mean exactly?

---

> ### Author Response · Authors · 2025-03-27
> **Addressing Weaknesses**
>
> First, We would like to thank the reviewer for going over the manuscript in quite detail and pointing out mistakes even we had missed.
>
> With regards to the Weaknesses and requested changes:
> We have uploaded a new revised version, which tries to address some of the weakness as you have suggested.
>
> Here we outline the revisions made in order as they were presented above.
>
> 1. The technical error in Appendix C has now been rectified where we have now the assumption $V=I$, the identity matrix
> 2. For Theorem 3.7, we have added an empirical study (in that subsection itself), which might alleviate some of the concerns. Do let us know if you had something else in mind.
> 3. We have brought over the results from the Appendix in place of Theorem 4.2, which hopefully, clarifies the meaning.
> 4, We have clarified the sampling distribution set that we are using. We did this again by bringing in the results from the Appendix to the main section
> 5. We have clarified the model class. Hopefuly this will clarify matters
> 6. We added a clarification about feature subspace.
>
>
> Thanks

---

> > ### Comment · Reviewer_SefY · 2025-04-14
> > **Response**
> >
> > 1. This is confusing. It is not possible to set the matrix V=I every time during the forced exploration phase. It depends on the feature vectors available. There still seems to be a silly inaccuracy here - surely it can be corrected!
> >
> > 2. I thank the authors for providing this plot. I think this is sufficient to alleviate my concern.
> >
> > 3. The authors did not address my concern regarding Example 2 - I asked for a similar analysis as in the linear regression case. What is the exact model estimate in the defined model class and how is it a convex combination?
> >
> > 4. My concerns number 3 and 4 (in original review) has also not been addressed. The same issues still persist and the answers from the authors are vague.

---

> ### Author Response · Authors · 2025-04-14
> **Author response**
>
> **Regarding $V=I$**
>
> Thanks to the reviewer for their patience in following up with this concern. The concern is valid but there is a clear way to resolve this. We propose, in the revision, to have the forced exploration phase to last for $2d$ time instants instead of the original $d$.
>
> With this, $\mathrm{trace}(V)$ is $\sum_{i=1}^{2d}\|\|\phi_i\|\|^2 \leq 2d$. On the other hand, we also have $\mathrm{trace}(V) \geq d \lambda_{\min}(V)$. Thus, we have from the two inequalities that $\lambda_{\min}(V) \leq 2$. This now is consistent with the Assumption C.3, i.e., $\lambda_{\min}(V) \geq 1$. This has now been updated in the manuscript.
>
> We would like to additionally mention that Appendix E ("Weighted Ridge Regression") does away with the need for a forced exploration by LinUCB and instead redefines the robust region in terms of a ridge regression regularization parameter $\lambda$.
>
> **Clarifying Example 2**.
>
> Thanks for your request to make this example more precise. The explicit calculation is presented as follows. Note that the model class is $|y-x| = \epsilon$, and the bandit instance is $(x, y)$ where $y > x + \epsilon$. In this setting, any projection of the bandit instance $(x,y)$ on the model class $|y-x| = \epsilon$ would be just on the line $y = x + \epsilon$. Thus, this problem is reduced to a misspecified bandit instance, which one is trying to model using a linear function, and thus, the argument given in the first example follows. Explicitly,
>
> Let $\mu_1$ be sampled $\alpha$ fraction of times and $\mu_2$ be sampled $1-\alpha$ fraction of times. Then the model estimate for $(\mu_1, \mu_2)$ is $\big(\alpha\mu_1 + (1-\alpha)\mu_2 - (1-\alpha)\epsilon, \alpha\mu_1 + (1-\alpha)\mu_2+\alpha\epsilon\big)$, which is a convex combination of the points $(\mu_1, \mu_1+\epsilon)$ and $(\mu_2-\epsilon, \mu_2)$. We shall include this computation in the revised version.
>
> **With regards to Theorem 4.10 and 4.13**.
> Context: In the present revision, Theorem 4.10 is the same whereas the erstwhile Theorem 4.12 has become Theorem 4.13. Theorem 4.10, as the reviewer rightly notes, is stated in the asymptotic sense as $T \to \infty$, $\lim \frac{Regret}{\sqrt{T}} \to 0$. About Theorem 4.13 (the erstwhile Theorem 4.12), however, we emphasize that it is an anytime (not asymptotic) result valid for all $t$. More clearly, Theorem 4.13 states,   $$ \sum_{s=1}^t \mu_{X_s, \mathrm{OPT}(X_s)} - \mu_{X_s, A_S} \leq \frac{4\sqrt{t}R\Delta_{\max}}{\Delta_{\min}} \sqrt{\log \Big(\frac{(1 + t/d)^{d/2}}{\delta}\Big)}\sqrt{\log{(1 + t/d)^d}}$$
>
> with probability at least $1-\delta$ for all $t\geq 1$. Here $\Delta_{\max} = \max_{(x,a) \in X \times A} \mu_{x, \mathrm{OPT}(x)} - \mu_{x,a}$ and $\Delta_{\min}$ is an instance-dependent constant defined in Lemma 4.12. The instance-dependent constant $\Delta_{\min}$, is not a function of time and (no matter how small) is a positive constant. This is incorporated into the revision. We see this result as a novel robustness property of LinUCB that has been discovered, which we probably take to be the reason behind the reviewer's skepticism. We will be happy to engage in more discussion if the reviewer can point out the exact source of their confusion with this result.
>
> **Clarifying the Sample Distributions**.
>
> As pointed out in the revision (definition of $\mathcal{P}(\Delta_K)$ in Definition 3.7), only those probability distributions are allowed, which ensures $\Phi^\top\Lambda\Phi$ are invertible. Namely only those sampling distributions
> $\lambda_{i=1}^K$ are allowed such that $\lambda_{i=1}^K \in \Delta_K\;$ and $\Phi^\top\Lambda\Phi$ is invertible, where $\Lambda = \mathrm{diag}(\lambda_{i=1}^K)$.
> This definition, consequently, does not allow the Dirac type distributions the reviewer raises, but allows for close approximations, in the sense that $1-\epsilon$ mass is placed on one arm, whereas all the other arms have a joint mass of $\epslion$, uniformly. To address the Dirac type of distributions, we invite the reviewer to peruse section E (notably Definition E.1).

---

> > ### Author Response · Authors · 2025-04-15
> > **More clarifications on Example 2**
> >
> > Consider the following extension of the previous example for a non-linear *parameterized* model class.
> > The class is defined as the set of all $(x,y) \in \mathbb{R}^2$, which satisfies the following piecewise constraints
> > $$
> > (x(t), y(t)) = \\
> > \begin{array}{ll}
> > x_0 + 400t\vec{u}, & \text{if } 0 \leq t \leq \frac{1}{4} \\\\
> > x_0 + 100\vec{u} - (2t - 1) \cdot 2\varepsilon\vec{v}, & \text{if } \frac{1}{4} \leq t \leq \frac{1}{2} \\\\
> > x_0 + 100\vec{u} - 2\varepsilon\vec{v} - 100(4t - 2)\vec{u}, & \text{if } \frac{1}{2} \leq t \leq \frac{3}{4} \\\\
> > x_0 - 2\varepsilon\vec{v} + 2\varepsilon(4t - 3)\vec{v}, & \text{if } \frac{3}{4} \leq t \leq 1
> > \end{array}
> > $$
> > where $\vec{u} = (1, 1)$ and $\vec{v} = (-1, 1)$, and $x_0 = [-50-\epsilon, -50+\epsilon]^\top$. This represents a long narrow rectangle oriented along the $y=x$, with length $100$ and width $2\epsilon$. The previous example can be considered a zoomed-in picture of this model class.
> >
> > Consider, again the bandit instance $[\mu_1, \mu_2] = [-10, 10]$ as in the previous example. If arm-$1$ has been sampled $\alpha_1$ fraction of times and arm $2$ has been sampled $\alpha_2$ fraction of times, it is clear the the estimate $\hat{t} = \frac{\alpha_1(40+\epsilon) + \alpha_2(60 - \epsilon)}{400}$, can be written as a convex combination of $\frac{40+\epsilon}{400}$ and $\frac{60-\epsilon}{400}$. The corresponding model estimate is $\begin{bmatrix}
> >     -10\alpha_1 + (10-2\epsilon)\alpha_2 \\
> >     (-10 + 2\epsilon)\alpha_1 + 10\alpha_2
> > \end{bmatrix}\;,$ which itself is a convex combination of the extreme points $\begin{bmatrix}
> >     -10\\
> >     -10+2\epsilon
> > \end{bmatrix}$ and $\begin{bmatrix}
> >     10 - 2\epsilon\\
> >     10
> > \end{bmatrix},$ corresponding to the estimates $\frac{40+\epsilon}{400}$ and $\frac{60-\epsilon}{400}$ respectively.
> > We have included this example into the new revised version.

---

### Review · Reviewer_SnWg · 2025-03-19

**Summary Of Contributions:**

This paper studies mis-specified bandits and MDPs. The key knowledge gap that this paper tries to fill is that previous papers like Liu et al. (2023) defined misspecification with respect to gaps, whereas this paper defined them with respect to features and proposes algorithms that can minimize regret/robust to these misspecifications. They define the misspecification as the vector of true means, the greedy region as the region that shares the same optimal arm, and finally, a robust region which means that under some perturbation of the sampling strategy, the final estimation of the optimal arm (or the reward vector) is close to the true with high probability. To translate this in the linear bandit setting the reward means, the greedy and robust region are now defined with respect to the feature matrix, and the goal is that the estimation of $\hat{\theta}$ and $\theta$ must lie within the same robust observation region. In essence, the misspecification can be arbitrary and to get a feasible solution (algorithm) they have to fix an instance in the robust observation region that is solvable.  I believe this formulation of misspecification is novel. They prove that for any instance (i.e., the vector of true mean arm rewards) in the robust observation region, both (i) the $\varepsilon$-greedy algorithm, with least-squares parameter estimation and an exploration rate of $1 / \sqrt{t}$ in each round $t$, and (ii) the LinUCB (or OFUL) algorithm, achieve $O(\sqrt{T})$ cumulative regret in time $T$. I did not look into the MDP formulation as I believe the contribution to the bandit/linear/contextual setting is sufficient.

**Audience:**

Yes

**Broader Impact Concerns:**

Not applicable.

**Claims And Evidence:**

Yes

**Requested Changes:**

1. I understand the standard misspecified domain defined on the gaps. However, this paper proposes a new mis-specification regime in definitions 3.3-3.6. I find the setting slightly convoluted. It will be great if the authors start with a motivation as to why this specific setting should be studied. I suggest at the end of each of these definitions write a few lines as to the purpose of it. For example at the end of Definition 3.4, state why $\mathbf{P}^{\boldsymbol{\Lambda}}(\boldsymbol{\mu})$ matters. Similarly for

2. The paper would really benefit from some restructuring. First, try to put back the algorithms from Appendix into the main paper. At the very least put Remark B.1 in the main paper detailing the proof technique of the greedy algorithm. The same thing for the LinUCB algorithm. Currently, it is very hard to follow the LinUCB proof with so many assumptions (prima facie these assumptions are standard) in the appendix.

3. I think the problem formulation would be more expanded. Then the multi-armed bandit, linear, and contextual bandit sections are sufficient (with the two interesting examples shown). The authors should seriously rethink whether putting the MDP result is buying them anything extra. Maybe put more emphasis on the bandit sections, expanding discussion on the result (see (2)).

4. As I stated several areas need more explanations (see 1,2,3). I also feel that the paper needs more experiments. I understand that this is a theoretical paper, but even bandit papers these days provide really compelling experiments. I feel the non-linear approximation section can be further augmented with really nice experiments.

Decision: I think the paper needs significant restructuring and a better story telling.

**Strengths And Weaknesses:**

Strengths:

1. The paper proposes a new misspecification/robustness regime. Here the mis-specification can be arbitrarily large and defined on features of the arms as opposed to gaps.
2. They propose two algorithms $\varepsilon$-greedy algorithm, and the LinUCB algorithm and show that under some conditions they reach the regret bound of their counterparts in well-specified regimes.
3. The MDP section may have some appeal, but I have not checked it more carefully.

Weakness:
1. The paper needs more motivation regarding the specific setting they are proposing.
2. The writing and the flow of the paper need to improve significantly.
3. I think the paper is trying to cover too much and needs to be condensed a bit.
4. Several areas need more explanation and needs more experiments.

---

> ### Author Response · Authors · 2025-03-27
> **Addressing requested changes**
>
> We would like to thank the reviewer for the suugested changes. We have added the revised manuscript incorporating most of the changes requested. We present here a small summary of the changes made. Do let us know of what you think.
>
> 1. We tried to incorporate a few clarifications about the motivation and nature of the definitions that we have made, as the reviewer suggested, in the bandits section
> 2. We have brought over major material from the Appendix (espically for LinUCB), while being careful not to add too much technical detail in the main paper, as suggested.
> 3. We feel that the MDP section is relevant as well, as it adds new results.
> 4. While we agree that more experiments would have been nice, we were limited to the ones that were presented in Appendix F. We found that constructing non trivial examples for the non-linear function approximation to be too difficult and we didnt want to settle for trivial examples. We found that our example for linear functions were non trivial which we have added in App F.

---

### Decision · Action_Editor_DUbs · 2025-04-21

**Recommendation:** Reject

**Comment:**

This paper studies model misspecification in some of the most basic frameworks for online learning. Roughly speaking, the authors identify conditions under which model misspecification does not affect learning algorithms and then reanalyze existing algorithms. I like the main idea of this paper a lot.

The common denominator in all reviews was that the paper needs a revision because too many things are unclear. The authors revised the paper. This addressed only some concerns of the reviewers: two reviewers weakly support acceptance of the paper and one suggests a rejection. To break the tie, I read the paper until page 8 and here are my comments:

* Pages 5-6: I kept wondering why all these claims for a fixed $\mu$ are needed since we will estimate $\mu$ later.

* Definition 3.5: "the the set" should be "the set".

* Theorem 3.12: Figure 2 and its description on page 7 suggest that the theorem can be used to construct $\mathcal{R}_k$. It is not clear what this construction is. Make the algorithm / linear program clear.

* Page 7: It is not clear how Figure 2 is plotted. Specifically:

  * How is $\Phi$ sampled?

  * To estimate the number of robust regions, it seems that sampling of $\theta$ is needed. How is $\theta$ sampled? If $\theta$ is not sampled, then make clear what the algorithm for constructing $\mathcal{R}_k$ (Theorem 3.12) is.

  * Figure 2 still does not answer how $\mathcal{R}_k$ look. What are these regions?

* Page 8: "SaubGaussian nature" should be "sub-Gaussian nature". "sub=optimality gap" should be "sub-optimality gap".

* Theorem 3.20: Key assumptions to understand the theorem are in Appendix and not discussed.

I agree with the reviewers that this paper is hard to read. I suggest that the authors take time to properly revise the paper, including fixing all the typos.

**Audience:**

This paper studies model misspecification in some of the most basic frameworks for online learning. No problem with audience.

**Claims And Evidence:**

This is a theory paper with precisely stated claims. The problem is that the paper is not easy to read, partially because it contains too much content with too few discussions. Therefore, while there is a plenty of evidence, it is neither convincing nor clear.

**Resubmission Of Major Revision:**

The authors may consider submitting a major revision at a later time.